# Generalization and Scaling Laws for Mixture-of-Experts Transformers

## Abstract

We develop a theory of generalization and scaling for Mixture-of-Experts (MoE) Transformers that cleanly separates *active* per-input capacity from *routing* combinatorics. Conditioning on fixed routing patterns and union-bounding across them, we obtain a sup-norm covering-number bound whose metric entropy scales with the active parameter budget and incurs a MoE-specific overhead. Combining this with a standard ERM argument for squared loss we provided a generalization bound under a $d$-dimensional manifold model ($d$ is the intrinsic dimension of the training data) and $C^\beta$ targets, showing that approximation and estimation trade off in the same way as in dense networks once active parameters are counted appropriately. We further prove a constructive approximation theorem for MoE architectures, demonstrating that accuracy can be improved either by scaling active capacity or by increasing the number of available experts, with the better of the two mechanisms prevailing. From these results we derive neural scaling laws, covering model scaling, data scaling and compute–optimal tradeoffs. The theory highlights that enlarging the expert pool at fixed sparsity influences performance only through a mild logarithmic routing term, whereas increasing active capacity per input drives the main gains in generalization and approximation. These insights provide principled guidance for the design of efficient sparse Transformer systems and clarify the fundamental tradeoffs underlying their empirical scaling behavior.

## 1 Introduction

Mixture-of-Experts (MoE) Transformers enable *conditional computation*: for each token, only a small subset of experts is activated, so the *active* parameters per input are much smaller than the *total* parameters of the model. This yields a practical win compute roughly proportional to the number of active experts while preserving a very large total capacity for specialization. Despite striking empirical gains, a theory that *separates* the benefits of active capacity from the *overheads of routing* has been missing. In particular, existing scaling laws for dense models track error as a function of total parameters, dataset size, and compute, whereas MoE models demand a refined accounting: (i) approximation should scale with the *active* parameter budget, not the total; (ii) generalization should reflect the multiplicity of routing patterns and (iii) compute–optimal training should couple data and *active* capacity rather than total parameters. This paper closes that gap by providing uniform generalization and approximation results for MoE Transformers that expose the distinct roles of active parameters and routing combinatorics. Classical bounds indexed by the *ambient* dimension $D$ predict unrealistically slow rates for high-dimensional inputs, whereas empirical scaling for Transformers is far faster. A growing body of theory and evidence shows that when data concentrate on a $d$-dimensional manifold, both approximation and estimation rates and thus the observed scaling exponents are governed by the *intrinsic* dimension $d$ rather than $D$. Recent work (Havrilla & Liao, 2024) develops approximation and statistical theories for Transformers on intrinsically low-dimensional data, predicting data/model scaling exponents that depend explicitly on $d$ and validating these predictions on language modeling benchmarks. Motivated by these findings, we adopt $d$ as the geometric control parameter throughout: it aligns theory with practice, explains faster-than-$1/D$ behavior, and yields MoE design rules that are sensitive to the data geometry (e.g., how active capacity should scale with tokens when $d$ is small relative to $D$).

The main contributions of our work can be summarized as follows:

- **Separation of capacity and routing.** We prove a covering-number bound in which metric entropy splits additively into an *active-capacity* term and a *routing* term $R_{\text{route}}$.

- **Approximation and Uniform generalization bound.** We give a manifold-based, k-sparse partition-of-unity construction of the approximation exposing two accuracy routes: scaling active capacity or increasing experts. The empirical risk minimization (ERM) under squared loss cleanly separating approximation, estimation, and routing overhead.
- **Scaling laws and compute optimality.** We recover dense-model exponents measured against *active* capacity: model scaling (in the approximation regime), data scaling and compute error scaling under $C \approx n \, N_{\text{eff}}$. We characterize the optimal number of active experts $k^\star$ (up to architecture/log factors) and show that the expert pool $M$ enters only logarithmically via routing, motivating $M = \Theta(k)$.
- **Empirical validation on MoE LMs.** Beyond the purely theoretical development, we also validate our framework on Mixture-of-Experts language models. We train small/medium-scale MoE Transformers on three open source data sets, estimate the intrinsic dimension $d$ and effective smoothness $\beta$, and show that the empirical model- and data-scaling exponents $(\widehat{\alpha}_N, \widehat{\alpha}_D)$ are consistent with our theoretical predictions.
- **Design guidance.** Practical levers follow directly: keep $\log(e \, M/k) = \Theta(1)$ (avoid $M \gg k$), operate near $k^\star$, and mitigate routing on long sequences by reducing $\ell$ or sharing gates yielding efficient MoE training at fixed compute.

## 2 RELATED WORK

**Transformers: expressivity, approximation, and generalization.** The Transformer architecture (Vaswani et al., 2017) implements content-addressable computation through multi-head self-attention and positionwise MLPs. Formal expressivity results show that (suitably parameterized) Transformers are universal approximators for a broad class of sequence-to-sequence maps (Yun et al., 2020), with positional encodings playing a crucial role in breaking permutation symmetries and enabling length-dependent behaviors. Subsequent analyses refine which architectural ingredients control approximation power number of heads, attention bandwidth, depth vs. width trade-offs, and the effect of pre-norm vs. post-norm residual designs often revealing that depth amplifies compositional expressivity while LayerNorm stabilizes Lipschitz constants across layers. From the learning-theoretic perspective, generalization guarantees for deep networks leverage norm-based capacity measures (e.g., spectral-complexity/margin bounds, path norms) (Bartlett et al., 2017; Neyshabur et al., 2015) together with classical complexity tools (Rademacher/Gaussian complexities and covering numbers) (Bartlett & Mendelson, 2002b). Our analysis adopts this toolbox but tailors it to sparse Transformers by (i) conditioning on a fixed routing pattern so that the model reduces to a smooth active subnetwork and (ii) union-bounding over routing patterns to account for combinatorial choices. For approximation, we rely on deep ReLU theory that provides near-optimal rates $\|f - \mathcal{N}\|_\infty \lesssim N^{-\beta/d}$ for $C^\beta$ functions on $d$-dimensional domains (Yarotsky, 2017; Schmidt-Hieber, 2020; Petersen & Voigtlaender, 2018). We lift these Euclidean results to data on manifolds by working in local charts and gluing approximants with partitions of unity, which pairs naturally with the MoE router's $k$-sparse mixing and underlies our approximation theorem.

**Mixture-of-Experts (MoE): conditional computation and systems.** Sparsely-gated MoE layers (Shazeer et al., 2017) activate only a small subset of experts per token, decoupling *total* parameters from per-example *active* compute. This idea enabled large sparse Transformers such as GShard (Lepikhin et al., 2021), Switch Transformers (Fedus et al., 2022), and GLaM (Du et al., 2022); in vision, V-MoE validated similar benefits for ViTs (Riquelme et al., 2021). Practical deployments introduced load-balancing losses and alternative dispatchers (BASE layers, expert-choice routing) to reduce hotspotting and improve stability, and specialized runtimes (DeepSpeed-MoE, Tutel) to scale training/inference (Lewis et al., 2021; Zhou et al., 2022; Rajbhandari et al., 2022; Hwang et al., 2023). Recent systems emphasize expert specialization and the gap between total and active parameters e.g., Mixtral (8×7B, top-2 routing) and DeepSeek-V3 (hundreds of billions total, tens of billions active) (Jiang et al., 2024; DeepSeek-AI, 2024). These trends motivate our theoretical focus on *active* per-input capacity as the relevant approximation budget and on an explicit routing-combinatorics term that scales like $k \log(eM/k)$ in covering-number bounds. In our framework, hard top-$k$ and softmax gating with temperature/load-balancing behave similarly at the level of uniform bounds as long as the effective sparsity is $k$.

**Intrinsic-dimension theory for Transformers.** Our analysis builds on work showing that Transformer scaling is governed by the *intrinsic* rather than ambient dimension when data concentrate on low-dimensional structure. In particular, Havrilla & Liao (2024) develop an approximation statistical framework for Transformers on intrinsically low-dimensional data, deriving data/model exponents that depend explicitly on the manifold dimension and validating these predictions empirically. We adopt this viewpoint to interpret MoE:

active per-token capacity plays the role of the effective model-size axis, while routing acts as an additive overhead. Our analysis builds on their dense-Transformer framework and extends it to MoE with routing. Theorem 3.1 mirrors their approximation result but constructs MoE layers with top-$k$ routing and separates active capacity from total parameters. This aligns our MoE scaling laws with intrinsic-dimension theory recovering dense-model exponents measured against *active* capacity and clarifying when routing can shift constants without changing rates.

**Scaling laws: dense vs. MoE.** Dense language models exhibit empirical power-law scaling of loss with model size, data, and compute (Hestness et al., 2017; Kaplan et al., 2020); the Chinchilla study argues for compute-optimal training by scaling model and data jointly under a fixed budget (Hoffmann et al., 2022). For MoE, recent empirical work proposes *MoE-specific* scaling formulations that disentangle *total* from *active* parameters and incorporate routing effects. Fine-grained MoE scaling analyzes expert granularity, token/expert budgets, and the efficiency frontier as expert shards become smaller and more numerous (Krajewski et al., 2024). Upcycling laws study converting dense checkpoints into MoE while retaining predictable scaling behavior (Liew et al., 2025). Joint dense–MoE laws integrate active parameters, dataset size, and expert count to explain memory/throughput trade-offs (Ludziejewski et al., 2025), and unified efficient-MoE laws relate expert activation ratios, granularity, and compute (Tian et al., 2025). Large-scale case studies (e.g., extending fine-grained MoE beyond 50B parameters) provide further evidence that MoE can match or exceed dense scaling at comparable or lower active compute (Krajewski et al., 2025). Our theory complements these findings: we recover the dense-network exponents but measure them against the *active* parameter budget $N_{\text{eff}}$, and we make explicit a routing overhead proportional to $k \log(eM/k)$ that shifts constants and determines finite-sample crossovers (e.g., long sequences or large $M \gg k$). This yields compute–optimal prescriptions consistent with practice grow data and *active* capacity together while clarifying when adding experts primarily changes constants rather than exponents.

## 3 MoE Transformer Model and Approximation Theory

### 3.1 MoE approximation bound under Hölder smoothness

**Theorem 3.1.** *Let $\mathcal{M} \subset \mathbb{R}^D$ be a compact $d$-dimensional $C^1$ submanifold and $f \in C^\beta(\mathcal{M})$ with Hölder norm $\|f\|_{C^\beta} \leq B$, where $\beta > 0$. Consider the MoE transformer class $\mathcal{T}_{\text{MoE}}$ (see Definition D.2 in appendix) with $L_T$ blocks, sequence length $\ell$, and an MoE FFN sublayer per block with $M$ experts, of which at most $k$ are active per token, and each expert is a ReLU MLP of depth $L_{\text{FFN}}$, width $w_{\text{FFN}}$ and parameter budget $\Pi_{\text{exp}}$; attention/non-expert parameters per block are $\Pi_{\text{attn}}$. Assume the router can implement a $k$-sparse partition of unity and weights are bounded by $\kappa$. We define the* active attention+expert budget $N_{\text{eff}} := L_T \Pi_{\text{attn}} + L_T k \Pi_{\text{exp}}$. *Then we proved in appendix F that there exist constants $C, c > 0$ depending on $(B, \beta, d, \mathcal{M})$ and architecture polynomials in $(L_T, w_{\text{FFN}}, L_{\text{FFN}}, \kappa)$ such that*

$$\inf_{T \in \mathcal{T}_{\text{MoE}}} \|T - f\|_\infty^2 \ \leq \ C \cdot \min\left\{ N_{\text{eff}}^{-2\beta/d} \ , \quad M^{-2\beta/d} \right\} \tag{1}$$

**Remark 3.2** (Dependence on $d$ and constants). *The exponent $-2\beta/d$ is the classical minimax exponent for $C^\beta$ approximation on a $d$-dimensional manifold. The constants $C, c$ in Theorem 3.1 depend on $(B, \beta, d, \mathcal{M})$ and architecture polynomials; standard manifold approximation arguments show that this dependence is at most polynomial in $d$ for manifolds with bounded regularity. The only exponential-in-$d$ behavior is the unavoidable covering-number term that produces the exponent itself.*

Appendix F provides a constructive proof of Theorem 3.1 using manifold charts and a $k$-sparse partition of unity. (i) We cover $\mathcal{M}$ with coordinate patches $\{U_\nu\}_{\nu=1}^N$ of diameter $\leq r$ and bounded overlap $s_0(d)$, and construct a smooth partition of unity $\{\varphi_\nu\}$ subordinate to the cover; volume regularity gives $N \asymp r^{-d}$. (ii) On each chart, $f$ is locally approximated by its degree-$\lfloor \beta \rfloor$ Taylor polynomial $P_\nu$ (Lee, 2013). (iii) Each expert MLP (ReLU) uniformly approximates $P_\nu$ to accuracy $\mathcal{O}(r^\beta)$ using a fixed architecture $(L_{\text{FFN}}, w_{\text{FFN}})$ and parameter budget $\Pi_{\text{exp}}$, by invoking near-optimal $C^\beta$ ReLU approximation results in $d$ dimensions (Yarotsky, 2017; Schmidt-Hieber, 2020; Petersen & Voigtlaender, 2018). Parameter-magnitude constraints $\|\theta\|_\infty \leq \kappa$ are enforced by standard layer-rescaling techniques (Xiong et al., 2020; Ba et al., 2016). (iv) The router implements a $k$-sparse mixture by outputting nonnegative weights $w_\nu(x)$ with at most $k$ nonzeros per $x$; choosing $k \geq s_0(d)$ suffices to uniformly approximate the partition $\{\varphi_\nu\}$, matching standard top-$k$ MoE gating schemes (Shazeer et al., 2017; Fedus et al., 2022).

## 3.2 MoE generalization bound

**Theorem 3.3.** *Let $\mathcal{T}(\varepsilon)$ be a class of MoE Transformers such that for every $f \in C^\beta(\mathcal{M})$ there exists $T \in \mathcal{T}(\varepsilon)$ with $\|T - f\|_{L^2(Q)} \leq \varepsilon$. Then the empirical risk minimization (ERM) $\hat{T}_n \in \mathcal{T}(\varepsilon)$ satisfies*

$$\mathbb{E}\|\hat{T}_n - f\|_{L^2(Q)}^2 \leq \varepsilon^2 + \tilde{O}\left(\frac{N_{\text{eff}}}{n} + \frac{L_T \ell k \log(eM/k)}{n}\right).$$

Here $\tilde{O}(g)$ hides polylogarithmic factors: $\tilde{O}(g) = O\big(g \cdot \text{polylog}\big(n, N_{\text{eff}}, L_T \ell k, M/k, \kappa, R, M_0\big)\big)$.

**Corollary 3.4** (Generalization bound with MoE approximation rate)**.** *Combining Theorem 3.2 and the previous theorem with $\varepsilon^2 \asymp N_{\text{eff}}^{-2\beta/d}$ gives*

$$\mathbb{E}\|\hat{T}_n - f\|_{L^2(Q)}^2 \lesssim N_{\text{eff}}^{-2\beta/d} + \frac{N_{\text{eff}}}{n} + \frac{L_T \ell k \log(eM/k)}{n},$$

*up to polylogarithmic factors.*

Where N is shorthand for $N_{\text{eff}}$; in the sequel we write $N_{\text{eff}}$ explicitly.

**Remark 3.5** (Looseness of the routing union bound)**.** *The factor $L_T \ell k \log(eM/k)$ comes from a worst-case union bound over all top-$k$ routing patterns,*

$$\log|\Pi| \leq L_T \ell k \log(eM/k).$$

*This is conservative but enters only logarithmically into the metric entropy and generalization bound, affecting constants rather than exponents. In practice, router specialization reduces the effective number of routing patterns, so the bound is expected to be loose but safe.*

Theorem 3.3 is proved in appendix E. As in dense case, the rate exponents are governed by the intrinsic dimension $d$ and smoothness $\beta$, but the *model-size axis* is the active parameter budget $N_{\text{eff}}$ rather than total parameters; routing enters only via a logarithmic overhead in $M/k$. In the proof, we proceed by (i) conditioning on a fixed top-$k$ routing pattern so the MoE reduces to a smooth active subnetwork, (ii) proving Lipschitz bounds for attention and FFN modules under parameter perturbations, (iii) covering the active-parameter cube for each pattern $\pi$ and (iv) union-bounding over routing patterns to add $L_T \ell k \log(eM/k)$.

**Lemma 3.6** (MoE covering number)**.** *Let $\mathcal{X} \subset \mathbb{R}^D$ be compact with $\|x\|_\infty \leq M_0$. Consider the Mixture-of-Experts transformer class $\mathcal{T}_{\text{MoE}}(D, M, k, L_T, L_{\text{FFN}}, w_{\text{FFN}}, d_{\text{emb}}, m, \kappa, R)$ defined as in Definition D.2, except that each feed-forward sublayer is an MoE layer with $M$ experts and the router selects (hard) top-$k$ experts per token per layer. Assume each learned parameter entry satisfies $\|\theta\|_\infty \leq \kappa$ and each network output is bounded by $\|T(x)\|_\infty \leq R$.*

*Let $\Pi_{\text{attn}}$ be the number of scalar parameters in the attention / non-expert parts per block and let $\Pi_{\text{exp}}$ be the number of scalar parameters of a single expert MLP (depth $L_{\text{FFN}}$, width $w_{\text{FFN}}$, input/output dimension $d_{\text{emb}}$). Then for any $\delta \in (0,1)$ the covering number of $\mathcal{T}_{\text{MoE}}$ under the sup-norm satisfies*

$$\log N\big(\delta, \mathcal{T}_{\text{MoE}}, \|\cdot\|_\infty\big) \leq C_1\Big(\Pi_{\text{attn}} + L_T k \Pi_{\text{exp}}\Big)\log\Big(\frac{C_2 \kappa R M_0}{\delta}\Big) + C_3 L_T \ell k \log\Big(\frac{eM}{k}\Big) \tag{2}$$

*for absolute constants $C_1, C_2, C_3 > 0$ depending polynomially on $d_{\text{emb}}, w_{\text{FFN}}, m, L_T, L_{\text{FFN}}, \ell$ (and we may hide small poly-log factors in the $\tilde{O}(\cdot)$ version).*

Lemma 3.6 is proved in appendix H. The proof conditions on a fixed top-$k$ routing pattern so that each block's MoE layer reduces to a deterministic active subnetwork and then union-bounds over all routing patterns. For a fixed pattern, we establish a *parameter–to–function* Lipschitz bound by composing module-wise sensitivities for multi-head attention and FFNs using basic matrix norm inequalities and the row-softmax Lipschitz property, with LayerNorm controlling scale across residual connections (Vaswani et al., 2017; Ba et al., 2016). An $\eta$-grid of the parameter cube $[-\kappa, \kappa]^{N_{\text{active}}}$ therefore gives a $\delta$-cover of the corresponding function subclass, giving the first term in equation 2 in the usual covering-number style (Bartlett & Mendelson, 2002b).

## 4 Neural Scaling Laws

We derive explicit scaling laws for Mixture-of-Experts transformers that expose the dependence on sample size $n$, number of experts $M$, number of *active* experts per token $k$, and parameter budgets. Throughout we use the generalization bound from theorem 3.3 in the shorthand form

$$\mathbb{E}\,\|\hat{T}_n - f\|^2_{L^2(Q)} \;\lesssim\; \underbrace{N_{\text{eff}}^{-2\beta/d}}_{\text{approximation}} \;+\; \underbrace{\frac{N_{\text{active}}}{n}}_{\text{estimation}} \;+\; \underbrace{\frac{R_{\text{route}}}{n}}_{\text{routing}} \tag{3}$$

where $N_{\text{active}} := N_{\text{attn}} + N_{\text{exp}} = L_T\Pi_{\text{attn}} + L_T k\Pi_{\text{exp}}, \qquad R_{\text{route}} := L_T\ell k \log\left(\frac{eM}{k}\right).$

### 4.1 Data scaling

Lets study how test error decreases with the number of samples $n$ when the architecture is fixed or mildly tuned with $n$. Ignoring $R_{\text{route}}$ for the moment (we reintroduce it in §4.4), we balance approximation and estimation by minimizing, over the active budget $N_{\text{eff}}$. Differentiating and setting to 0 gives

$$-\frac{2\beta}{d}N_{\text{eff}}^{-2\beta/d-1} + \frac{c}{n} = 0 \quad\implies\quad N_{\text{eff}}^\star \asymp \left(\frac{n}{c}\right)^{\frac{d}{2\beta+d}} \;\Rightarrow\; , N_{\text{eff}}^\star \;\asymp\; n^{\frac{d}{2\beta+d}}$$

Substituting $N^\star$ back we obtain the *data scaling law*

$$\mathbb{E}\,\|\hat{T}_n - f\|^2_{L^2(Q)} \;\asymp\; n^{-\alpha_D}, \qquad \alpha_D = \frac{2\beta}{2\beta+d} \tag{4}$$

This exponent matches the dense case (Havrilla & Liao, 2024); the MoE twist is that $N_{\text{eff}}$ is the *active* budget per input (depending on $k$), not the total parameter count.

### 4.2 Model scaling

We quantify how the error scales with the *active parameter budget* when the sample size $n$ is fixed.

**Proposition 4.1** (Model-scaling law in the approximation-dominated regime)**.** *For a fixed $n$, lets suppose that $N_{\text{eff}}$ lies in the regime where the estimation and routing terms are dominated by the approximation term, namely there exist absolute constants $c_1, c_2 > 0$ such that*

$$\frac{N_{\text{active}}}{n} \;\leq\; c_1\,N_{\text{eff}}^{-2\beta/d} \qquad and \qquad \frac{R_{\text{route}}}{n} \;\leq\; c_2\,N_{\text{eff}}^{-2\beta/d} \tag{5}$$

*Then we get*

$$\mathbb{E}\,\|\hat{T}_n - f\|^2_{L^2(Q)} \;\asymp\; N_{\text{eff}}^{-\alpha_N}, \qquad \alpha_N = \frac{2\beta}{d} \tag{6}$$

**Proposition 4.2** (Where the approximation-dominated regime holds)**.** *Since $N_{\text{active}} \asymp N_{\text{eff}}$ (within architecture-dependent constants), the first condition in 5 is equivalent to*

$$N_{\text{eff}}^{1+2\beta/d} \lesssim n \quad\Longleftrightarrow\quad N_{\text{eff}} \lesssim n^{\frac{d}{2\beta+d}} =: N_{\text{eff}}^\star(n)$$

*i.e. below the* estimation crossover $N_{\text{eff}}^\star(n)$. *The routing condition reads*

$$R_{\text{route}} \;\lesssim\; n\,N_{\text{eff}}^{-2\beta/d} \quad\Longleftrightarrow\quad N_{\text{eff}} \;\lesssim\; \left(\frac{n}{L_T\ell k \log\left(\frac{eM}{k}\right)}\right)^{\frac{d}{2\beta}} =: N_{\text{eff}}^\star(n, M, k, L_T, \ell)$$

*Therefore the model-scaling law equation 6 holds uniformly whenever*

$$N_{\text{eff}} \;\leq\; \min\big\{\, N_{\text{eff}}^\star(n),\; N_{\text{eff}}^\star(n, M, k, L_T, \ell)\,\big\} \tag{7}$$

**Remark 4.3** (Crossover and beyond)**.** *At the estimation crossover $N_{\text{eff}} \approx N_{\text{eff}}^\star(n)$, the approximation and estimation terms balance:*

$$N_{\text{eff}}^{-2\beta/d} \;\approx\; \frac{N_{\text{eff}}}{n} \quad\Rightarrow\quad N_{\text{eff}} \asymp n^{\frac{d}{2\beta+d}}, \qquad \mathbb{E}\,\|\hat{T}_n - f\|^2_{L^2(Q)} \asymp n^{-\frac{2\beta}{2\beta+d}}$$

*For $N_{\text{eff}} \gg N_{\text{eff}}^\star(n)$ (holding $n$ fixed), the bound is dominated by $N_{\text{active}}/n \asymp N_{\text{eff}}/n$ (plus routing), so increasing $N_{\text{eff}}$ worsens the bound. Thus equation 6 describes the* subcritical *(approximation-limited) regime, analogous to the dense case (Havrilla & Liao, 2024) but with $N_{\text{eff}}$ the* active *parameter budget.*

**Corollary 4.4** (Model-scaling with experts-only approximation)**.** *If attention does not contribute to approximation (so $N_{\text{eff}} = N_{\text{exp}} = L_T k\Pi_{\text{exp}}$), then in the subcritical regime equation 7*

$$\mathbb{E}\,\|\hat{T}_n - f\|^2_{L^2(Q)} \;\asymp\; (L_T k\Pi_{\text{exp}})^{-\frac{2\beta}{d}} \tag{8}$$

### 4.3 Optimizing the number of active experts $k$

Let $A := L_T \Pi_{\text{exp}}$ and absorb the attention budget into constants. With experts dominate approximation, $N_{\text{eff}} \propto k$, the bound equation 3 reduces to the function of $k$:

$$\mathcal{E}(k) \lesssim \underbrace{(Ak)^{-2\beta/d}}_{\text{approx}} + \underbrace{\frac{Ak}{n}}_{\text{estimation}} + \underbrace{\frac{L_T \ell\, k\, \log(eM/k)}{n}}_{\text{routing}} \tag{9}$$

**Ignoring the log first.** Lets drop $\log(eM/k)$ and treat it as a slowly varying constant near the optimizer. Minimize $g(k) = (Ak)^{-2\beta/d} + \frac{\tilde{B}}{n}k$ over $k > 0$. Differentiating gives (up to architecture constants)

$$k^\star \asymp n^{\frac{d}{2\beta+d}} \quad \text{and} \quad \mathcal{E}(k^\star) \asymp n^{-\frac{2\beta}{2\beta+d}} \tag{10}$$

Enforce $k^\star \leq M$; otherwise the optimum saturates at $k = M$.

**Reintroducing the log.** With the routing term present, the derivative of the first-order condition gives

$$\frac{2\beta}{d} A^{-2\beta/d} k^{-2\beta/d-1} = \frac{1}{n}\Big(A + L_T \ell \log \frac{eM}{k} - L_T \ell\Big) \tag{11}$$

This matches the stated first-order condition whose closed form involves a Lambert-$W$ factor if solved exactly. Since $\log(eM/k)$ varies slowly near the optimizer, the solution is

$$k^\star \asymp \min\Big\{M, \Big(\frac{n}{A + L_T \ell\, \log(eM/k^\star)}\Big)^{\frac{d}{2\beta+d}}\Big\} \tag{12}$$

Thus the presence of $M$ affects $k^\star$ via a *logarithm only*; the rate in $n$ remains $n^{d/(2\beta+d)}$. The full proof of these calculations are provided in appendix G.1.

### 4.4 Routing-dominated vs. power-law regime

Routing dominates when

$$\frac{R_{\text{route}}}{n} \gtrsim n^{-\frac{2\beta}{2\beta+d}} \iff n \lesssim n_{\text{thresh}} := \Big(L_T \ell k \log \frac{eM}{k}\Big)^{\frac{2\beta+d}{d}}$$

$$\text{if } n \ll n_{\text{thresh}}: \quad \mathbb{E}\|\hat{T}_n - f\|^2_{L^2(Q)} \approx \frac{L_T \ell k \log(eM/k)}{n}; \quad \text{if } n \gg n_{\text{thresh}}: \quad \text{power law } n^{-\frac{2\beta}{2\beta+d}} \text{ prevails.}$$

Ways to reduce routing dominance include architectural and design adjustments: fewer MoE layers ($L_T$) or shorter sequence lengths ($\ell$) directly lower the multiplicative factor in the routing term, while choosing a smaller activation sparsity $k$ or setting $M$ close to $k$ shrinks the combinatorial overhead $\log(eM/k)$. These strategies align with empirical system-level practices in sparse Transformers: for instance, Switch Transformers (Fedus et al., 2022) and GShard (Lepikhin et al., 2021) emphasized top-1 or top-2 routing for efficiency, while later refinements such as BASE Layers (Lewis et al., 2021) and expert-choice routing (Zhou et al., 2022) reduced routing variance and overhead. Large-scale MoE deployments such as GLaM (Du et al., 2022) and V-MoE (Riquelme et al., 2021) also highlight the importance of balancing $M$ and $k$ to avoid routing bottlenecks. From the theoretical side, our bound formalizes these intuitions: the routing-dominated regime is suppressed precisely when the effective expert activation ($k$) and the combinatorial explosion in $M$ are controlled, allowing the system to operate in the favorable power-law regime where sample complexity scales as $n^{-2\beta/(2\beta+d)}$.

### 4.5 Sample complexity for target error $\varepsilon$

In the power-law regime, where routing overhead is negligible relative to the approximation and estimation terms, the classical nonparametric rate applies. Specifically, achieving a target population error $\varepsilon$ requires

$$n(\varepsilon) \asymp \varepsilon^{-\frac{2\beta+d}{\beta}}, \qquad N_{\text{eff}}(\varepsilon) \asymp \varepsilon^{-\frac{d}{\beta}} \tag{13}$$

The first relation quantifies the number of samples needed as a function of smoothness $\beta$ and intrinsic dimension $d$; the second gives the corresponding effective parameter budget scaling to match the approximation rate.

Both coincide with the dense-network theory (Yarotsky, 2017; Schmidt-Hieber, 2020), except that in MoE models $N_{\text{eff}}$ is interpreted as the *active* parameter budget per input rather than the total parameter count.

In contrast, in the routing-dominated regime the error floor is determined by the combinatorial overhead of selecting experts. In this case, sample complexity to achieve accuracy $\varepsilon$ grows only linearly in $1/\varepsilon$:

$$n(\varepsilon) \ \asymp\ \frac{L_T \ell k \log(eM/k)}{\varepsilon} \tag{14}$$

This expression highlights that the overhead scales with the number of MoE layers ($L_T$), sequence length ($\ell$), and active experts per layer ($k$), while depending only logarithmically on the total number of experts $M$. Thus, if $M \gg k$, routing dominates and inflates sample complexity significantly, while $M = \Theta(k)$ mitigates this effect. In practice, this explains why architectures such as Switch Transformers (Fedus et al., 2022) or GLaM (Du et al., 2022) adopt small $k$ (top-1 or top-2 routing) and balance $M$ against $k$ to remain in the favorable power-law regime. Our theoretical analysis therefore provides a principled characterization of when MoE gains translate into efficient sample usage and when routing combinatorics instead dictate learning dynamics.

### 4.6 COMPUTE-OPTIMAL TRADEOFFS

We derive the optimal allocation of *active parameters per input $N_{\text{eff}}$* and *number of training samples $n$* under a fixed training compute budget $C$. Throughout we use the bound in equation 3 and we adopt the standard compute model in which per-token (or per-sample) FLOPs scale linearly with the active parameter budget (Hoffmann et al., 2022; Kaplan et al., 2020), so the total training compute satisfies

$$C \ \asymp\ n \cdot N_{\text{eff}}. \tag{15}$$

**Proposition 4.5** (Compute-optimal allocation of $N_{\text{eff}}$ and $n$)**.** *Assume the compute budget equation 15. Then the excess error bound is minimized (up to constants) by*

$$N_{\text{eff}}^\star(C) \ \asymp\ C^{\frac{d}{2\beta+2d}} \qquad and \qquad n^\star(C) \ =\ \frac{C}{N_{\text{eff}}^\star(C)} \ \asymp\ C^{\frac{2\beta+d}{2\beta+2d}} \tag{16}$$

*The resulting compute error scaling law is*

$$\mathbb{E}\,\|\hat{T}_{n^\star} - f\|_{L^2(Q)}^2 \ \asymp\ C^{-\frac{\beta}{\beta+d}} \tag{17}$$

Thus, under a fixed compute budget, both the dataset size and the per-sample active parameters should grow with $C$ according to the exponents in equation 16. The error decays as a power law in compute with exponent $\beta/(\beta+d)$, matching nonparametric theory and providing the MoE analogue of compute scaling laws. See appendix G.3 for proof.

## 5 PREDICTING EMPIRICAL SCALING LAWS AND VALIDATION ON LLMS

Our empirical study is designed to test the scaling exponents and routing effects predicted by our theory under controlled but realistic conditions. We validate our theoretical predictions on a family of causal decoder-only Transformers with Mixture-of-Experts (MoE) feed-forward layers. For these experiments we use three public text corpora of varying complexity and intrinsic dimension: **TinyStories** (low-dimensional synthetic children's tales) (Eldan & Li, 2023), **WikiText-103** (medium-scale, moderately heterogeneous Wikipedia articles)(Merity et al., 2016) and **OpenWebText** (large, highly heterogeneous corpus recreated from Reddit links) (Gokaslan & Cohen, 2019). See Appendix A for full description.

### 5.1 EMPIRICAL VALIDATION ON LLMS

**Intrinsic dimension $d$ estimates in practice.** We estimate the intrinsic dimension $d$ of the data manifold from hidden representations using standard intrinsic-dimension estimators. Concretely, we pass corpus tokens through a fixed pretrained GPT-2 model, collect hidden states at a given layer, and apply the Levina–Bickel $k$-NN MLE, which is more robust than TwoNN in high dimensions (Fukunaga & Olsen, 1971; Levina & Bickel, 2004; Facco et al., 2017). $k$-nearest-neighbor distances are computed with FAISS, and the ID is evaluated layer-wise. To reduce variance, we repeat the procedure over multiple random subsamples and report the

**median** ID (with median absolute deviation as uncertainty). Unless stated otherwise, we use the **middle Transformer layers** as the global summary of $d$. This procedure yields stable and reproducible ID estimates that are consistent with theoretical predictions from neural scaling laws, allowing us to define a characteristic ID ($d$) for each corpus that captures its inherent complexity, as summarized in table 1 and in the depth curves (Figure 3 and 4 in appendix A ).

**Consistency of $\beta$ across scaling regimes.**  Our theoretical framework introduces a smoothness exponent $\beta$ that governs both model scaling, through $\alpha_N = 2\beta/d$, and data scaling, through $\alpha_D = 2\beta/(2\beta + d)$. When estimating $\beta$ from the empirically fitted slopes $\widehat{\alpha}_N$ and $\widehat{\alpha}_D$, we observe that the resulting values $\beta_{\mathrm{model}}$ and $\beta_{\mathrm{data}}$ need not coincide exactly (see Figure 1 and 2 ). This discrepancy is expected: model-scaling curves are typically cleaner and dominated by approximation error, whereas data-scaling curves are more sensitive to optimization noise, finite-sample effects, and in the MoE setting routing overhead terms of the form $k \log(eM/k)$. Similar inconsistencies between model- and data-derived exponents have been reported in empirical studies of dense and MoE language models (Kaplan et al., 2020; Hoffmann et al., 2022; Krajewski et al., 2024; Ludziejewski et al., 2025). The fact that the two estimates of $\beta$ remain broadly consistent nevertheless supports the qualitative validity of our theoretical exponents.

Table 1: **Intrinsic Dimension (ID) Summary across Models and Datasets.** The ID estimates remain remarkably stable across different model sizes for each dataset. For the subsequent scaling analysis (Figure 2 and 1), we chose the following median ID values for each dataset: $d = 32$ for WikiText-103, $d = 45$ for OpenWebText, and $d = 23$ for TinyStories.

| Model (number of parameters) | Tinystories | | Wikitext-103 | | OpenWebText | |
|---|---|---|---|---|---|---|
| | Mean (CI) | Median | Mean (CI) | Median | Mean (CI) | Median |
| gpt2 (117 million) | $22.9 \pm 2.9$ | 23.1 | $31.6 \pm 2.9$ | 32.4 | $46.7 \pm 6.2$ | 49.8 |
| gpt2-medium (345 million) | $21.4 \pm 1.1$ | 21.9 | $31.6 \pm 1.7$ | 32.9 | $43.7 \pm 2.6$ | 45.0 |
| gpt2-large (774 million) | $22.1 \pm 0.9$ | 22.4 | $31.5 \pm 1.6$ | 32.1 | $44.7 \pm 2.6$ | 47.7 |
| gpt2-xl (1558 million) | $19.7 \pm 0.6$ | 19.6 | $30.3 \pm 1.0$ | 31.1 | $43.0 \pm 1.7$ | 43.9 |
| **Values Adopted for Scaling Analysis** | | | | | | |
| **Chosen $d$ (Median ID)** | $d = 23$ | | $d = 32$ | | $d = 45$ | |
| **Best-Fit $\beta$** | $\beta \approx 1.0$ | | $\beta \approx 1.0$ | | $\beta \approx 1.0 - 1.5$ | |

**Experimental setup.**  We operate in the small/medium regime to allow dense sweeps of the capacity grid. Unless noted otherwise, we use sequence length $\ell \in \{128, 256\}$, $m = 4$ attention heads, AdamW, and learning rate $3 \times 10^{-4}$. Model scaling varies $(L_T, d_{\mathrm{ff}})$ over a broad grid and explores several MoE configurations $(M, k)$. Data scaling uses a representative architecture and varies the token budget $D$ over $5 \times 10^4$–$8 \times 10^5$.

We evaluate three regimes mirroring the theory: (i) *model scaling*, where we estimate the exponent $\widehat{\alpha}_N$ by regressing $\log L$ on $\log N_{\mathrm{eff}}$; (ii) *data scaling*, where we regress $\log L$ on $\log D$ to extract $\widehat{\alpha}_D$; and (iii) *routing ablations*, where we vary $(M, k)$ to isolate the routing combinatorial term $L_T \ell k \log(eM/k)$ and identify the crossover from routing-dominated behavior to the power-law regime.

### 5.2   Fitting Exponents and Comparing to Theory

In all scaling experiments, we worked in the regime where routing overhead was either small relative to the validation loss or explicitly accounted for. When necessary, we fit a small loss floor $c$ such that we performed regression of $\log(L - c)$ on the relevant variable ($\log N_{\mathrm{act}}$ for model scaling or $\log D$ for data scaling). From these fits, we extracted the empirical exponents $(\widehat{\alpha}_N, \widehat{\alpha}_D)$. We compared these empirical exponents to the theoretical predictions $(\alpha_N, \alpha_D)$ derived from the Intrinsic Dimension (ID) theory. Specifically, the theoretical exponents are calculated using the chosen intrinsic dimension $d$ and the scaling exponent $\beta$. The best alignment of the empirical fits with the theoretical lines (as shown in the sensitivity plots of Figure 2 and 1) was used to determine the scaling exponent $\beta$. The theoretical scaling predictions, summarized in Table 2, confirm two key findings: first, the empirically fitted exponents $(\widehat{\alpha}_N, \widehat{\alpha}_D)$ align closely with the theoretical predictions $(\alpha_N, \alpha_D)$ derived from the Intrinsic Dimension framework using a consistent scaling exponent $\beta \approx 1.0 - 1.5$; and second, the exponents satisfy the crucial internal consistency identity for the compute-optimal frontier, $\alpha_D \approx \frac{\alpha_N}{1 + \alpha_N}$, demonstrating that the theoretical $\alpha_D$ values are in close proximity to the calculated $\alpha_N/(1 + \alpha_N)$ values, thereby supporting the unified theoretical relationship between model and data scaling.

Table 2: Theoretical Exponents derived from the Chosen $d$ and $\beta$.

| Dataset | Chosen $d$ | Best-Fit $\beta$ | Theoretical Exponents | | | Empirical Exponents | |
|---|---|---|---|---|---|---|---|
| | | | $\alpha_N = \frac{2\beta}{d}$ | $\alpha_D = \frac{2\beta}{2\beta+d}$ | Consistency $\frac{\alpha_N}{1+\alpha_N}$ | $\widehat{\alpha}_N$ | $\widehat{\alpha}_D$ |
| **WikiText-103** | 32 | 1.0 | 0.063 | 0.059 | 0.059 | **0.060** | **0.058** |
| **OpenWebText** | 45 | 1.0 | 0.044 | 0.043 | 0.042 | **0.045** | **0.043** |
| **OpenWebText** | 45 | 1.5 | 0.067 | 0.062 | 0.063 | **0.068** | **0.062** |
| **TinyStories** | 23 | 1.0 | 0.087 | 0.080 | 0.080 | **0.085** | **0.081** |

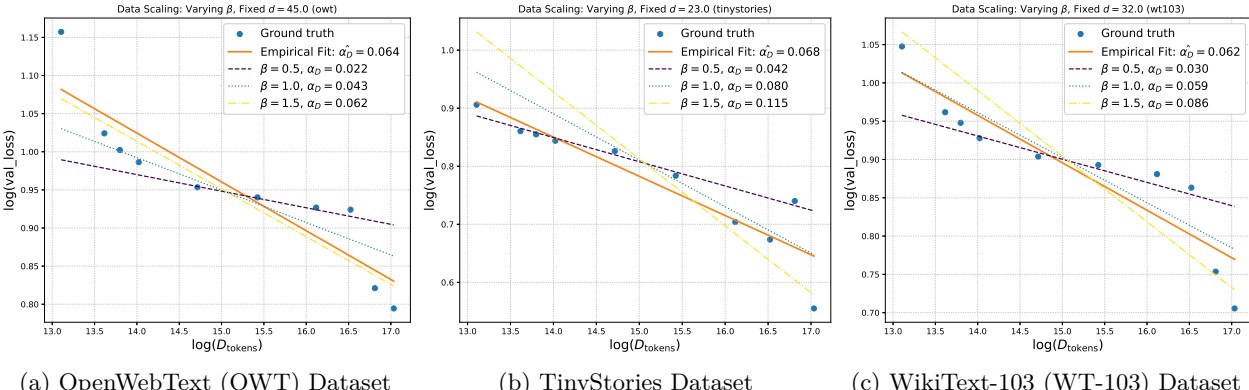

(a) OpenWebText (OWT) Dataset    (b) TinyStories Dataset    (c) WikiText-103 (WT-103) Dataset

Figure 1: **Data Scaling Results (Fixed $d$, Varying $\beta$).** Empirical validation of the Data Scaling Law (loss vs. data size, $D$) across different datasets, where the slope (power law exponent $\alpha_D$) is compared against theoretical lines derived by fixing the dataset-specific intrinsic dimension $d$ and varying the exponent $\beta$. The subfigures show: (a) OpenWebText (OWT) dataset, with the empirical fit aligning best with $\beta = 1.5$. (b) TinyStories dataset, aligning best with $\beta = 1.0$. (c) WikiText-103 (WT-103) dataset, aligning best with $\beta = 1.0$.

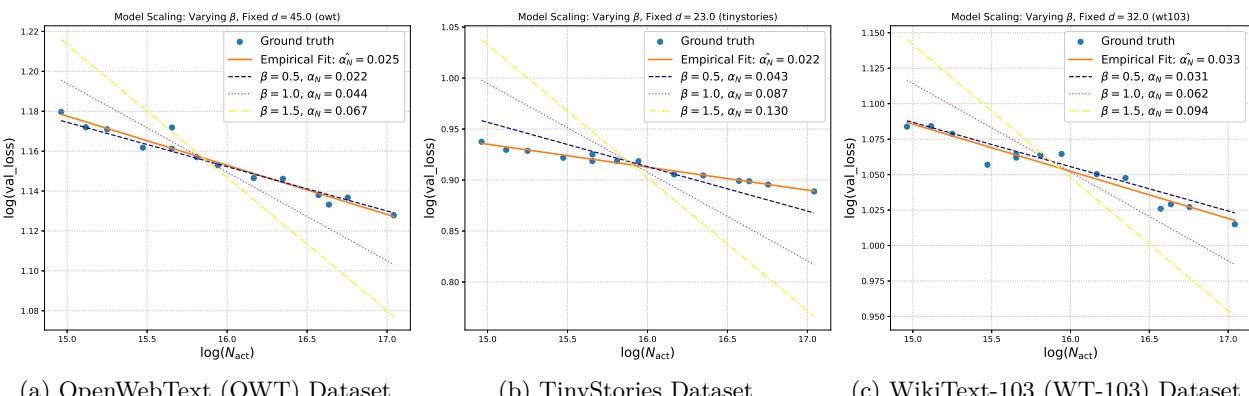

(a) OpenWebText (OWT) Dataset    (b) TinyStories Dataset    (c) WikiText-103 (WT-103) Dataset

Figure 2: **Model Scaling Results (Fixed $d$, Varying $\beta$).** Empirical validation of the Model Scaling Law (loss vs. active parameter count, $N_{\text{act}}$) for different datasets. For each plot, the dataset's intrinsic dimension $d$ is fixed (e.g., $d = 45$ for OWT) and theoretical lines are plotted for varying $\beta$. The theoretical scaling exponent is $\alpha_N = 2\beta/d$. The empirical fits align best with the following theoretical exponents: (a) OpenWebText: best alignment with $\beta = 0.5$. (b) TinyStories: best alignment with $\beta = 0.5$. (c) WikiText-103: best alignment with $\beta = 0.5$.

## 6 Discussion and Limitations

Our results suggest a simple mental model for MoE design: treat *active per-token capacity* as the main driver of accuracy, and view *routing* as an overhead to keep modest. In practice, this means budgeting first for the parameters actually used on each token, and only then adjusting the expert pool and gating so that routing remains stable and inexpensive. When this balance holds, we observe dense-style scaling behavior measured

against active capacity rather than total parameters. Our experiments on TinyStories, WikiText-103, and OpenWebText support this view: the fitted model- and data-scaling exponents are well explained by the intrinsic-dimension theory, and the observed routing effects match the predicted logarithmic dependence on the expert pool size. At the same time, the constants in our bounds are conservative, and the empirical regimes we probe are still far from the asymptotic limit. Three design levers matter most: (i) activation sparsity (how many experts fire per token), (ii) the size and granularity of experts, and (iii) the amount of routing diversity allowed before it starts to dominate training. Concretely, avoid expert pools that dwarf the number of activated experts; monitor route entropy and token–expert load balance during training; and, for long contexts, reduce activation to keep routing stable.

**Threats to validity.** These guarantees rest on uniform assumptions: data on a compact $C^1$ $d$-manifold, $f \in C^\beta(\mathcal{M})$, bounded outputs, pre-norm blocks with ReLU experts, and hard $k$-sparse routing analyzed via fixed routing patterns and a union bound. Such covering-number arguments are worst-case and may be loose compared with data-dependent complexities; they also abstract away optimization (SGD dynamics, label noise) and potential *specialization gains* from large $M$ that can improve constants without changing exponents. Soft gating, load-balancing losses, and pathwise route diversity primarily shift constants and the routing term. Sharpening the theory likely requires (i) routing-aware localized complexities (e.g., entropy of *observed* routes), (ii) smoother gate stability bounds with temperature/regularization, and (iii) compute-aware analyses that include step count, batch size, memory, and communication costs.

**Scope limitations.** We use theoretical FLOPs as a platform-agnostic proxy for cost. This omits memory bandwidth, communication, kernel fusion, parallelism, and systems effects that often dominate wall-clock. Our bounds target regression-style squared loss; cross-entropy, instruction tuning, and RLHF may show different constants or crossovers. Inference-time considerations (latency, cache locality, quantization) are also outside scope, though they can reshape the practical Pareto frontier.

**When the simple rules can break.** Routing can dominate in small data regimes, with very long sequences, or when the expert pool is much larger than the number of activated experts. Soft/temperature-controlled gating, auxiliary load-balancing losses, and non-separable interactions between data size and active capacity may also shift constants and move the crossover points. In all these cases, monitoring route entropy, per-expert utilization, and stability metrics is essential.

## 7 CONCLUSION

We developed a theory of MoE generalization and scaling that cleanly separates *active* per-token capacity from routing combinatorics. On the capacity side, a covering-number analysis yields a constructive approximation bound and a uniform generalization bound whose excess risk decomposes into $N_{\text{eff}}^{-2\beta/d}$ (approximation), $N_{\text{active}}/n$ (estimation), and $R_{\text{route}}/n$ (routing). From these we derive neural scaling laws that mirror dense-model exponents but are measured against *active* parameters. The theory predicts $k^\star \asymp n^{d/(2\beta+d)}$ and shows that enlarging $M$ at fixed $k$ changes rates only logarithmically via routing, motivating designs with $M = \Theta(k)$ and modest routing on long sequences. We complement this analysis with empirical validation on three MoE language models trained on TinyStories, WikiText-103, and OpenWebText: we estimate the intrinsic dimension $d$ and smoothness $\beta$ and show that the fitted model- and data-scaling exponents closely follow the predicted power laws. Empirically, trends reported by Joint MoE Scaling Laws (JMSL) and recent EL-based studies align with these prescriptions: (i) the ratio of data-to-model exponents grows with expertization, indicating amplified data efficiency at fixed active capacity; (ii) compute-optimal allocations shift toward more tokens per active parameter as $E$ increases; and (iii) efficiency gains accrue primarily from scaling *active* capacity while keeping routing manageable. Practically, this translates into simple rules of thumb: scale data and active parameters jointly, keep $\log(eM/k) = \Theta(1)$, operate near $k^\star$, and mitigate routing costs on long sequences (e.g., via gate sharing or smaller $k$).

**Future work and Outlook.** Promising extensions include replacing crude routing complexity terms (e.g., $\log|\Pi|$) with data-dependent measures based on the entropy or compressibility of *observed* routes and modeling specialization gains to explain how enlarging $M$ at fixed $k$ improves approximation *constants* via expert diversityOverall, the path forward is sharper, data-dependent routing analytics and optimization-aware compute models, turning our high-level message scale *active* per-sample capacity while keeping routing overhead modest into predictive, design-ready scaling rules for sparse Transformers.

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

## A    EMPIRICAL VALIDATION ON MoE TRANSFORMER LLMs

This section provides a detailed description of the architectures, hyperparameter grids, and scaling protocols used to empirically validate the theoretical predictions. Our goal is to cleanly isolate the roles of (i) effective model capacity, (ii) dataset size, and (iii) routing combinatorics in MoE Transformers.

### A.1    DATASETS.

We consider three publicly available text corpora with varying complexity and intrinsic dimension: **Stories / TinyStories** is a corpus of short, synthetically generated children's stories, following (Eldan & Li, 2023). This dataset is known to lie on an intrinsically low-dimensional manifold, and has been used in prior work on theoretical scaling laws for Transformers. **WikiText-103** is a curated subset of English Wikipedia articles (Merity et al., 2016), commonly used for language modeling. It provides a medium-scale, moderately heterogeneous domain. **OpenWebText** is an open recreation of the WebText corpus (Gokaslan & Cohen, 2019), constructed from outgoing Reddit links. It is substantially larger and more heterogeneous than WikiText-103, with higher intrinsic dimension. For each corpus we extract a plain-text training stream (one document per line). For scaling experiments we sub-sample the stream to match the desired token budgets.

### A.2    INTRINSIC DIMENSION ESTIMATION (EXTENDED DESCRIPTION)

The intrinsic dimension (ID) of a representation manifold quantifies the effective degrees of freedom of the learned features. To estimate the ID of transformer representations, we employ the Levina–Bickel Maximum Likelihood Estimator (MLE) (Levina & Bickel, 2004), which models the local likelihood of $k$-nearest-neighbor distances on a smooth manifold. Compared to the TwoNN estimator, which relies only on the ratio of the first two nearest neighbors, the MLE approach uses the full local neighborhood and exhibits substantially lower variance in high-dimensional settings an essential property when studying large language models.

We extract hidden states from each block of a pretrained GPT-2 model (gpt2-medium) by sampling tokens from a large text corpus. For each layer $\ell$, we compute all pairwise distances to its $k$ nearest neighbors using FAISS in 32-bit precision, and apply the MLE estimator to the resulting distance matrix $D_\ell$. The estimation pipeline is repeated for multiple random subsamples to mitigate sampling noise; we report the **median MLE estimate** for each layer, along with the **median absolute deviation (MAD)** as a robust measure of uncertainty.

Following established practice in intrinsic-dimension analyses of deep networks, we summarize the model-level intrinsic dimension using the **middle layers** 40–60% depth). Shallow layers typically show inflated dimensionality due to local embedding variation, whereas deeper layers tend to collapse. This procedure yields stable and reproducible ID estimates that are consistent with theoretical predictions from neural scaling laws, allowing us to define a characteristic ID ($d$) for each corpus that captures its inherent complexity, as summarized in the depth curves (Figure 3) and the scaling law fits (Figure 4).

### A.3    MODEL FAMILY: MoE DECODER-ONLY TRANSFORMERS

All experiments use a causal decoder-only Transformer with Mixture-of-Experts (MoE) feed-forward layers. Each model consists of token and positional embeddings, followed by $L_T$ Transformer blocks. Each block contains:

- **Self-attention:** multi-head attention with $m$ heads, model dimension $d_{\text{model}}$, causal masking, and standard projection matrices.
- **MoE feed-forward network:** a routing network selecting the top-$k$ experts (hard top-$k$ gating). Each expert is an MLP with hidden width $d_{\text{ff}}$ and a GELU nonlinearity. Outputs of the selected experts are combined via a softmax over routing logits.

The *active* parameter budget per input,

$$N_{\text{eff}} = L_T\big(\Pi_{\text{attn}} + k\,\Pi_{\text{exp}}\big),$$

counts only the parameters that contribute to computation for a given token. This is the quantity that appears in our approximation and covering-number bounds, and serves as the scaling variable in model-capacity experiments.

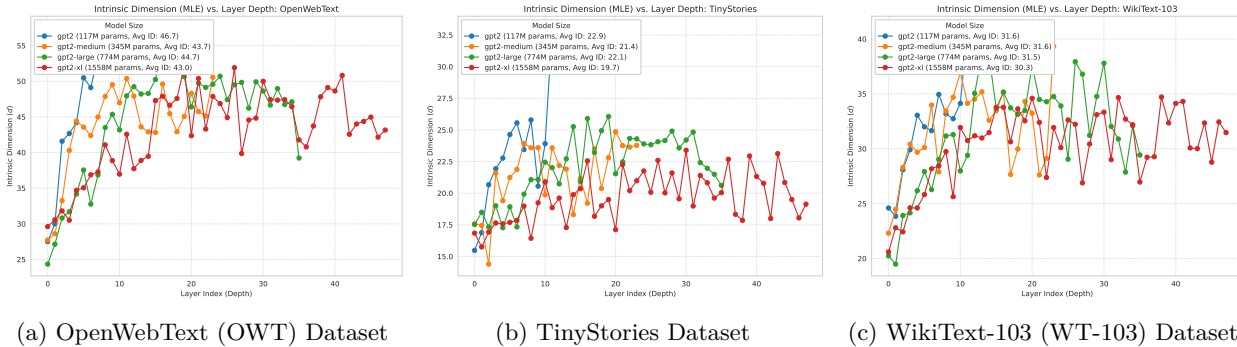

(a) OpenWebText (OWT) Dataset     (b) TinyStories Dataset     (c) WikiText-103 (WT-103) Dataset

Figure 3: **Intrinsic Dimension (ID) Evolution Across Model Depth.** The plots show the Maximum Likelihood Estimation (MLE) of the Intrinsic Dimension, $d$, calculated for the representations at each layer of the four GPT-2 models (varying size). The ID remains stable across different model sizes for a given dataset but varies significantly across data corpora. (a) **OpenWebText (OWT):** Shows the highest intrinsic dimension, reflecting its diverse and complex nature. (b) **TinyStories:** Exhibits the lowest intrinsic dimension, consistent with its synthetic, simple structure. (c) **WikiText-103 (WT-103):** Has an intermediate intrinsic dimension.

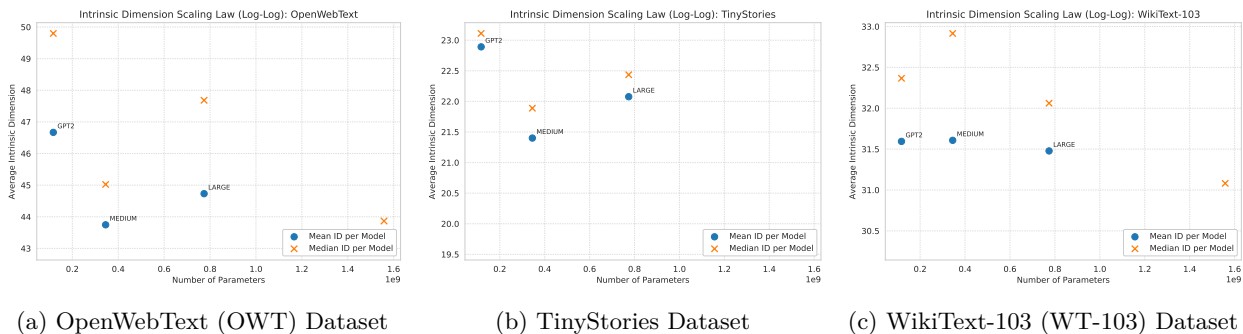

(a) OpenWebText (OWT) Dataset     (b) TinyStories Dataset     (c) WikiText-103 (WT-103) Dataset

Figure 4: **Intrinsic Dimension Scaling Law: $\bar{d}$ versus Number of Parameters ($N$).** The plots show the relationship between the **Average Intrinsic Dimension ($\bar{d}$)** (log-log scale) and the **Number of Parameters ($N$)** across the four GPT-2 model sizes for each dataset. This relationship is modeled as a power law, $\bar{d} \sim N^{\beta}$, where the slope of the linear fit determines the **ID Scaling Exponent** $\beta$. The results demonstrate that $\bar{d}$ exhibits weak or negligible scaling with model size across all three corpora.

### A.4 HYPERPARAMETERS AND CAPACITY GRIDS

We operate in the small-to-medium compute regime to allow dense parameter sweeps and low-noise log–log regressions. Unless otherwise stated, we use:

- sequence length $\ell \in \{128, 256\}$,
- $m = 4$ attention heads,
- AdamW with learning rate $3 \times 10^{-4}$ and weight decay 0.1.

**Model-scaling grid.** To vary the active capacity over 1–2 orders of magnitude, we sweep
$$(L_T, d_{\text{ff}}) \in \{2, 3, 4, 5, 6, 8\} \times \{256, 384, 512, 640, 768, 896, 1024, 1280, 1536\},$$
combined with MoE configurations $M \in \{4, 8, 16\}$ and $k \in \{1, 2\}$. We avoid extremely large $M$ or $k$ to remain within GPU memory while still covering a wide range of effective capacities.

### A.5 SCALING PROTOCOLS

We run three classes of experiments that map directly onto the theoretical decomposition of the generalization bound.

**(1) Model scaling.** For a fixed dataset, we train all models in the sweeping grid under a common token budget $D_{\text{fixed}}$. After convergence, we record the validation loss $L$ and fit a linear regression of $\log L$ versus $\log N_{\text{eff}}$ over the region where routing is negligible. The slope yields the empirical model exponent $\widehat{\alpha}_N$.

**(2) Data scaling.** We choose a representative "base" architecture (e.g., $L_T = 4$, $d_{\text{model}} = 128$, $d_{\text{ff}} = 512$, $M = 8$, $k = 2$) and vary the number of training tokens

$$D \in \{5 \times 10^4,\, 10^5,\, 2 \times 10^5,\, 4 \times 10^5,\, 8 \times 10^5\}.$$

For each value of $D$ we adjust the number of optimization steps to keep the total token budget fixed. A regression of $\log L$ on $\log D$ yields the empirical data exponent $\widehat{\alpha}_D$.

**(3) Routing ablations.** To isolate the influence of routing combinatorics, we fix both the base architecture and the token budget, and vary the expert configuration according to

$$k \in \{1, 2\}, \qquad M \in \{2k, 4k, 8k\}.$$

For each $(M, k)$ pair we train a model, record its validation loss, and compute the theoretical routing term $L_T \ell k \log(eM/k)$. Plotting loss against this term reveals the routing-dominated regime and the predicted crossover into the power-law region governed by $(N_{\text{eff}}, D)$. The models considered in this part are listed in Table 3.

| $L_T$ | $d_{ff}$ | $N_{\text{act}}$ | $d_{\text{model}}$ | $n_{\text{heads}}$ | $M$ | $k$ |
|---|---|---|---|---|---|---|
| 2 | 256 | 3,148,800 | 512 | 4 | 8 | 2 |
| 2 | 384 | 3,673,600 | 512 | 4 | 8 | 2 |
| 2 | 512 | 4,198,400 | 512 | 4 | 8 | 2 |
| 2 | 768 | 5,248,000 | 512 | 4 | 8 | 2 |
| 2 | 1024 | 6,297,600 | 512 | 4 | 8 | 2 |
| 4 | 256 | 6,297,600 | 512 | 4 | 8 | 2 |
| 4 | 384 | 7,347,200 | 512 | 4 | 8 | 2 |
| 4 | 512 | 8,396,800 | 512 | 4 | 8 | 2 |
| 4 | 768 | 10,496,000 | 512 | 4 | 8 | 2 |
| 4 | 1024 | 12,595,200 | 512 | 4 | 8 | 2 |
| 4 | 1536 | 16,793,600 | 512 | 4 | 8 | 2 |
| 6 | 768 | 15,744,000 | 512 | 4 | 8 | 2 |
| 6 | 1024 | 18,892,800 | 512 | 4 | 8 | 2 |
| 8 | 1024 | 25,190,400 | 512 | 4 | 8 | 2 |

Table 3: **Model configurations used in the scaling experiments for each data set**. Each row corresponds to a distinct Transformer-MoE model used in the model-scaling sweep. The effective active parameter count $N_{\text{act}}$ varies with the number of layers and FFN width, while $d_{\text{model}} = 512$, $n_{\text{heads}} = 4$, $M = 8$, $k = 2$, and sequence length $\ell = 256$ are held fixed.

## B   EXTENDED RESULTS: EMPIRICAL COMPARISON TO PRIOR MoE SCALING LAWS

We align our theoretical exponents model scaling with the empirical laws reported in EL (Tian et al., 2025) and *Joint MoE Scaling Laws* (JMSL) (Ludziejewski et al., 2025). In the power-law (routing-negligible) regime, our theory implies two *parameter-free* identities that can be checked directly:

$$\alpha_D^{theory} \;=\; \frac{\alpha_N^{theory}}{1 + \alpha_N^{theory}}, \qquad \alpha_C^{theory} \;=\; \frac{\alpha_N^{theory}}{2 + \alpha_N^{theory}} \tag{18}$$

Accordingly, we test our theory by comparing empirical exponents to these identities, *without* estimating the intrinsic dimensionality.

We align our theoretical exponents model scaling $\alpha_N$, data scaling $\alpha_D$, and compute scaling $\alpha_C$ with the empirical laws reported in *Joint MoE Scaling Laws* (JMSL) (Ludziejewski et al., 2025). They supposed that, for a fixed number of experts $E$, the validation loss follows a Chinchilla-style surface

$$L(N_{\text{act}}, D \,|\, E) \;=\; m(E)\, N_{\text{act}}^{\mu(E)} \;+\; n(E)\, D^{\nu(E)} \;+\; c, \tag{19}$$

and then introduces an *expert-dependent* joint law by making both the prefactors and exponents functions of $E$ (ultimately via a smoothed $\hat{E}$). Concretely,

$$L(N_{\text{act}}, D, \hat{E}) \;=\; a\,\hat{E}^{\delta}\,N_{\text{act}}^{\alpha+\gamma\ln\hat{E}} \;+\; b\,\hat{E}^{\omega}\,D^{\beta+\zeta\ln\hat{E}} \;+\; c, \tag{20}$$

which reduces to equation 19 when $E$ is held fixed, with $\mu(E) = \alpha + \gamma\ln\hat{E}$ and $\nu(E) = \beta + \zeta\ln\hat{E}$. (Ludziejewski et al., 2025) provides fitted coefficients, per-$E$ reduced exponents, and a compute-optimal analysis under the training-FLOPs proxy $F = 6\,N_{\text{act}}\,D$. We adopt Their counting conventions for *active* parameters and FLOPs when fitting/reading off exponents. From their per-expert reduction (their Eq. (5) and Table 4), we read exponents $\mu(E)$ and $\nu(E)$ and map

$$\hat{\alpha}_N(E) := -\,\mu(E), \qquad \hat{\alpha}_D(E) := -\,\nu(E).$$

JMSL reports that as $E$ increases the magnitude of $\nu(E)$ *increases* (improved data efficiency) while $|\mu(E)|$ slightly *decreases*. The resulting values and the induced compute exponent are summarized in Table 4.

**Compute-optimal frontier and induced $\alpha_C$.**   On compute-optimal slices with the constraint $F = 6\,N_{\text{act}}D$, the separable form equation 19 implies closed-form optima (JMSL Eq. (9)):

$$N_{\text{act}}^{\text{opt}}(F, E) = G(E)\left(\frac{F}{6}\right)^{\frac{\nu(E)}{\mu(E)+\nu(E)}}, \qquad D^{\text{opt}}(F, E) = G(E)^{-1}\left(\frac{F}{6}\right)^{\frac{\mu(E)}{\mu(E)+\nu(E)}},$$

for a known $G(E)$ depending on $m(E), n(E), \mu(E), \nu(E)$. Along this frontier, the *compute* scaling exponent is

$$\hat{\alpha}_C(E) \;=\; \frac{\hat{\alpha}_N(E)\,\hat{\alpha}_D(E)}{\hat{\alpha}_N(E) + \hat{\alpha}_D(E)}. \tag{21}$$

We use JMSL's $F$-proxy and the per-$E$ $(\mu, \nu)$ to report $\hat{\alpha}_C(E)$ and to compare the token-to-parameter ratio drift with $E$ (JMSL's Finding 1).

### B.1   Our theory ⇒ empirical mapping and consistency checks

**Theoretical exponents.**   In the power-law (routing-negligible) regime, our theory predicts

$$\alpha_N^{theory} = \frac{2\beta}{d}, \qquad \alpha_D^{theory} = \frac{2\beta}{2\beta + d}, \qquad \alpha_C^{theory} = \frac{\beta}{\beta + d}.$$

This implies two *parameter-free* identities that can be checked directly against JMSL:

$$\alpha_D^{theory} \;=\; \frac{\alpha_N^{theory}}{1 + \alpha_N^{theory}}, \qquad \alpha_C^{theory} \;=\; \frac{\alpha_N^{theory}}{2 + \alpha_N^{theory}} \;=\; \frac{\alpha_N^{theory}\,\alpha_D^{theory}}{\alpha_N^{theory} + \alpha_D^{theory}} \tag{22}$$

Thus, for *each* expert count $E$, a simple test is whether the empirical pair $\big(\hat{\alpha}_N(E), \hat{\alpha}_D(E)\big)$ approximately satisfies $\hat{\alpha}_D \approx \hat{\alpha}_N/(1 + \hat{\alpha}_N)$, and whether $\hat{\alpha}_C(E)$ computed from equation 21 aligns with the right-hand identity in equation 18. Deviations quantify where practice departs from the smooth-manifold asymptotics (finite-sample regime, tokenization/entropy floors $c$, training schedules, or mild routing side-effects).

### B.2   Per-$E$ identity checks

For each expert count $E$, we compare the empirical pair $\big(\hat{\alpha}_N(E), \hat{\alpha}_D(E)\big)$ against the parameter-free identities

$$\alpha_D^{\text{pred}}(E) = \frac{\hat{\alpha}_N(E)}{1 + \hat{\alpha}_N(E)}, \qquad \alpha_C^{\text{pred}}(E) = \frac{\hat{\alpha}_N(E)}{2 + \hat{\alpha}_N(E)} = \frac{\hat{\alpha}_N(E)\,\alpha_D^{\text{pred}}(E)}{\hat{\alpha}_N(E) + \alpha_D^{\text{pred}}(E)}.$$

Defining residuals

$$r_D(E) = \hat{\alpha}_D(E) - \alpha_D^{\text{pred}}(E), \qquad r_C(E) = \hat{\alpha}_C(E) - \alpha_C^{\text{pred}}(E),$$

we observe systematic, monotone-positive deviations across (see Table 4). Quantitatively, the mean absolute error is $\text{MAE}[r_D] = 0.083$ and $\text{MAE}[r_C] = 0.0186$, with $\text{MAPE}[r_D] = 58.7\%$ and $\text{MAPE}[r_C] = 24.2\%$. Both gaps grow with $E$: $(r_D, r_C) \approx (0.043, 0.011)$ at $E{=}1$ $\rightarrow$ $(0.130, 0.026)$ at $E{=}32$. Equivalently, Figure 5a show that the ratio $\hat{\alpha}_D/\hat{\alpha}_N$ increases with $E$ (from $\sim 1.08$ at $E{=}1$ to $\sim 1.70$ at $E{=}32$), indicating stronger-than-idealized data-side gains.

A parsimonious correction that captures the trend is an $E$-dependent amplification of the identity,

$$\hat{\alpha}_D(E) \;\approx\; \kappa(E)\,\frac{\hat{\alpha}_N(E)}{1 + \hat{\alpha}_N(E)}, \qquad \kappa(E) > 1,$$

with empirical multipliers $\kappa(1){\approx}1.28$, $\kappa(2){\approx}1.37$, $\kappa(4){\approx}1.47$, $\kappa(8){\approx}1.62$, $\kappa(16){\approx}1.79$, $\kappa(32){\approx}1.97$. A log-linear form $\kappa(E) = A + B\ln E$ provides a simple, testable fit (see 5b).

| $E$ | $\mu(E)$ | $\nu(E)$ | $\alpha_N$ | $\alpha_D$ | $\alpha_C$ | $\alpha_D^{\text{pred}}$ | $\alpha_C^{\text{pred}}$ | $r_D$ | $r_C$ | rel_err$_D$ | rel_err$_C$ |
|---|---|---|---|---|---|---|---|---|---|---|---|
| 1 | -0.1817 | -0.1965 | 0.1817 | 0.1965 | 0.0944 | 0.1537 | 0.0832 | 0.0427 | 0.0111 | 0.2779 | 0.1335 |
| 2 | -0.1780 | -0.2065 | 0.1780 | 0.2065 | 0.0955 | 0.1511 | 0.0817 | 0.0553 | 0.0138 | 0.3666 | 0.1697 |
| 4 | -0.1731 | -0.2192 | 0.1731 | 0.2192 | 0.0967 | 0.1475 | 0.0796 | 0.0716 | 0.0170 | 0.4855 | 0.2142 |
| 8 | -0.1676 | -0.2338 | 0.1676 | 0.2338 | 0.0976 | 0.1435 | 0.0773 | 0.0902 | 0.0203 | 0.6287 | 0.2625 |
| 16 | -0.1617 | -0.2494 | 0.1617 | 0.2494 | 0.0980 | 0.1391 | 0.0748 | 0.1102 | 0.0232 | 0.7917 | 0.3114 |
| 32 | -0.1557 | -0.2652 | 0.1557 | 0.2652 | 0.0981 | 0.1347 | 0.0722 | 0.1304 | 0.0258 | 0.9684 | 0.3582 |

Table 4: Per-$E$ identity checks. We set $\alpha_N(E) = -\mu(E)$ and $\alpha_D(E) = -\nu(E)$. The compute-optimal exponent is $\alpha_C(E) = \frac{\alpha_N(E)\alpha_D(E)}{\alpha_N(E)+\alpha_D(E)}$. Predicted exponents use $\alpha_D^{\text{pred}} = \alpha_N/(1+\alpha_N)$ and $\alpha_C^{\text{pred}} = \alpha_N/(2+\alpha_N)$. Residuals are $r_D = \alpha_D - \alpha_D^{\text{pred}}$ and $r_C = \alpha_C - \alpha_C^{\text{pred}}$; relative errors are $r$/pred.

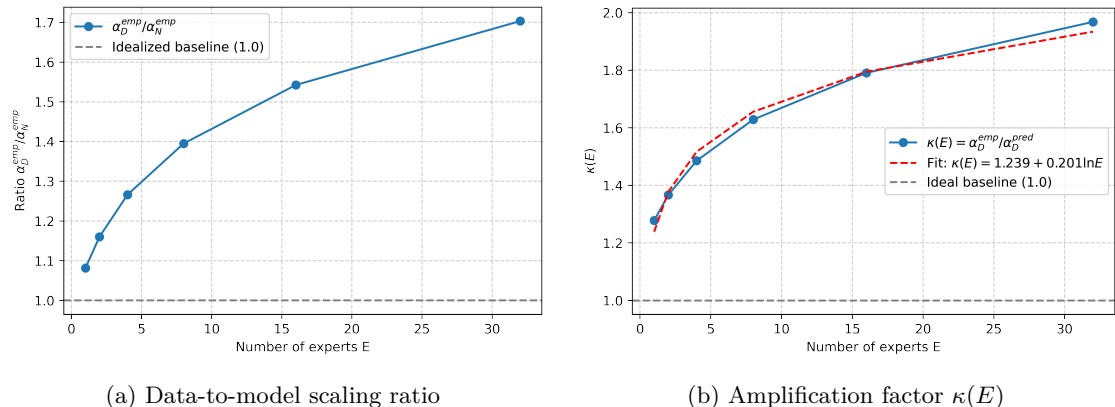

(a) Data-to-model scaling ratio

(b) Amplification factor $\kappa(E)$

Figure 5: Empirical amplification patterns and cross-budget fits.

**Compute-optimal allocation vs. experts.** JMSL reports compute-optimal allocations $(N_{\text{act}}^*(F,E), D^*(F,E))$, allowing us to test our predicted *tokens-to-parameters* drift. Our theory (via equation 18) imposes a fixed relationship between the model and data exponents, and along the compute frontier ($F = 6N_{\text{act}}D$) it implies the loss–compute exponent equation 21. Empirically, JMSL finds that, at fixed compute $F$, increasing the number of experts $E$ raises $D^*$ and lowers $N_{\text{act}}^*$ (Table 1 and isoFLOPs plots), consistent with a larger effective data exponent at higher $E$. This pattern appears directly in Fig. 6b: the ratio $D^*/N^*$ increases *monotonically* with $E$ for every budget $C$, meaning the compute-optimal allocation shifts toward *more tokens per active parameter* as expertization grows. Moreover, the slope of $D^*/N^*$ versus $E$ becomes *shallower* at larger budgets (e.g., the gain from $E=1\to16$ is roughly $\times 4.5$ at $10^{20}$ but only $\times 3.0$ at $10^{21}$), suggesting diminishing marginal amplification with $C$.

**Theory vs. EL (Tian et al., 2025).** Empirically, Tian et al. (2025) corroborate these predictions: their *Efficiency Leverage* metric (EL) grows as a power law in compute and (inverse) activation ratio while expert granularity modulates EL via a log-polynomial term with a stable optimum; their compute-optimal allocations favor *more data and smaller active models* than dense baselines, and the Ling-mini-beta case study (0.85B active, 17.5B total) matches a 6.1B dense model at $>7\times$ lower compute consistent with our view that active capacity drives gains and routing is a secondary, mostly constant-factor overhead (Tian et al., 2025).

**Theory vs. JMSL (Ludziejewski et al., 2025).** JMSL found that, for a fixed number of experts $E$, validation loss follows a Chinchilla-style surface and then introduces an *expert-dependent* joint law by letting both prefactors and exponents be functions of $E$ (via a smoothed $\hat{E}$). For each $E$, our theory suggests two simple checks: whether the empirical pair $(\hat{\alpha}_N(E), \hat{\alpha}_D(E))$ approximately satisfies $\hat{\alpha}_D \approx \hat{\alpha}_N/(1+\hat{\alpha}_N)$, and whether $\hat{\alpha}_C(E)$ computed from equation 21 aligns with the identity in equation 18. In the JMSL data we observe systematic, monotone-positive deviations: the ratio $\hat{\alpha}_D/\hat{\alpha}_N$ increases with $E$ (Table 4, Fig. 5a in appendix B), indicating stronger-than-idealized data-side gains, plausibly due to finite-sample effects, tokenization/entropy floors $c$, training schedules, or mild routing side-effects. Consistent with our compute-frontier geometry, JMSL's compute-optimal allocations $(N_{\text{act}}^*, D^*)$ show that $D^*/N^*$ increases *monotonically*

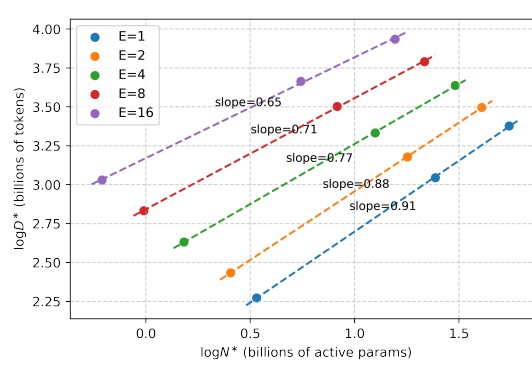 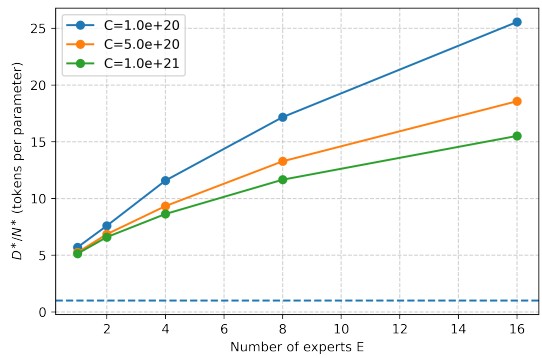

(a) Per-$E$ cross-budget fits: $D^*$ vs. $N^*$  (b) Tokens-to-parameters drift across E (per budget)

Figure 6: Empirical amplification patterns and cross-budget fits. (a) Cross-budget slopes for each $E$. (b) Tokens-to-parameters drift across experts (per budget).

with $E$ for every budget $C$ (Fig. 6b in appendix B), i.e., more tokens per active parameter as expertization grows, with a shallower slope at larger budgets, suggesting diminishing marginal amplification with compute.

*Practical implication.* At a fixed training budget $C$, increasing $E$ should allocate relatively fewer active parameters and *more* tokens (higher $D^\star/N^\star$), improving data efficiency.

## C   ADDITIONAL PRELIMINARIES

### C.1   NOTATION AND BASIC DEFINITIONS

**Sets, measures, and expectations.**   For a measurable space $(\mathcal{X}, \mathcal{B})$ and probability measure $Q$ on $(\mathcal{X}, \mathcal{B})$, we write $X \sim Q$ for a random variable taking values in $\mathcal{X}$, and $\mathbb{E}[\cdot]$ for expectation with respect to $Q$. The symbol $\mathbb{P}$ refers to probability.

**Covering numbers and metric entropy.**   Let $(\mathcal{F}, \rho)$ be a metric space. An $\delta$-*cover* of $\mathcal{F}$ is a finite subset $\{f_1, \ldots, f_N\} \subset \mathcal{F}$ such that for every $f \in \mathcal{F}$ there exists $j \in [N]$ with $\rho(f, f_j) \leq \delta$. The *covering number* Vershynin (2018) is

$$N(\delta, \mathcal{F}, \rho) := \min\{ N \in \mathbb{N} : \exists\ \delta\text{-cover of } \mathcal{F} \text{ with } N \text{ elements} \}$$

The (log) *metric entropy* is $\log N(\delta, \mathcal{F}, \rho)$. We often use $\rho(f, g) = \|f - g\|_\infty$.

**Vectors, matrices, tensors.**   We use bold uppercase (e.g., $H$) for matrices and bold lowercase (e.g., $h$) for vectors when convenient. For a vector $v \in \mathbb{R}^d$, define the entrywise $\ell_\infty$ norm

$$\|v\|_\infty := \max_{1 \leq i \leq d} |v_i|$$

For a matrix $A \in \mathbb{R}^{d_{\mathrm{out}} \times d_{\mathrm{in}}}$, we use two norms: $\|A\|_\infty := \max_{i,j} |A_{ij}|$

For an embedding matrix $H \in \mathbb{R}^{\ell \times d_{\mathrm{emb}}}$ (sequence length $\ell$, embedding size $d_{\mathrm{emb}}$), we adopt

$$\|H\|_\infty := \max_{t \in [\ell],\, j \in [d_{\mathrm{emb}}]} |H_{t,j}|$$

**Asymptotic notation.**   For nonnegative quantities $a, b$, $a \lesssim b$ means $a \leq C\, b$ for an absolute constant $C$; $a \simeq b$ means $a \lesssim b$ and $b \lesssim a$. We write $\widetilde{O}(\cdot)$ to hide polylogarithmic factors in problem parameters.

### C.2   LOSS AND LEARNING RULE

**Squared regression error.**   Given a predictor $T : \mathcal{X} \to \mathbb{R}$, the population (squared) risk under $Q$ is

$$L(T) := \mathbb{E}\big[(T(X) - f(X))^2\big] = \|T - f\|_{L^2(Q)}^2$$

Empirically, for samples $\{x_i\}_{i=1}^n$, the empirical risk is

$$L_n(T) := \frac{1}{n} \sum_{i=1}^n \big(T(x_i) - f(x_i)\big)^2$$

**Empirical risk minimization (ERM).** An empirical risk minimizer (ERM) over $\mathcal{T}$ is any measurable selection $\hat{T}_n \in \mathcal{T}$ satisfying

$$\hat{T}_n \in \arg\min_{T \in \mathcal{T}} L_n(T)$$

## C.3 INPUT-LIPSCHITZ BOUND

There exists $L_{\text{in}}^{(\text{dense})} \geq 1$, depending polynomially on $(\ell, m, d_{\text{emb}}, w_{\text{FFN}}, L_{\text{FFN}})$ and on $\kappa$, such that for all $H, H' \in \mathbb{R}^{\ell \times d_{\text{emb}}}$,

$$\|B_{\text{dense}}(\theta, H) - B_{\text{dense}}(\theta, H')\|_\infty \;\leq\; L_{\text{in}}^{(\text{dense})} \|H - H'\|_\infty, \tag{23}$$

with the schematic scaling (hiding absolute constants and activation Lipschitz factors)

$$L_{\text{in}}^{(\text{dense})} \;\lesssim\; 1 \;+\; C_{\text{LN}}\,\kappa\Big(\; \underbrace{\ell\,m\,d_{\text{emb}}^2}_{\text{MHA sensitivity}} \;+\; \underbrace{d_{\text{emb}}\,w_{\text{FFN}}^{L_{\text{FFN}}}}_{\text{FFN sensitivity}} \;\Big).$$

## C.4 SOFTMAX LIPSCHITZ

We work with the *temperature-$\tau$ softmax* $\sigma_\tau : \mathbb{R}^m \to \Delta^{m-1}$,

$$\sigma_\tau(u)_i \;=\; \frac{e^{u_i/\tau}}{\sum_{j=1}^m e^{u_j/\tau}}, \qquad \tau > 0.$$

Its Jacobian is

$$J_\tau(u) \;=\; \frac{1}{\tau}\Big(\text{Diag}(\sigma_\tau(u)) - \sigma_\tau(u)\sigma_\tau(u)^\top\Big).$$

**Lemma C.1** (Softmax is $(2\tau)^{-1}$-Lipschitz in $\|\cdot\|_\infty$). *For any $u, v \in \mathbb{R}^m$,*

$$\|\sigma_\tau(u) - \sigma_\tau(v)\|_\infty \;\leq\; \frac{1}{2\tau}\|u - v\|_\infty.$$

*Proof.* By the mean value theorem, $\sigma_\tau(u) - \sigma_\tau(v) = \int_0^1 J_\tau\big(v + t(u-v)\big)(u-v)\,dt$. Thus

$$\|\sigma_\tau(u) - \sigma_\tau(v)\|_\infty \;\leq\; \sup_{w \in [u,v]} \|J_\tau(w)\|_{\infty \to \infty}\|u - v\|_\infty.$$

Row $i$ of $J_\tau(w)$ has entries $\frac{1}{\tau}\sigma_i(\delta_{ij} - \sigma_j)$. Summing absolute values in row $i$:

$$\sum_{j=1}^m \frac{1}{\tau}\sigma_i|\delta_{ij} - \sigma_j| = \frac{1}{\tau}\Big(\sigma_i(1-\sigma_i) + \sum_{j\neq i}\sigma_i\sigma_j\Big) = \frac{2}{\tau}\,\sigma_i(1-\sigma_i) \;\leq\; \frac{1}{2\tau},$$

since $x(1-x) \leq 1/4$. Hence $\|J_\tau(w)\|_{\infty \to \infty} \leq 1/(2\tau)$ uniformly, proving the claim. $\square$

**Corollary C.2** (Row-wise softmax on matrices). *Let $S, S' \in \mathbb{R}^{\ell \times \ell}$ and apply $\text{softmax}_\tau$ row-wise to obtain $A = \text{softmax}_\tau(S)$ and $A' = \text{softmax}_\tau(S')$. Then*

$$\|A - A'\|_\infty \;\leq\; \frac{1}{2\tau}\|S - S'\|_\infty.$$

*In scaled attention, $\tau = \sqrt{d_h}$, hence $\|A - A'\|_\infty \leq \frac{1}{2\sqrt{d_h}}\|S - S'\|_\infty$.*

**Remark C.3** (Row-stochastic contraction). *If $A$ is row-stochastic (each row in $\Delta^{\ell-1}$), then $\|A\|_{\infty \to \infty} = 1$ and $\|AV\|_\infty \leq \|V\|_\infty$ for any $V$ under operator norm. When using entrywise $\|\cdot\|_\infty$, we will also invoke the safe bound $\|AV\|_\infty \leq \ell\|A\|_\infty\|V\|_\infty$ (cf. C.4).*

## C.5 Matrix norm identities and product bounds

**Lemma C.4** (Basic matrix inequalities). *Let $A \in \mathbb{R}^{p \times q}$, $B \in \mathbb{R}^{q \times r}$, $u \in \mathbb{R}^q$, $H \in \mathbb{R}^{\ell \times q}$.*

*(a) **Operator submultiplicativity:** $\|AB\|_{\infty \to \infty} \leq \|A\|_{\infty \to \infty} \|B\|_{\infty \to \infty}$.*

*(b) **Entrywise-to-operator:** $\|A\|_{\infty \to \infty} \leq q \|A\|_\infty$ and $\|Au\|_\infty \leq q \|A\|_\infty \|u\|_\infty$.*

*(c) **Products in entrywise norm:** $\|AB\|_\infty \leq q \|A\|_\infty \|B\|_\infty$ and $\|HB\|_\infty \leq q \|H\|_\infty \|B\|_\infty$.*

*(d) **Row-stochastic contraction:** If each row of $A$ sums to 1 and is nonnegative, then $\|A\|_{\infty \to \infty} = 1$ and $\|AB\|_{\infty \to \infty} \leq \|B\|_{\infty \to \infty}$.*

*(e) **Hadamard product:** $\|A \odot B\|_\infty \leq \|A\|_\infty \|B\|_\infty$.*

*Proof.* Parts (a) and (d) follow from definitions; (b)–(c) use $\|A\|_{\infty \to \infty} = \max_i \sum_j |A_{ij}| \leq q \|A\|_\infty$ and standard norm inequalities; (e) is immediate from entrywise multiplication. $\square$

**Lemma C.5** (Scaled dot-product attention ingredients). *Let $H \in \mathbb{R}^{\ell \times d_{\mathrm{emb}}}$, and $W_Q, W_K, W_V$ satisfy $\|W_\bullet\|_\infty \leq \kappa$. Set $Q = HW_Q$, $K = HW_K$, $V = HW_V$, scores $S = QK^\top/\sqrt{d_h}$ and $A = \mathrm{softmax}_{\sqrt{d_h}}(S)$ (row-wise). Then*

$$\|Q\|_\infty, \ \|K\|_\infty, \ \|V\|_\infty \leq d_{\mathrm{emb}} \, \kappa \, \|H\|_\infty,$$

$$\|S\|_\infty \leq \sqrt{d_h} \, d_{\mathrm{emb}}^2 \, \kappa^2 \, \|H\|_\infty^2,$$

$$\|A - A'\|_\infty \leq \frac{1}{2\sqrt{d_h}} \|S - S'\|_\infty \qquad \text{(row-wise softmax; C.2)},$$

$$\|AV\|_\infty \leq \ell \, \|A\|_\infty \, \|V\|_\infty \quad \text{(entrywise bound; cf. C.3)}.$$

## C.6 Taylor remainder on manifolds

Let $\mathcal{M} \subset \mathbb{R}^D$ be a compact $C^1$ submanifold of intrinsic dimension $d$. Fix a coordinate chart $\phi : U \to V \subset \mathbb{R}^d$ that is bi-Lipschitz with constants $L, L' > 0$: $L^{-1} \|x - y\| \leq \|\phi(x) - \phi(y)\| \leq L \|x - y\|$ for $x, y \in U$ and similarly for $\phi^{-1}$ on $V$.

**Lemma C.6** (Local Taylor remainder for $C^\beta$ functions on $\mathcal{M}$). *Let $f \in C^\beta(\mathcal{M})$ with $\beta > 0$, $s = \lfloor \beta \rfloor$, and $\alpha = \beta - s \in [0, 1)$. Fix a chart $(U, \phi)$ and a point $x_0 \in U$. Let $P$ be the degree-$s$ Taylor polynomial of $f \circ \phi^{-1}$ at $z_0 = \phi(x_0)$ in local coordinates, mapped back to $\mathcal{M}$ by $P_{\mathcal{M}} := P \circ \phi$. Then for every $x \in U$,*

$$|f(x) - P_{\mathcal{M}}(x)| \leq C(\beta, d, L, L') \|f\|_{C^\beta(U)} \|\phi(x) - \phi(x_0)\|^\beta \leq C' \|f\|_{C^\beta(U)} \|x - x_0\|^\beta.$$

*In particular, on any chart of diameter at most $r$, $\sup_{x \in U} |f(x) - P_{\mathcal{M}}(x)| \leq C \, r^\beta$.*

*Proof.* Apply the standard Taylor remainder in $\mathbb{R}^d$ to $f \circ \phi^{-1} \in C^\beta(V)$ and use bi-Lipschitz distortion to convert local distance in $V$ to ambient distance in $\mathcal{M}$; constants depend polynomially on $L, L'$ and on bounds of chart derivatives. $\square$

## C.7 Partitions of unity with bounded overlap

**Lemma C.7** (Partition of unity subordinate to a bounded-overlap cover). *Let $\mathcal{M} \subset \mathbb{R}^D$ be a compact $C^1$ manifold and fix $r \in (0, 1)$. There exists a finite cover of $\mathcal{M}$ by open sets $\{U_\nu\}_{\nu=1}^N$ such that: (i) $\mathrm{diam}(U_\nu) \leq r$, (ii) each $x \in \mathcal{M}$ belongs to at most $s_0(d)$ sets (bounded overlap), and (iii) a $C^\infty$ partition of unity $\{\varphi_\nu\}_{\nu=1}^N$ subordinate to $\{U_\nu\}$ with $0 \leq \varphi_\nu \leq 1$, $\sum_\nu \varphi_\nu(x) = 1$ for all $x$, and the Lipschitz bound*

$$\|\nabla \varphi_\nu(x)\| \leq \frac{C(d)}{r} \qquad \text{for all } x \in \mathcal{M} \text{ and all } \nu.$$

*Consequently, at every $x$ at most $s_0(d)$ of the $\varphi_\nu(x)$ are nonzero.*

*Proof.* **Step 1: A bounded-overlap cover by small balls.** Work with the metric induced on $M$ by the ambient Euclidean norm on $\mathbb{R}^D$. Since $M$ is compact and $C^1$, there exists a finite atlas $\{(V_j, \varphi_j)\}_{j=1}^J$ such that:

- each $\varphi_j : V_j \to \mathbb{R}^d$ is a bi-Lipschitz chart onto its image, and

- on each $V_j$ the volume of metric balls is comparable to that of Euclidean $d$-balls in $\mathbb{R}^d$ (with constants depending only on $d$ and the chart regularity).

Fix a small radius $\rho > 0$ proportional to $r$ (we will specify it below), and choose a *maximal* $\rho/2$-separated set $\{x_\nu\}_{\nu=1}^N \subset M$, i.e.,

$$\|x_\nu - x_\mu\| \; \geq \; \frac{\rho}{2} \quad \text{for all } \nu \neq \mu,$$

and the union of balls $B(x_\nu, \rho)$ (intersection with $M$) covers $M$. Such a set exists by a standard greedy argument: repeatedly add points that are at distance at least $\rho/2$ from all previously chosen ones until no further point can be added.

Define

$$U_\nu := B(x_\nu, \rho) \cap M$$

By construction, $\{U_\nu\}_{\nu=1}^N$ is an open cover of $M$. Choosing $\rho \leq r/2$ ensures $(U_\nu) \leq r$.

We now show that the overlap multiplicity is bounded. Fix any $x \in M$ and consider the index set

$$I(x) := \{\nu : x \in U_\nu\} = \{\nu : \|x - x_\nu\| < \rho\}$$

For each $\nu \in I(x)$, the balls $B(x_\nu, \rho/4)$ are pairwise disjoint (by $\rho/2$-separation) and all lie inside $B(x, 2\rho)$:

$$B(x_\nu, \rho/4) \subset B(x, \rho + \rho/4) \subset B(x, 2\rho)$$

Using the chart bi-Lipschitz property and volume comparability on $M$, there exist constants $c_1, c_2 > 0$ depending only on $d$ and the atlas such that for any ball $B_M(y, t)$ in $M$ (with respect to the induced metric),

$$c_1 t^d \; \leq \; \mathrm{Vol}_M \left( B_M(y, t) \right) \; \leq \; c_2 t^d$$

Hence, the volumes of the disjoint balls $\{B(x_\nu, \rho/4)\}_{\nu \in I(x)}$ satisfy

$$|I(x)| \cdot c_1 (\rho/4)^d \; \leq \; \sum_{\nu \in I(x)} \mathrm{Vol}_M \left( B(x_\nu, \rho/4) \right) \; \leq \; \mathrm{Vol}_M \left( B(x, 2\rho) \right) \; \leq \; c_2 (2\rho)^d$$

Therefore

$$|I(x)| \; \leq \; \frac{c_2 (2\rho)^d}{c_1 (\rho/4)^d} \; = \; C(d) \quad \text{for some constant } C(d),$$

depending only on $d$ and the chart regularity. Setting $s_0(d) := C(d)$ yields the bounded-overlap property: each $x \in M$ belongs to at most $s_0(d)$ of the sets $U_\nu$.

**Step 2: Smooth bump functions subordinate to the cover.** Let $\eta : \mathbb{R}^D \to [0, 1]$ be a fixed $C^\infty$ bump function such that:

$$\eta(z) = 1 \text{ if } \|z\| \leq 1/2, \qquad \eta(z) = 0 \text{ if } \|z\| \geq 1, \qquad \|\nabla \eta(z)\| \leq C_0(d)$$

For each $\nu$, define

$$\psi_\nu(x) \; := \; \eta\left(\frac{x - x_\nu}{\rho}\right), \qquad x \in M$$

Then $\psi_\nu \in C^\infty(M)$, $\psi_\nu(x) \in [0, 1]$, and

$$\psi_\nu(x) = 1 \quad \text{if } \|x - x_\nu\| \leq \frac{\rho}{2}, \qquad \psi_\nu(x) = 0 \quad \text{if } \|x - x_\nu\| \geq \rho$$

Hence $\mathrm{supp}(\psi_\nu) \subset U_\nu$, i.e., each $\psi_\nu$ is subordinate to $U_\nu$. Moreover,

$$\|\nabla \psi_\nu(x)\| \; = \; \frac{1}{\rho} \left\| \nabla \eta\left(\frac{x - x_\nu}{\rho}\right) \right\| \; \leq \; \frac{C_0(d)}{\rho}$$

Because the balls $B(x_\nu, \rho/2)$ cover $M$, for every $x \in M$ there exists at least one index $\nu$ with $\|x - x_\nu\| \leq \rho/2$, hence $\psi_\nu(x) = 1$. Therefore

$$S(x) := \sum_{\mu=1}^N \psi_\mu(x) \; \geq \; 1 \quad \text{for all } x \in M.$$

On the other hand, for each fixed $x$, at most $s_0(d)$ of the $\psi_\mu(x)$ are nonzero (because $\psi_\mu$ vanish outside $U_\mu$ and the $U_\mu$ have overlap at most $s_0(d)$), and each $\psi_\mu(x) \leq 1$, so

$$1 \leq S(x) \leq s_0(d) \quad \text{for all } x \in M.$$

**Step 3: Normalize to obtain a partition of unity and bound gradients.** Define

$$\phi_\nu(x) := \frac{\psi_\nu(x)}{S(x)}, \qquad x \in M$$

Then each $\phi_\nu \in C^\infty(M)$, $0 \leq \phi_\nu \leq 1$, and

$$\sum_{\nu=1}^N \phi_\nu(x) = \frac{1}{S(x)} \sum_{\nu=1}^N \psi_\nu(x) = 1 \quad \text{for all } x \in M.$$

Moreover, $\mathrm{supp}(\phi_\nu) \subset \mathrm{supp}(\psi_\nu) \subset U_\nu$, so $\{\phi_\nu\}$ is subordinate to the cover $\{U_\nu\}$.

We now bound the gradient. Using the quotient rule,

$$\nabla \phi_\nu(x) = \frac{\nabla \psi_\nu(x) S(x) - \psi_\nu(x) \nabla S(x)}{S(x)^2}, \qquad \nabla S(x) = \sum_{\mu=1}^N \nabla \psi_\mu(x).$$

Thus

$$\|\nabla \phi_\nu(x)\| \leq \frac{\|\nabla \psi_\nu(x)\| \, S(x) + |\psi_\nu(x)| \, \|\nabla S(x)\|}{S(x)^2}$$

$$\leq \frac{\|\nabla \psi_\nu(x)\|}{S(x)} + \frac{\|\nabla S(x)\|}{S(x)^2}$$

We already know $1 \leq S(x) \leq s_0(d)$; hence $1/S(x), 1/S(x)^2 \leq 1$. Also, for each $x$, at most $s_0(d)$ functions $\psi_\mu$ are nonzero, so

$$\|\nabla S(x)\| = \Big\| \sum_\mu \nabla \psi_\mu(x) \Big\| \leq \sum_{\mu : \psi_\mu(x) \neq 0} \|\nabla \psi_\mu(x)\| \leq s_0(d) \cdot \frac{C_0(d)}{\rho}$$

Combining these bounds gives

$$\|\nabla \phi_\nu(x)\| \leq \frac{C_0(d)}{\rho} + \frac{s_0(d) C_0(d)}{\rho} \leq \frac{C(d)}{\rho}$$

for some constant $C(d)$ depending only on $d$ and the atlas. Choosing $\rho$ proportional to $r$ (e.g., $\rho = r/2$ as above) yields

$$\|\nabla \phi_\nu(x)\| \leq \frac{C(d)}{r}$$

Finally, since at most $s_0(d)$ of the $\psi_\nu(x)$ are nonzero at any $x$, the same holds for the $\phi_\nu(x)$, and the "consequently" statement follows. $\square$

**Remark C.8** (From partition of unity to $k$-sparse mixing)**.** *Since at most $s_0(d)$ functions are nonzero at any $x$, choosing $k \geq s_0(d)$ allows a router to implement a $k$-sparse mixture $\sum_{\nu \in S(x)} w_\nu(x) E_\nu(x)$ with weights $w_\nu = \varphi_\nu$ (or an approximation thereof), as used in the approximation construction.*

## C.8 Useful norm conversions for attention chains

For attention chains $H \mapsto Q \mapsto K \mapsto S \mapsto A \mapsto O = AV$, combining Lemmas C.4, C.5 and C.2 yields the schematic bounds used in the main text:

$$\|Q - Q'\|_\infty, \ \|K - K'\|_\infty, \ \|V - V'\|_\infty \leq d_{\mathrm{emb}} \, \kappa \, \|H - H'\|_\infty + d_{\mathrm{emb}} \|W_\bullet - W'_\bullet\|_\infty \|H'\|_\infty,$$

$$\|S - S'\|_\infty \lesssim \sqrt{d_h} \, d_{\mathrm{emb}}^2 \, \kappa \Big( \|Q - Q'\|_\infty + \|K - K'\|_\infty \Big),$$

$$\|A - A'\|_\infty \leq \frac{1}{2\sqrt{d_h}} \|S - S'\|_\infty,$$

$$\|AV - A'V'\|_\infty \leq \underbrace{\|A - A'\|_{\infty \to \infty} \|V\|_\infty}_{\text{attention change}} + \underbrace{\|A'\|_{\infty \to \infty} \|V - V'\|_\infty}_{\text{value change}} \lesssim \|A - A'\|_\infty \|V\|_\infty + \|V - V'\|_\infty,$$

using $\|A'\|_{\infty \to \infty} = 1$. These estimates justify the polynomial dependence on $(\ell, m, d_{\mathrm{emb}}, \kappa)$ that appears in the MHA stability bound.

## D  FULL MODEL SPECIFICATION

**Definition D.1** (Dense Transformer Block). *We define the dense residual block based on the canonical Transformer structure Vaswani et al. (2017). Let the input to block $j$ be $H \in \mathbb{R}^{\ell \times d_{\mathrm{emb}}}$ with tokenwise $\ell_\infty$-bound $\|H\|_\infty \leq M_0$. The block maps $H \mapsto H_{\mathrm{out}}$ via the sequential steps:*

$$\widetilde{H} = H + \mathrm{MHA}_\psi(H), \tag{24}$$

$$H_{\mathrm{out}} = \widetilde{H} + \mathrm{FFN}_\chi(\widetilde{H}), \tag{25}$$

*where $\theta = \{\psi, \chi\}$ collects all block parameters.*

**Layer Normalization (LN) Ba et al. (2016).**  *For each token $t \in [\ell]$,*

$$\mathrm{LN}_{\gamma,\beta}(H)_t = \gamma \odot \frac{H_t - \mu(H_t)\mathbf{1}}{\sqrt{\sigma^2(H_t) + \epsilon}} + \beta, \qquad \gamma, \beta \in \mathbb{R}^{d_{\mathrm{emb}}}$$

*with learned scale/shift $(\gamma, \beta)$ and small $\epsilon > 0$. We will use the uniform magnitude bounds*

$$\|\gamma\|_\infty \leq \kappa, \qquad \|\beta\|_\infty \leq \kappa \tag{26}$$

**Multi-Head Self-Attention (MHA).**  *Let the number of heads be $m$, and the per-head key/query/value dimension be $d_h$ (so $d_k = d_v = d_h$) such that $m\, d_h = d_{\mathrm{emb}}$. The input to MHA is $H$. For head $h \in [m]$ we have projections*

$$Q^{(h)} = HW_Q^{(h)}, \quad K^{(h)} = HW_K^{(h)}, \quad V^{(h)} = HW_V^{(h)}, \qquad W_Q^{(h)}, W_K^{(h)}, W_V^{(h)} \in \mathbb{R}^{d_{\mathrm{emb}} \times d_h},$$

*scores $S^{(h)} = \frac{1}{\sqrt{d_h}} Q^{(h)}(K^{(h)})^\top \in \mathbb{R}^{\ell \times \ell}$, row-wise $A^{(h)} = \mathrm{softmax}(S^{(h)})$, head output $O^{(h)} = A^{(h)}V^{(h)} \in \mathbb{R}^{\ell \times d_h}$, and*

$$\mathrm{MHA}_\psi(H) = \left[\mathrm{Concat}_{h=1}^m O^{(h)}\right] W_O, \qquad W_O \in \mathbb{R}^{(md_h) \times d_{\mathrm{emb}}}$$

*We assume the uniform parameter bounds*

$$\|W_Q^{(h)}\|_\infty, \ \|W_K^{(h)}\|_\infty, \ \|W_V^{(h)}\|_\infty, \ \|W_O\|_\infty \ \leq \ \kappa \tag{27}$$

**Positionwise Feed-Forward Network (FFN).**  *We use an $L_{\mathrm{FFN}}$-layer MLP applied tokenwise. Let widths be at most $W_{\mathrm{FFN}}$. For $\ell = 1, \ldots, L_{\mathrm{FFN}}$ and token $t$,*

$$z_t^{(0)} = (\widetilde{H})_t, \qquad z_t^{(\ell)} = \sigma\big(W^{(\ell)} z_t^{(\ell-1)} + b^{(\ell)}\big)$$

*with activations $\sigma$ (ReLU/GELU), $W^{(1)} \in \mathbb{R}^{W_{\mathrm{FFN}} \times d_{\mathrm{emb}}}$, $W^{(\ell)} \in \mathbb{R}^{W_{\mathrm{FFN}} \times W_{\mathrm{FFN}}}$ for $2 \leq \ell \leq L_{\mathrm{FFN}} - 1$, and $W^{(L_{\mathrm{FFN}})} \in \mathbb{R}^{d_{\mathrm{emb}} \times W_{\mathrm{FFN}}}$; biases $b^{(\ell)}$ are conformal. The FFN output is $\mathrm{FFN}_\chi(\widetilde{H})_t = z_t^{(L_{\mathrm{FFN}})}$, and we assume*

$$\|W^{(\ell)}\|_\infty \leq \kappa, \qquad \|b^{(\ell)}\|_\infty \leq \kappa, \qquad \ell = 1, \ldots, L_{\mathrm{FFN}} \tag{28}$$

**Output bound and architectural polynomial.**  We assume the block output is uniformly bounded by $\|B_{\mathrm{dense}}(\theta, H)\|_\infty \leq R$ for all admissible $\theta$ and $\|H\|_\infty \leq M_0$ Havrilla & Liao (2024); Chen et al. (2022). Constants in our bounds will be expressed via a fixed architecture polynomial

$$P_{\mathrm{arch}}^{(\mathrm{dense})} = \mathrm{poly}\big(\ell, m, d_{\mathrm{emb}}, w_{\mathrm{FFN}}, L_{\mathrm{FFN}}\big)$$

**Hypothesis classes.**  We consider a hypothesis class $\mathcal{T}$ of predictors. In this paper, $\mathcal{T}$ is either the dense Transformer class (D.1) or the MoE class $\mathcal{T}_{\mathrm{MoE}}$ (D.2). We always assume $\sup_{T \in \mathcal{T}} \|T\|_\infty \leq R$.

### D.1  MOE TRANSFORMER

**Definition D.2** (MoE Transformer Class). *Fix ambient input dimension $D$, sequence length $\ell$, embedding dimension $d_{\mathrm{emb}}$, number of heads $m$, number of Transformer blocks $L_T$, number of experts per MoE layer $M$, and top-$k$ activation per token. Let $L_{\mathrm{FFN}}$ and $W_{\mathrm{FFN}}$ denote the depth and width of each expert MLP (and of the non-expert FFN). Let $\kappa_{\mathcal{B}} > 0$ bound the magnitude of all learned scalar parameters and let $R_{\max} > 0$ bound the output of the final block.*

*The input is a sequence $x = (x_1, \ldots, x_\ell) \in \mathcal{X} \subset \mathbb{R}^{\ell \times D}$ with $\|x\|_\infty \leq M_0$. It is mapped to token embeddings $H^{(0)} = E(x) + \mathrm{PE} \in \mathbb{R}^{\ell \times d_{\mathrm{emb}}}$ by a bounded embedding map $E$ and a fixed positional encoding PE.*

**Sequential MoE Block.** *Let the input to block $j$ be $H^{(j-1)} \in \mathbb{R}^{\ell \times d_{\mathrm{emb}}}$ with tokenwise $\ell_\infty$-bound $\|H^{(j-1)}\|_\infty \leq M_0$. For $j = 1, \ldots, L_T$, the block maps $H^{(j-1)} \mapsto H^{(j)}$ via the following sequential steps, where $\mathrm{LN}_1$ and $\mathrm{LN}_2$ denote layer normalization:*

1. ***Self-Attention Sub-block:***

$$\widetilde{H}^{(j-1)} = H^{(j-1)} + \mathrm{MHA}_{\psi_j}\Big(\mathrm{LN}_1(H^{(j-1)})\Big)$$

2. ***MoE/FFN Sub-block (Tokenwise):*** *The output $\widetilde{H}^{(j-1)}$ is then passed through an FFN path which is a composition of a non-expert FFN and an expert mixture (MoE).*

$$H^{(j)} = \widetilde{H}^{(j-1)} + \mathrm{FFN}^{\mathrm{mix}}_{\chi_j}\Big(\mathrm{LN}_2(\widetilde{H}^{(j-1)})\Big)$$

*Here $\mathrm{FFN}^{\mathrm{mix}}_{\chi_j}(H)$ denotes the combination of the non-expert FFN and the MoE:*

$$\mathrm{FFN}^{\mathrm{mix}}_{\chi_j}(H) = \underbrace{\mathrm{FFN}^{\mathrm{nonexp}}_{\chi_j}(H)}_{\textit{non-expert FFN}} + \underbrace{\mathrm{MoE}^{(j)}_{\phi_j}(H)}_{\textit{expert mixture}}$$

*The total scalar parameters for the FFN path are $\chi_j = (\chi_{j,\mathrm{nonexp}}, \phi_j)$.*

*Here $\mathrm{LN}$ is tokenwise LayerNorm (Ba et al., 2016) with learned $(\gamma, \beta) \in \mathbb{R}^d_{\mathrm{emb}}$. $\mathrm{MHA}_{\psi_j}$ is standard multi-head self-attention with $m$ heads and per-head size $d_h = d_{\mathrm{emb}}/m$; $\mathrm{FFN}^{\mathrm{nonexp}}_{\chi_{j,\mathrm{nonexp}}}$ is a tokenwise MLP of depth $L_{\mathrm{FFN}}$ and width $W_{\mathrm{FFN}}$ mapping $\mathbb{R}^d_{\mathrm{emb}} \to \mathbb{R}^d_{\mathrm{emb}}$. All their scalar parameters are collected in $\psi_j$ and $\chi_j$.*

**Experts and router.** Each block $j$ has $M$ experts $\{E_{j,i}\}_{i=1}^M$, each a tokenwise MLP $E_{j,i} : \mathbb{R}^d_{\mathrm{emb}} \to \mathbb{R}^d_{\mathrm{emb}}$ of depth $L_{\mathrm{FFN}}$ and width $w_{\mathrm{FFN}}$ sharing the same architecture. For a token $t \in [\ell]$, let $Z^{(j)}_t = H^{(j-1)}_t \in \mathbb{R}^d_{\mathrm{emb}}$ be the MoE input. The router in block $j$ computes logits $g_{j,i}(Z^{(j)}_t)$ for $i \in [M]$ (by a bounded parametric map) and selects the *hard* top-$k$ index set

$$S_{j,t}(H^{(j-1)}) \subset [M], \qquad |S_{j,t}(H^{(j-1)})| = k$$

consisting of the $k$ largest logits. The MoE output at token $t$ is then the *$k$-sparse* mixture

$$\mathrm{MoE}^{(j)}\big(Z^{(j)}\big)_t = \sum_{i \in S_{j,t}(H^{(j-1)})} w_{j,i}(Z^{(j)}_t)\, E_{j,i}(Z^{(j)}_t), \qquad w_{j,i}(Z^{(j)}_t) \geq 0, \quad \sum_{i \in S_{j,t}} w_{j,i}(Z^{(j)}_t) = 1 \qquad (29)$$

with router weights $w_{j,i}$ restricted to the selected indices.

**Readout and parameter set.** A fixed linear readout $R : \mathbb{R}^{\ell \times d_{\mathrm{emb}}} \to \mathbb{R}$ produces the scalar prediction

$$T_\theta(x) = R\big(H^{(L_T)}\big).$$

We define the *MoE Transformer network class* as the set of all functions

$$\mathcal{T}_{\mathrm{MoE}}(D, M, k, L_T, L_{\mathrm{FFN}}, w_{\mathrm{FFN}}, d_{\mathrm{emb}}, m, \kappa, R) := \Big\{ T_\theta : \mathcal{X} \to \mathbb{R} \,\Big|\, \|\theta\|_\infty \leq \kappa, \; \|T_\theta\|_\infty \leq R \Big\} \qquad (30)$$

### D.2 Per-block parameter counts (dense baseline).

Ignoring biases for clarity and taking the standard choice $d_h = d_{\mathrm{emb}}/m$:

$$\Pi^{(\mathrm{dense})}_{\mathrm{attn}} = \underbrace{3\, d_{\mathrm{emb}} d_h\, m}_{W_Q, W_K, W_V} + \underbrace{(m d_h) d_{\mathrm{emb}}}_{W_O} = 4\, d^2_{\mathrm{emb}},$$

$$\Pi^{(\mathrm{dense})}_{\mathrm{FFN}} = d_{\mathrm{emb}} w_{\mathrm{FFN}} + (L_{\mathrm{FFN}} - 2)\, w^2_{\mathrm{FFN}} + w_{\mathrm{FFN}} d_{\mathrm{emb}} = 2\, d_{\mathrm{emb}} w_{\mathrm{FFN}} + (L_{\mathrm{FFN}} - 2)\, w^2_{\mathrm{FFN}},$$

. With biases, add $3\, m d_h + d_{\mathrm{emb}}$ (attention) and $w_{\mathrm{FFN}}(L_{\mathrm{FFN}} - 1) + d_{\mathrm{emb}}$ (FFN).

**Parameter-Lipschitz (baseline).** For two parameter sets $\theta, \theta'$ with $\|\theta - \theta'\|_\infty \leq \eta$ and fixed input $H$,

$$\|B_{\text{dense}}(\theta, H) - B_{\text{dense}}(\theta', H)\|_\infty \leq C_{\text{block}}^{(\text{dense})} \left( \ell\, m\, d_{\text{emb}}^2 + d_{\text{emb}}\, w_{\text{FFN}}^{L_{\text{FFN}}} \right) \kappa\, \eta \tag{31}$$

obtained by summing the MHA and FFN parameter-perturbation bounds (each applied after LN) and absorbing LN perturbations into constants. These two baseline inequalities equation 31-equation 23 are the dense counterparts of the MoE block bounds where the FFN sensitivity term gains an additional factor $k$ from the $k$ active experts.

### D.3 ROUTING PATTERNS.

For each block $j \in [L_T]$ and token $t \in [\ell]$, the router chooses a $k$-subset $S_{j,t} \subset [M]$. A *routing pattern* is the collection

$$\pi = \{ S_{j,t} \subset [M] : |S_{j,t}| = k,\ j = 1, \ldots, L_T,\ t = 1, \ldots, \ell \}$$

Let $\Pi$ denote the set of all such patterns. Its cardinality is bounded by

$$|\Pi| \leq \left( \binom{M}{k} \right)^{L_T \ell} \quad \Longrightarrow \quad \log |\Pi| \leq L_T\, \ell\, k\, \log\!\left( \frac{e\, M}{k} \right) \tag{32}$$

using $\binom{M}{k} \leq (e\, M/k)^k$. For a fixed $\pi \in \Pi$, denote by $\mathcal{T}_\pi \subset \mathcal{T}_{\text{MoE}}$ the subclass with *deterministic* routing equal to $\pi$. Then

$$\mathcal{T}_{\text{MoE}} = \bigcup_{\pi \in \Pi} \mathcal{T}_\pi$$

## E PROOF OF THEOREM 3.3

Assume $\sup_{T \in \mathcal{T}_{\text{MoE}}} \|T\|_\infty \leq R$ and $\|f\|_\infty \leq R$. Let $\hat{T}_n \in \arg\min_{T \in \mathcal{T}_{\text{MoE}}} L_n(T)$ be an ERM for squared loss $L(T) = \mathbb{E}[(T(X) - f(X))^2]$. Then for any cover scale $\delta \in (0, 1)$, via the standard bias–variance decomposition described in lemma E.1, we have the following inequality

$$\mathbb{E}\big[\|\hat{T}_n - f\|_{L^2(Q)}^2\big] \leq 3 \inf_{T \in \mathcal{T}_{\text{MoE}}} \|T - f\|_\infty^2 + C\, R^2 \left( \sqrt{\frac{\log N(\delta, \mathcal{T}_{\text{MoE}}, \|\cdot\|_\infty)}{n}} + \delta \right) \tag{33}$$

for an absolute constant $C > 0$. Equivalently,

$$\mathbb{E}\big[\|\hat{T}_n - f\|_{L^2(Q)}^2\big] \lesssim \inf_{T \in \mathcal{T}_{\text{MoE}}} \|T - f\|_\infty^2 + \frac{\log N(\delta, \mathcal{T}_{\text{MoE}}, \|\cdot\|_\infty)}{n} + \delta \tag{34}$$

where $\lesssim$ hides universal constants and a factor $R^2$.

Now it remains to bound the two terms of 34. The first is bounded in theorem 3.1 as:

$$\inf_{T \in \mathcal{T}_{\text{MoE}}} \|T - f\|_\infty^2 \leq \epsilon^2 \tag{35}$$

The second terms is bounded using the cover number from lemma 3.6 as follows

$$\log N(\delta, \mathcal{T}_{\text{MoE}}, \|\cdot\|_\infty) \leq C_1 \Big( \Pi_{\text{attn}} + L_T k \Pi_{\text{exp}} \Big) \log\!\left( \frac{C_2 \kappa R M_0}{\delta} \right) + C_3 L_T \ell k \log\!\left( \frac{eM}{k} \right) \tag{36}$$

Combining equation 35 and equation 53 yields to the desired inequality

$$\mathbb{E}\big[\|\hat{T}_n - f\|_{L^2(Q)}^2\big] \leq \widetilde{O}\!\left( N^{-2\beta/d} + \frac{N + L_T \ell k \log(eM/k)}{n} \right) \tag{37}$$

$\square$

**Lemma E.1** (ERM bound via symmetrization and covering numbers). *Write*

$$L(T) = \|T - f\|_{L^2(Q)}^2 \quad and \quad L_n(T) = \frac{1}{n}\sum_{i=1}^n (T(x_i) - f(x_i))^2.$$

*Fix $\delta \in (0,1)$ and let $\mathcal{G}$ be a minimal $\delta$-net of $\mathcal{T}_{\mathrm{MoE}}$ in $\|\cdot\|_\infty$, so $|\mathcal{G}| = N(\delta, \mathcal{T}_{\mathrm{MoE}}, \|\cdot\|_\infty)$ and for all $T$ there exists $G \in \mathcal{G}$ with $\|T - G\|_\infty \le \delta$.*

(i) Lipschitz transfer from predictors to losses. *For any $a, b, c \in [-R, R]$,*

$$|(a-c)^2 - (b-c)^2| = |(a-b)(a+b-2c)| \le 4R\,|a-b|$$

*Hence for all $T, G$,*

$$|L(T) - L(G)| \le 4R\,\|T - G\|_\infty, \qquad |L_n(T) - L_n(G)| \le 4R\,\|T - G\|_\infty \tag{38}$$

(ii) ERM reduction to the finite cover. *Let $T^\star \in \arg\min_{T \in \mathcal{T}_{\mathrm{MoE}}} L(T)$ and choose $G^\star \in \mathcal{G}$ with $\|G^\star - T^\star\|_\infty \le \delta$. By ERM optimality, $L_n(\hat{T}_n) \le L_n(G^\star)$. Adding and subtracting $L$ and using equation 38,*

$$L(\hat{T}_n) \le L_n(G^\star) + \sup_{G \in \mathcal{G}} |L(G) - L_n(G)| \le L(G^\star) + \sup_{G \in \mathcal{G}} |L(G) - L_n(G)| + 4R\,\delta$$

*Also $L(G^\star) \le L(T^\star) + 4R\,\delta$. Thus*

$$L(\hat{T}_n) \;\le\; \inf_{T \in \mathcal{T}_{\mathrm{MoE}}} L(T) \;+\; 8R\,\delta \;+\; \sup_{G \in \mathcal{G}} |L(G) - L_n(G)| \tag{39}$$

(iii) Symmetrization over the finite cover. *Define the bounded loss class $\mathcal{F} = \{x \mapsto (G(x) - f(x))^2 : G \in \mathcal{G}\}$ with range in $[0, (2R)^2]$. By symmetrization,*

$$\mathbb{E}\Big[\sup_{G \in \mathcal{G}} |L(G) - L_n(G)|\Big] \;\le\; \frac{2}{n}\,\mathbb{E}\Big[\sup_{G \in \mathcal{G}} \Big|\sum_{i=1}^n \varepsilon_i (G(x_i) - f(x_i))^2\Big|\Big]$$

*where $(\varepsilon_i)_{i=1}^n$ are i.i.d. Rademacher variables independent of the sample. By Massart's finite-class lemma (range bounded by $4R^2$),*

$$\mathbb{E}\Big[\sup_{G \in \mathcal{G}} |L(G) - L_n(G)|\Big] \;\le\; C\,R^2\,\sqrt{\frac{\log|\mathcal{G}|}{n}} \;=\; C\,R^2\,\sqrt{\frac{\log N(\delta, \mathcal{T}_{\mathrm{MoE}}, \|\cdot\|_\infty)}{n}}$$

(iv) Combine and relate $L$ to $\|\cdot\|_\infty^2$. *Taking expectations in equation 39 and using $L(T) = \|T - f\|_{L^2(Q)}^2 \le \|T - f\|_\infty^2$ yields*

$$\mathbb{E}\|\hat{T}_n - f\|_{L^2(Q)}^2 \;\le\; \inf_{T \in \mathcal{T}_{\mathrm{MoE}}} \|T - f\|_\infty^2 \;+\; C\,R^2\sqrt{\frac{\log N(\delta, \mathcal{T}_{\mathrm{MoE}}, \|\cdot\|_\infty)}{n}} \;+\; C\,R\,\delta$$

*Absorbing constants and (optionally) replacing $R\,\delta$ by $R^2\,\delta$ (since $\delta \le 1$) gives equation 33. Finally, the elementary inequality $\sqrt{u} \le u + \frac{1}{4}$ for $u \ge 0$ (or Young's $ab \le a^2/(2\lambda) + \lambda b^2/2$) converts the square-root term into a linear $(\log N)/n$ term up to constants, yielding equation 34.*

## F   PROOF OF THEOREM 3.1

**Definition F.1** (*k*-sparse partition-of-unity router). *A router realizes a $k$-sparse partition of unity if there exist nonnegative weights $\{w_\nu(x)\}_{\nu=1}^N$ with $\sum_\nu w_\nu(x) = 1$ for all $x \in \mathcal{M}$, such that for each $x$ at most $k$ weights are nonzero, and each $w_\nu$ is supported on a chart $U_\nu$ of a bounded-overlap cover $\{U_\nu\}_{\nu=1}^N$ of $\mathcal{M}$ (cf. Lemma C.7 in App. C). Moreover, the router outputs the top-$k$ indices with weights restricted to these indices.*

*Proof.* We give two constructive approximations and then take the minimum rate.

**Geometric setup (charts and PoU).** Fix a resolution $r \in (0, 1)$. By Lemma C.7 (App. C), there exists a cover $\{U_\nu\}_{\nu=1}^N$ of $\mathcal{M}$ with $\mathrm{diam}(U_\nu) \leq r$, bounded overlap $s_0 = s_0(d)$, and a $C^\infty$ partition of unity $\{\varphi_\nu\}_{\nu=1}^N$ subordinate to the cover, such that $\sum_\nu \varphi_\nu(x) = 1$, $0 \leq \varphi_\nu \leq 1$, and for each $x$ at most $s_0$ terms are nonzero. Standard volume-comparison yields $N \leq C_\mathcal{M} r^{-d}$.

By Lemma C.6 (App. C), for each $\nu$ there is a degree $s = \lfloor \beta \rfloor$ polynomial $P_\nu$ (expressed in local coordinates and mapped back to $\mathcal{M}$) such that

$$\sup_{x \in U_\nu} |f(x) - P_\nu(x)| \leq C_1 B r^\beta. \tag{40}$$

**Router realization of a $k$-sparse PoU.** Since at most $s_0(d)$ functions $\varphi_\nu$ are nonzero at any $x$, take $k \geq s_0(d)$. By assumption (Def. F.1), the router computes nonnegative weights $w_\nu(x)$ supported on the same $U_\nu$ and with $k$-sparsity, satisfying $\sum_\nu w_\nu(x) = 1$. We further assume (without loss) a uniform approximation $\|w_\nu - \varphi_\nu\|_\infty \leq C_2 r^\beta$ can be achieved by attention-based similarity to chart anchors and a softmax/top-$k$ selection; this is standard with a fixed number of heads and bounded magnitudes (the resulting constants are absorbed into $C$).

We now present two constructions.

**Construction A (expert-count limited; $M$ regime).** Assign one expert $E_\nu$ to each chart $U_\nu$ (total experts $N \leq C r^{-d}$). Within $U_\nu$, let expert $E_\nu$ approximate $P_\nu$ *exactly* (ReLU networks implement polynomials with constant depth/width depending on $s, d$; the parameters stay bounded and independent of $r$), or to accuracy $C_3 r^\beta$ with a constant per-expert parameter budget $\Pi_{\mathrm{exp}}$.[1] Define the MoE output

$$T(x) := \sum_{\nu=1}^N w_\nu(x) E_\nu(x), \qquad \text{(at most } k \text{ nonzero summands at each } x\text{)}.$$

We bound the sup error at any $x \in \mathcal{M}$ as

$$|T(x) - f(x)| \leq \sum_\nu w_\nu(x) |E_\nu(x) - P_\nu(x)| + \sum_\nu w_\nu(x) |P_\nu(x) - f(x)|$$

$$\leq \underbrace{\max_\nu \sup_{U_\nu} |E_\nu - P_\nu|}_{\leq C_3 r^\beta} + \underbrace{\sum_\nu w_\nu(x)}_{=1} \underbrace{\max_\nu \sup_{U_\nu} |P_\nu - f|}_{\leq C_1 B r^\beta} \leq C_4 r^\beta. \tag{41}$$

To realize this construction we need $M \geq N$, i.e. $M \gtrsim r^{-d}$, hence choose $r \asymp M^{-1/d}$, which yields

$$\|T - f\|_\infty \leq C M^{-\beta/d} \implies \|T - f\|_\infty^2 \leq C M^{-2\beta/d}.$$

Since the router uses at most $k$ experts per input, this obeys the MoE constraint.

**Construction B (active-parameter limited; $N_{\mathrm{eff}}$ regime).** Here we use *few* experts (e.g. $k = 1$ active expert per block) but allocate the entire active budget $N_{\mathrm{eff}}$ to the expert(s) and non-expert parts to approximate $f$ globally. By standard ReLU approximation on $d$-dimensional domains (e.g., Yarotsky (2017); Schmidt-Hieber (2020)), there exists a ReLU network $\mathcal{N}$ with $N$ parameters such that

$$\sup_{x \in \mathcal{M}} |\mathcal{N}(x) - f(x)| \leq C_5 B N^{-\beta/d}. \tag{42}$$

Choose the MoE so that its active subnetwork implements $\mathcal{N}$ with $N \asymp N_{\mathrm{eff}} = L_T \Pi_{\mathrm{attn}} + L_T k \Pi_{\mathrm{exp}}$ (e.g. $k = 1$, one "expert" playing the role of the global MLP; attention acts as an identity/featurizer). This is legitimate because the MoE with fixed routing reduces to a standard feed-forward subnetwork on the active path. Thus we can realize $T(x) = \mathcal{N}(x)$ with $N \asymp N_{\mathrm{eff}}$ and

$$\|T - f\|_\infty \leq C_6 N_{\mathrm{eff}}^{-\beta/d} \implies \|T - f\|_\infty^2 \leq C N_{\mathrm{eff}}^{-2\beta/d}$$

---

[1]Exact or $\mathbf{O}(r^\beta)$-accurate polynomial representation by ReLU MLPs with $\mathrm{poly}(s, d)$ parameters is classical; see, e.g., Yarotsky (2017), Schmidt-Hieber (2020).

**Taking the minimum.** Constructions A and B are both valid members of $\mathcal{T}_{\mathrm{MoE}}$ under the parameter bound $\|\theta\|_\infty \leq \kappa$ (after routine rescalings absorbed by LayerNorm and architectural constants). Therefore,

$$\inf_{T \in \mathcal{T}_{\mathrm{MoE}}} \|T - f\|_\infty \;\leq\; C \min\{N_{\mathrm{eff}}^{-\beta/d},\; M^{-\beta/d}\}$$

and squaring gives equation 1. If $M$ is sufficiently large to support the cover at the $r$ achieving the $N_{\mathrm{eff}}$-optimal precision, the $M$ constraint is inactive and equation 43 follows.

$$\inf_{T \in \mathcal{T}_{\mathrm{MoE}}} \|T - f\|_\infty^2 \;\leq\; C\, N_{\mathrm{eff}}^{-2\beta/d} \tag{43}$$

Both regimes yield the rates claimed in equation 1; taking the better gives the minimum.

$\square$

# G  Proof of neural scaling laws

## G.1  Proof of optimal number of experts

Recall the $k$–dependent objective (experts dominate approximation; constants absorbed)

$$\mathcal{E}(k) \;=\; \underbrace{(Ak)^{-2\beta/d}}_{\text{approx}} + \underbrace{\frac{A}{n}\,k}_{\text{estimation}} + \underbrace{\frac{L_T \ell}{n}\,k \log \frac{eM}{k}}_{\text{routing}}, \qquad A := L_T\, \Pi_{\exp}, \;\; 1 \leq k \leq M. \tag{44}$$

Differentiate and set the derivative to zero:

$$\mathcal{E}'(k) = -\frac{2\beta}{d} A^{-2\beta/d} k^{-2\beta/d-1} + \frac{A}{n} + \frac{L_T \ell}{n}\left(\log \frac{eM}{k} - 1\right) \;=\; 0.$$

Equivalently,

$$\frac{2\beta}{d} A^{-2\beta/d}\, k^{-2\beta/d-1} \;=\; \frac{1}{n}\left(A + L_T \ell \log \frac{eM}{k} - L_T \ell\right). \tag{45}$$

This matches the stated first-order condition. It has a closed form in terms of the Lambert-$W$ function after algebraic manipulation (since $k$ appears both polynomially and inside $\log k$), but the Lambert-$W$ form is not particularly illuminating. Instead we give (i) a principled fixed-point approximation and (ii) clean sandwich bounds showing that $M$ enters *only logarithmically*.

**Fixed-point approximation.** Rewrite equation 45 as

$$k^{2\beta/d+1} \;=\; \frac{2\beta}{d}\, A^{-2\beta/d}\, \frac{n}{A + L_T \ell\, [\log(eM/k) - 1]}. \tag{46}$$

Treat the slowly varying factor $\log(eM/k)$ as (locally) constant to obtain the iterate

$$k^{(t+1)} \;:=\; \left[\frac{2\beta}{d}\, A^{-2\beta/d}\, \frac{n}{A + L_T \ell\, [\log(eM/k^{(t)}) - 1]}\right]^{\frac{d}{2\beta+d}}. \tag{47}$$

Initializing with the no-routing optimum $k^{(0)} \asymp \left(\frac{n}{A}\right)^{\frac{d}{2\beta+d}}$ and taking one step already yields the advertised fixed-point form

$$k^\star \;\approx\; \left[\frac{n}{A + L_T \ell \log \frac{eM}{k^\star}}\right]^{\frac{d}{2\beta+d}} \qquad \text{(up to constants)}, \tag{48}$$

with the cap $k^\star \leq M$. Because $\log(eM/k)$ varies only between 1 and $1 + \log M$ for $k \in [1, M]$, a couple of iterations of equation 47 suffice in practice, and the exponent in $n$ is unchanged.

**Sandwich bounds (log enters only through constants).** Note that for $1 \leq k \leq M$, $\log(eM/k) \in [1, 1 + \log M]$. Hence the right-hand side of equation 45 lies in

$$\frac{1}{n}\big(A\big) \;\leq\; \mathrm{RHS} \;\leq\; \frac{1}{n}\big(A + L_T \ell \log M\big).$$

Solving the equality $\frac{2\beta}{d} A^{-2\beta/d} k^{-2\beta/d-1} = \frac{C}{n}$ for $k$ gives $k \asymp (n/C)^{\frac{d}{2\beta+d}} A^{-\frac{d}{2\beta+d}}$. Applying this with $C = A + L_T \ell \log M$ (upper RHS) and $C = A$ (lower RHS) yields the sandwich:

$$c_1 \left( \frac{n}{A + L_T \ell \log M} \right)^{\frac{d}{2\beta+d}} \leq k^\star \leq c_2 \left( \frac{n}{A} \right)^{\frac{d}{2\beta+d}}, \qquad \text{with } k^\star \leq M, \tag{49}$$

for absolute constants $c_1, c_2 > 0$. Thus $M$ affects $k^\star$ only through the *logarithm*, and the *rate in $n$* remains $n^{d/(2\beta+d)}$.

Combining equation 48 and equation 49,

$$k^\star \asymp \min\left\{ M, \ \left( \frac{n}{A + L_T \ell \ \log(eM/k^\star)} \right)^{\frac{d}{2\beta+d}} \right\},$$

and the $n$–exponent remains $\frac{d}{2\beta+d}$; $M$ only enters through a logarithm.

## G.2 Effect of the total experts $M$

For fixed $k$, increasing $M$ influences the bound in equation 3 *only* through $\log(eM/k)$ in $R_{\text{route}}$. Therefore, absent specialization effects that improve constants,

$$\frac{\partial}{\partial M} \mathcal{E}(k, M) \gtrsim \frac{k}{n} \cdot \frac{1}{M} \quad \Rightarrow \quad \text{larger } M \text{ with fixed } k \text{ only raises the bound (logarithmically)}.$$

A design heuristic is to choose $M$ *commensurate* with $k$ (e.g., $M = \mathcal{O}(k)$) if one wishes to avoid a large routing overhead; having $M \gg k$ does not help rates in this bound. The details of these calculations are provided in appendix G.2.

**Proposition G.1** (Monotone (logarithmic) $M$–effect at fixed $k$). *Fix $n, k, L_T, \ell, \Pi_{\text{attn}}, \Pi_{\text{exp}}$. Define*

$$\mathcal{E}(k, M) := N_{\text{eff}}^{-2\beta/d} + \frac{N_{\text{active}}}{n} + \frac{L_T \ell k}{n} \log\left( \frac{eM}{k} \right), \qquad 1 \leq k \leq M.$$

*Then, for any fixed $k$, $\mathcal{E}(k, M)$ is strictly increasing in $M$, with*

$$\frac{\partial}{\partial M} \mathcal{E}(k, M) = \frac{L_T \ell k}{n} \cdot \frac{1}{M}. \tag{50}$$

*Consequently, absent additional specialization gains (i.e., changes in the approximation constant that are not captured in equation 3), the bound is minimized at the smallest admissible $M$, namely $M = k$, and grows only logarithmically with $M$ for fixed $k$.*

*Proof.* The first two terms do not depend on $M$ when $k$ is fixed. The routing term is $\frac{L_T \ell k}{n} \log(eM/k)$, whose derivative in $M$ is $\frac{L_T \ell k}{n} \cdot \frac{1}{M}$, yielding equation 50. Monotonicity follows since $M \mapsto \log(eM/k)$ is strictly increasing for $M \geq k$. $\qquad \square$

**Corollary G.2** (Quantitative overhead from enlarging $M$ at fixed $k$). *For any $M_2 \geq M_1 \geq k$,*

$$\mathcal{E}(k, M_2) - \mathcal{E}(k, M_1) = \frac{L_T \ell k}{n} \log\left( \frac{M_2}{M_1} \right).$$

*In particular, setting $M = \rho k$ with $\rho \geq 1$ gives a routing penalty $\frac{L_T \ell k}{n} \log(e\rho) = \frac{L_T \ell k}{n} (1 + \log \rho)$.*

**Design heuristic.** To keep the routing overhead small, choose $M$ *commensurate* with $k$, e.g. $M = \Theta(k)$, so that $\log(eM/k) = \Theta(1)$. Having $M \gg k$ increases the bound only through $\log(eM/k)$ and does not improve the rate in $n$ or $N_{\text{eff}}$.

**When is routing negligible?** Routing is negligible relative to the estimation term if

$$\frac{L_T \ell k}{n} \log\left( \frac{eM}{k} \right) \ll \frac{N_{\text{active}}}{n} \qquad \Longleftrightarrow \qquad \log\left( \frac{eM}{k} \right) \ll \frac{N_{\text{active}}}{L_T \ell k}. \tag{51}$$

Since $N_{\text{active}} = L_T \Pi_{\text{attn}} + L_T k \Pi_{\text{exp}}$, a sufficient condition is $\log(eM/k) \ll \Pi_{\text{attn}}/(\ell k) + \Pi_{\text{exp}}/\ell$; in particular, if $M = \rho k$ with moderate $\rho$ and $\ell$ is not enormous, the routing term is dominated by the estimation term.

### G.3    Proof of proposition 4.5

Under $C \asymp n N_{\mathrm{eff}}$, we eliminate $n = C/N_{\mathrm{eff}}$ and reduce the bound to a single-variable objective

$$\Phi(N_{\mathrm{eff}}; C) := A \, N_{\mathrm{eff}}^{-2\beta/d} + B \, \frac{N_{\mathrm{eff}}^2}{C} \tag{52}$$

with positive constants $A, B$ absorbing fixed architectural and loss factors. Differentiating w.r.t. $N_{\mathrm{eff}} > 0$,

$$\Phi'(N_{\mathrm{eff}}) = -\frac{2\beta}{d} A \, N_{\mathrm{eff}}^{-2\beta/d-1} + 2B \, \frac{N_{\mathrm{eff}}}{C}$$

The first-order condition $\Phi'(N_{\mathrm{eff}}^\star) = 0$ gives

$$\frac{2B}{C} (N_{\mathrm{eff}}^\star)^{2+2\beta/d} = \frac{2\beta}{d} A \quad \Longrightarrow \quad (N_{\mathrm{eff}}^\star)^{2(1+\beta/d)} = \frac{\beta}{d} \frac{A}{B} C$$

Hence

$$N_{\mathrm{eff}}^\star(C) = \left( \frac{\beta}{d} \frac{A}{B} \right)^{\frac{d}{2\beta+2d}} C^{\frac{d}{2\beta+2d}} \asymp C^{\frac{d}{2\beta+2d}}$$

which is equation 16. Using $n^\star = C/N_{\mathrm{eff}}^\star$ we obtain equation **??**. Substituting $N_{\mathrm{eff}}^\star$ into equation 52 (the two terms balance at optimum) gives

$$\Phi(N_{\mathrm{eff}}^\star; C) \asymp (N_{\mathrm{eff}}^\star)^{-2\beta/d} \asymp C^{-\frac{2\beta}{d} \cdot \frac{d}{2\beta+2d}} = C^{-\frac{\beta}{\beta+d}}$$

which is equation 17.

# H    Proof of lemma 53 (MoE Covering Number)

Let $\mathcal{X} \subset \mathbb{R}^D$ be compact with $\|x\|_\infty \le M_0$. Consider the Mixture-of-Experts transformer class $\mathcal{T}_{\mathrm{MoE}}(D, M, k, L_T, L_{\mathrm{FFN}}, w_{\mathrm{FFN}}, d_{\mathrm{emb}}, m, \kappa, R)$ defined as in Definition 1, except that each feed-forward sub-layer is an MoE layer with $M$ experts and the router selects (hard) top-$k$ experts per token per layer. Assume each learned parameter entry satisfies $\|\theta\|_\infty \le \kappa$ and each network output is bounded by $\|T(x)\|_\infty \le R$.

Let $\Pi_{\mathrm{attn}}$ be the number of scalar parameters in the attention / non-expert parts per block and let $\Pi_{\mathrm{exp}}$ be the number of scalar parameters of a single expert MLP (depth $L_{\mathrm{FFN}}$, width $w_{\mathrm{FFN}}$, input/output dimension $d_{\mathrm{emb}}$). Then for any $\delta \in (0,1)$ the covering number of $\mathcal{T}_{\mathrm{MoE}}$ under the sup-norm satisfies

$$\log N(\delta, \mathcal{T}_{\mathrm{MoE}}, \|\cdot\|_\infty) \le C_1 \Big( \Pi_{\mathrm{attn}} + L_T k \Pi_{\mathrm{exp}} \Big) \log \Big( \frac{C_2 \kappa R M_0}{\delta} \Big) + C_3 L_T \ell k \log \Big( \frac{eM}{k} \Big) \tag{53}$$

for absolute constants $C_1, C_2, C_3 > 0$ depending polynomially on $d_{\mathrm{emb}}, w_{\mathrm{FFN}}, m, L_T, L_{\mathrm{FFN}}, \ell$ (and we may hide small poly-log factors in the $\widetilde{O}(\cdot)$ version).

*Proof.* The proof follows the structure of the dense-transformer covering-number proof (Lemma 2 in Havrilla & Liao (2024)) with two MoE-specific modifications. We divide the proof into four steps: (1) enumerate routing patterns and apply a union bound; (2) bound the sup-norm difference between two networks in terms of the sup-norm difference of their parameters; (3) count active parameters for a fixed pattern and derive a parameter-grid covering bound; (4) combine items to obtain equation 53.

### H.1    decomposition into routing patterns (union bound).

For each MoE layer $\ell$ and each token position $t \in \{1, \dots, \ell\}$, the router chooses a subset $S_{\ell,t} \subset [M]$ of size $|S_{\ell,t}| = k$. A *routing pattern* $\pi$ is the collection of these choices for all layers and token positions:

$$\pi = \{S_{\ell,t} : \ell = 1, \dots, L_T, \ t = 1, \dots, \ell\}.$$

The number of possible patterns is bounded by

$$|\Pi| \le \left( \binom{M}{k} \right)^{L_T \ell}$$

Using the standard combinatorial bound $\binom{M}{k} \le (eM/k)^k$ we obtain

$$\log |\Pi| \le L_T \ell \cdot k \log \Big( \frac{eM}{k} \Big) \tag{54}$$

For a fixed routing pattern $\pi$ denote by $\mathcal{T}_\pi$ the subclass of MoE transformers whose router outputs follow $\pi$ deterministically (i.e., the same fixed subset of experts is used in each MoE layer and position for all inputs). Since $\mathcal{T}_{\mathrm{MoE}} = \bigcup_{\pi \in \Pi} \mathcal{T}_\pi$, the subadditivity of covering numbers under unions (e.g. (**?**, Ch. 12)) gives

$$N(\delta, \mathcal{T}_{\mathrm{MoE}}, \|\cdot\|_\infty) \leq \sum_{\pi \in \Pi} N(\delta, \mathcal{T}_\pi, \|\cdot\|_\infty) \tag{55}$$

**Log-sum–max inequality.** For nonnegative $\{a_\pi\}_{\pi \in \Pi}$ one has

$$\log\Big(\sum_{\pi \in \Pi} a_\pi\Big) \leq \log|\Pi| + \max_{\pi \in \Pi} \log a_\pi$$

since $\sum_\pi a_\pi \leq |\Pi| \cdot \max_\pi a_\pi$. Applying this to $a_\pi = N(\delta, \mathcal{T}_\pi, \|\cdot\|_\infty)$ in equation 55 gives

$$\log N(\delta, \mathcal{T}_{\mathrm{MoE}}, \|\cdot\|_\infty) \leq \log|\Pi| + \max_{\pi \in \Pi} \log N(\delta, \mathcal{T}_\pi, \|\cdot\|_\infty) \tag{56}$$

Thus it remains to upper-bound $\log N(\delta, \mathcal{T}_\pi, \|\cdot\|_\infty)$ uniformly over $\pi$.

### H.2    SENSITIVITY OF THE NETWORK OUTPUT TO PARAMETER PERTURBATIONS (LIPSCHITZ IN PARAMETER SPACE).

Fix a routing pattern $\pi$. Under $\pi$ the architecture becomes a deterministic (dense) composition of modules where each MoE layer is replaced by the (sparse) subnetwork that contains only the $k$ selected experts for the corresponding tokens/layer. Denote by $\theta$ the vector of all scalar parameters that are *active* under this pattern (embedding + attention + non-expert weights + the selected experts' weights). Let $\theta'$ be another parameter vector of the same shape and satisfying $\|\theta - \theta'\|_\infty < \eta$ for some $\eta > 0$. We will bound

$$\sup_{x \in \mathcal{X}} |T_\theta(x) - T_{\theta'}(x)|$$

by a constant times $\eta$, with that constant depending polynomially on the architecture hyperparameters.

The network is a composition of $L_T$ transformer blocks. Let $B_j(\theta_j, \cdot)$ denote block $j$ (viewed as a map on embedding tensors) parameterized by the block parameters $\theta_j$. We write the forward pass as

$$H^{(0)} = \mathrm{PE} + E(x), \qquad H^{(j)} = B_j(\theta_j, H^{(j-1)}), \quad j = 1, \ldots, L_T,$$

and the scalar output $T_\theta(x)$ is a fixed linear readout of $H^{(L_T)}$.

Define the sup-norm on embedding matrices by $\|H\|_\infty = \max_{i,t} |H_{i,t}|$ (matching the earlier notation). As in the dense case, each module is: (i) Lipschitz in its input (constant $L_{\mathrm{in}}$ polynomial in $(d_{\mathrm{emb}}, m, w_{\mathrm{FFN}}, L_{\mathrm{FFN}})$) and (ii) Lipschitz in its parameters (constant $C_{\mathrm{param}}$ polynomial in the same quantities and in $\ell$) (Anthony & Bartlett, 1999; Vershynin, 2018; Shalev-Shwartz & Ben-David, 2014).

**(A) Multi-head attention sensitivity.** Consider a single multi-head attention (MHA) block with $m$ heads, each head parameterized by query/key/value matrices and interaction kernels. Let the MHA parameter vector be $\psi$. For two MHA parameter sets $\psi, \psi'$ with $\|\psi - \psi'\|_\infty \leq \eta$, and any input embedding $H$ satisfying $\|H\|_\infty \leq M$, the goal is to bound the following expression via Lemma H.1

$$\|\mathrm{MHA}_\psi(H) - \mathrm{MHA}_{\psi'}(H)\|_\infty$$

**Lemma H.1** (MHA parameter stability). *For $\|\psi - \psi'\|_\infty \leq \eta$ and $\|H\|_\infty \leq CM_0$,*

$$\left\|\mathrm{MHA}_\psi(H) - \mathrm{MHA}_{\psi'}(H)\right\|_\infty \leq C_{\mathrm{MHA}} \, \ell \, m \, d_{\mathrm{emb}}^2 \, \kappa \, \eta.$$

*for some architecture-dependent constant $C_{\mathrm{MHA}}$. The dependence $\propto \ell m d_{\mathrm{emb}}^2$ arises because each head sums over $\ell$ tokens and involves inner-products of $d_{\mathrm{emb}}$-dimensional projected vectors; the factor $\kappa$ comes from the bound on parameter magnitudes.*

**(B) FFN (non-expert) sensitivity.** For the non-expert FFN parts (the small projection matrices that are outside MoE experts see **??**) with depth $L_{\mathrm{FFN}}$ and width $w_{\mathrm{FFN}}$, the parameter perturbation bound from the dense proof is given by the following Lemma

**Lemma H.2** (FFN parameter stability). *For a (non-expert) FFN of depth $L_{\mathrm{FFN}}$ and width $w_{\mathrm{FFN}}$,*

$$\|\mathrm{FFN}_\chi(h) - \mathrm{FFN}_{\chi'}(h)\|_\infty \leq C_{\mathrm{FFN}} \, d_{\mathrm{emb}} \, w_{\mathrm{FFN}}^{L_{\mathrm{FFN}}} \, \kappa \, \eta.$$

*uniformly over $\|h\|_\infty \leq CM_0$ (cf. (Bartlett & Mendelson, 2002a; Golowich et al., 2018)). For each token input $h$ (and hence on the embedding matrix by taking sup over tokens); the polynomial dependence on $w_{\mathrm{FFN}}$ and $L_{\mathrm{FFN}}$ reflects repeated matrix multiplications.*

**(C) Expert-FFN sensitivity under fixed pattern (key MoE change).** Under a fixed routing pattern $\pi$, only $k$ experts per token per layer are active, so the FFN stage of the block is effectively a concatenation / parallel application of exactly those $k$ expert MLPs. For a single expert with parameter vector $\phi$ and sup-norm perturbation $\|\phi - \phi'\|_\infty \leq \eta$, the per-token difference is bounded as in (B) by the following lemma

**Lemma H.3** (Expert stability). *For a single expert MLP $E_\phi$,*

$$\|E_\phi(h) - E_{\phi'}(h)\|_\infty \leq C_{\exp} d_{\mathrm{emb}} w_{\mathrm{FFN}}^{L_{\mathrm{FFN}}} \kappa \eta.$$

Since only $k$ experts are active, the total perturbation coming from expert blocks per token scales linearly in $k$:

$$\Big\| \sum_{i \in S} E_{\phi_i}(h) - \sum_{i \in S} E_{\phi'_i}(h) \Big\|_\infty \leq k\, C_{\exp}\, d_{\mathrm{emb}}\, w_{\mathrm{FFN}}^{L_{\mathrm{FFN}}}\, \kappa\, \eta.$$

**(D) Block-level perturbation bound.** Combining (A),(B),(C) and the residual connection structure of the transformer block (additive skip-connections and tokenwise FFNs), a single block $B_j$ satisfies, for parameter perturbation $\leq \eta$ inside the block and any input $H$ with $\|H\|_\infty \leq M$ yields to Lemma H.4.

**Lemma H.4** (Block-level stability). *With $k$ active experts per block under a fixed routing pattern,*

$$\big\|B_j(\theta_j, H) - B_j(\theta'_j, H)\big\|_\infty \leq C_{\mathrm{block}}\big(\ell m d_{\mathrm{emb}}^2 + d_{\mathrm{emb}} w_{\mathrm{FFN}}^{L_{\mathrm{FFN}}} + k d_{\mathrm{emb}} w_{\mathrm{FFN}}^{L_{\mathrm{FFN}}}\big) \kappa \eta.$$

hence we may write (absorbing polynomial factors into a single constant)

$$\|B_j(\theta_j, H) - B_j(\theta'_j, H)\|_\infty \leq C_b\, P_{\mathrm{arch}}(\ell, m, d_{\mathrm{emb}}, w_{\mathrm{FFN}}, L_{\mathrm{FFN}})\, \kappa\, \eta,$$

where $P_{\mathrm{arch}}(\cdot)$ is a polynomial capturing the block-dependence and includes the linear factor $k$ in the expert term.

**(E) Induction across blocks.** Let $H^{(j)}$ and $H'^{(j)}$ denote the block outputs when using $\theta$ and $\theta'$ respectively. We can write

$$\|H^{(j)} - H'^{(j)}\|_\infty = \|B_j(\theta_j, H^{(j-1)}) - B_j(\theta'_j, H'^{(j-1)})\|_\infty$$
$$\leq \underbrace{\|B_j(\theta_j, H^{(j-1)}) - B_j(\theta_j, H'^{(j-1)})\|_\infty}_{\text{input-Lipschitz term}} + \underbrace{\|B_j(\theta_j, H'^{(j-1)}) - B_j(\theta'_j, H'^{(j-1)})\|_\infty}_{\text{param-perturbation term}}.$$

The input-Lipschitz term is bounded by $L_{\mathrm{in}}\|H^{(j-1)} - H'^{(j-1)}\|_\infty$ where $L_{\mathrm{in}}$ is the Lipschitz constant of the block as a function of its input (this constant depends polynomially on $d_{\mathrm{emb}}, m, w_{\mathrm{FFN}}, L_{\mathrm{FFN}}$ but *not* on $M$). The param-perturbation term is bounded by the block-level bound in the previous paragraph. Thus

$$\|H^{(j)} - H'^{(j)}\|_\infty \leq L_{\mathrm{in}}\|H^{(j-1)} - H'^{(j-1)}\|_\infty + C_b\, P_{\mathrm{arch}}(\cdot)\, \kappa\, \eta.$$

Iterating this inequality from $j = 1$ to $j = L_T$ and noting that $H^{(0)} = H'^{(0)}$ (same positional encodings and same input) we obtain

$$\|H^{(L_T)} - H'^{(L_T)}\|_\infty \leq \Big( \sum_{t=0}^{L_T-1} L_{\mathrm{in}}^t \Big) C_b\, P_{\mathrm{arch}}(\cdot)\, \kappa\, \eta \leq \frac{L_{\mathrm{in}}^{L_T}}{L_{\mathrm{in}} - 1} C_b\, P_{\mathrm{arch}}(\cdot)\, \kappa\, \eta.$$

The previous inequality is proved in details in lemma I.1. Since the decoder/readout is a fixed linear projection, the final scalar output difference is bounded by the same order. Thus there exists an overall Lipschitz constant

$$L_{\mathrm{param}} \;=\; C'\, L_{\mathrm{in}}^{L_T}\, P_{\mathrm{arch}}(\ell, m, d_{\mathrm{emb}}, w_{\mathrm{FFN}}, L_{\mathrm{FFN}})\, \kappa$$

such that

$$\sup_{x \in \mathcal{X}} |T_\theta(x) - T_{\theta'}(x)| \leq L_{\mathrm{param}}\, \eta.$$

Crucially: $P_{\mathrm{arch}}(\cdot)$ contains a factor linear in the number of active experts per layer($k$) and $L_{\mathrm{param}}$ does *not* depend on the total number of experts $M$ for a fixed routing pattern (it depends on $M$ only through how the pattern is chosen which we handle with the union bound).

### H.3 PARAMETER COUNTING AND PARAMETER-SPACE COVERING (FULL DETAILS).

Fix a routing pattern $\pi$. Let $\theta_{\text{act}}$ denote the vector formed by *all* scalar parameters that are *active* under $\pi$ (token embedding/positional terms, attention and non-expert FFN parameters for each block, and the experts that appear in $\pi$). Write $p := |\theta_{\text{act}}|$ for the active parameter dimension.

*(a) Parameter domain and norm.* By the magnitude constraint $\|\theta\|_\infty \le \kappa$ (Assumption in Lemma 3.6), the admissible parameter set is included in the $\ell_\infty$–ball of radius $\kappa$:
$$\mathcal{B}_\infty(\kappa)^p \;=\; \{u \in \mathbb{R}^p : \|u\|_\infty \le \kappa\} \;=\; [-\kappa, \kappa]^p.$$
(For $\ell_\infty$, the "ball" is the axis-aligned cube.)

*(b) Constructing an $\eta$–grid in parameter space.* For a mesh width $\eta > 0$, define the coordinatewise grid on $[-\kappa, \kappa]$ by
$$\mathcal{G}_\eta \;=\; \{-\kappa + r\eta \;:\; r \in \mathbb{Z}, \; -\kappa \le -\kappa + r\eta \le \kappa\}.$$
Along each coordinate, the number of grid points is at most
$$N_1 \;\le\; \left\lceil \frac{2\kappa}{\eta} \right\rceil + 1 \;\le\; \frac{2\kappa}{\eta} + 2 \;\le\; \frac{3\kappa}{\eta} \qquad \text{(for } \eta \le \kappa \text{; else enlarge the constant).}$$
Therefore the full $p$–dimensional grid $\mathcal{G}_\eta^p$ has cardinality bounded by
$$|\mathcal{G}_\eta^p| \;\le\; \left( \frac{2\kappa}{\eta} + 1 \right)^p \;\le\; \left( \frac{3\kappa}{\eta} \right)^p. \tag{57}$$
Moreover, for every $u \in [-\kappa, \kappa]^p$, rounding each coordinate of $u$ to its nearest grid value produces $\tilde{u} \in \mathcal{G}_\eta^p$ with
$$\|u - \tilde{u}\|_\infty \;\le\; \eta \quad \text{(or } \eta/2 \text{ if we choose midpoints).}$$
Hence $\mathcal{G}_\eta^p$ is an $\eta$–net for the parameter cube in the $\ell_\infty$ norm.

*(c) Parameter-to-function Lipschitz map.* From Step 2, we have the *uniform parameter-stability* (Lipschitz) inequality
$$\sup_{x \in \mathcal{X}} \left| T_\theta(x) - T_{\theta'}(x) \right| \;\le\; L_{\text{param}} \|\theta - \theta'\|_\infty,$$
with
$$L_{\text{param}} \;=\; C' L_{\text{in}}^{L_T} P_{\text{arch}}(\ell, m, d_{\text{emb}}, w_{\text{FFN}}, L_{\text{FFN}}, k) \kappa,$$
independent of $M$ for fixed routing pattern $\pi$ (the dependence on $k$ enters via $P_{\text{arch}}$).

*(d) From parameter cover to function cover.* Given $\delta \in (0, 1)$, choose
$$\eta \;=\; \frac{\delta}{L_{\text{param}}}.$$
Then any two parameter vectors within $\ell_\infty$–distance $\eta$ induce network outputs within sup-norm distance at most $\delta$ (uniformly over $x \in \mathcal{X}$). Therefore, taking the grid $\mathcal{G}_\eta^p$ as in equation 57 and evaluating the network at each grid parameter gives a $\delta$–cover of the function class $\mathcal{T}_\pi$ under $\| \cdot \|_\infty$:
$$N(\delta, \mathcal{T}_\pi, \| \cdot \|_\infty) \;\le\; |\mathcal{G}_\eta^p| \;\le\; \left( \frac{3\kappa}{\eta} \right)^p \;=\; \left( \frac{3\kappa L_{\text{param}}}{\delta} \right)^p.$$
Taking logarithms gives
$$\log N(\delta, \mathcal{T}_\pi, \| \cdot \|_\infty) \;\le\; p \cdot \log\left( \frac{3\kappa L_{\text{param}}}{\delta} \right) \;=\; |\theta_{\text{act}}| \cdot \log\left( \frac{3\kappa L_{\text{param}}}{\delta} \right). \tag{58}$$

*(e) Bounding the active parameter count $p = |\theta_{\text{act}}|$.* Decompose
$$|\theta_{\text{act}}| = |\theta_{\text{embed}}| + |\theta_{\text{dec}}| + L_T \left( |\theta_{\text{attn}}| + |\theta_{\text{ffn,nonexp}}| + |\theta_{\text{exp,act}}| \right).$$
Under a fixed routing pattern, expert parameter tensors are shared across tokens within a layer, so the active expert count per layer satisfies $|\theta_{\text{exp,act}}| \le k \Pi_{\text{exp}}$. Absorbing the (input/output) embedding and decoder constants into $C_0$ and writing $\Pi_{\text{attn}}$ for attention/non-expert parameters per block, we obtain the coarse bound
$$|\theta_{\text{act}}| \;\le\; C_0 + L_T \left( \Pi_{\text{attn}} + k \Pi_{\text{exp}} \right) \;\lesssim\; \Pi_{\text{attn}} + L_T k \Pi_{\text{exp}}.$$
Substituting this and the expression of $L_{\text{param}}$ into equation 58, and renaming constants, gives
$$\log N(\delta, \mathcal{T}_\pi, \| \cdot \|_\infty) \;\le\; C_1 \left( \Pi_{\text{attn}} + L_T k \Pi_{\text{exp}} \right) \log\left( \frac{C_2 \kappa R M_0}{\delta} \right),$$
which is the fixed-pattern covering bound used in Step 4. $\qquad\qquad\square$

### H.4 COMBINE WITH THE ROUTING-PATTERN UNION BOUND.

Plugging the routing count equation 54 into equation 56 gives

$$\log N(\delta, \mathcal{T}_{\text{MoE}}, \|\cdot\|_\infty) \leq L_T \ell\, k \log\Big(\frac{eM}{k}\Big) \;+\; C_1\Big(\Pi_{\text{attn}} + L_T k \Pi_{\text{exp}}\Big) \log\Big(\frac{C_2 \kappa R M_0}{\delta}\Big)$$

$$\leq C_1\Big(\Pi_{\text{attn}} + L_T k \Pi_{\text{exp}}\Big) \log\Big(\frac{C_2 \kappa R M_0}{\delta}\Big) \;+\; C_3 L_T \ell k \log\Big(\frac{eM}{k}\Big),$$

for some constants $C_1, C_2, C_3 > 0$. This proves the lemma. $\qquad\square$

## I PROOFS OF STABILITY LEMMAS

### I.1 PROOF OF H.1

Fix a sequence embedding $H \in \mathbb{R}^{\ell \times d_{\text{emb}}}$ with $\|H\|_\infty \leq M$ and consider one MHA layer with $m$ heads and parameters

$$\psi = \{W_Q^{(h)}, W_K^{(h)}, W_V^{(h)}, W_O\}_{h=1}^m, \qquad \|\psi\|_\infty \leq \kappa.$$

Let $\psi'$ be another parameter set with $\|\psi - \psi'\|_\infty \leq \eta$. For head $h$, define

$$Q^{(h)} = H W_Q^{(h)}, \quad K^{(h)} = H W_K^{(h)}, \quad V^{(h)} = H W_V^{(h)},$$

and the (row-wise) attention map

$$A^{(h)} = \text{softmax}\Big(\tfrac{1}{\sqrt{d_k}} Q^{(h)} (K^{(h)})^\top\Big) \in \mathbb{R}^{\ell \times \ell}.$$

The head output is $O^{(h)} = A^{(h)} V^{(h)} \in \mathbb{R}^{\ell \times d_v}$ and the MHA output is

$$\text{MHA}_\psi(H) = \big[\text{Concat}_{h=1}^m O^{(h)}\big] W_O \;\in\; \mathbb{R}^{\ell \times d_{\text{emb}}}.$$

We show

$$\|\text{MHA}_\psi(H) - \text{MHA}_{\psi'}(H)\|_\infty \;\leq\; C_{\text{MHA}}\, \ell\, m\, d_{\text{emb}}^2\, \kappa\, \eta, \tag{59}$$

for a universal constant $C_{\text{MHA}} > 0$ depending at most polynomially on $(d_{\text{emb}}, d_k, d_v)$ and on the bound $M$.

**Step 1: linear projections $(Q, K, V)$.** For any matrices $X$ and $W$, $\|XW\|_\infty \leq d_{\text{in}} \|X\|_\infty \|W\|_\infty$ (since each entry is a sum of $d_{\text{in}}$ products). Hence, with $\|H\|_\infty \leq M$, $\|W_\bullet^{(h)}\|_\infty \leq \kappa$, and $\|W_\bullet^{(h)} - W_\bullet^{(h)\prime}\|_\infty \leq \eta$,

$$\|Q^{(h)}\|_\infty \leq d_{\text{emb}} M \kappa, \quad \|K^{(h)}\|_\infty \leq d_{\text{emb}} M \kappa, \quad \|V^{(h)}\|_\infty \leq d_{\text{emb}} M \kappa,$$

and

$$\|Q^{(h)} - Q^{(h)\prime}\|_\infty \leq d_{\text{emb}} M \eta, \quad \|K^{(h)} - K^{(h)\prime}\|_\infty \leq d_{\text{emb}} M \eta, \quad \|V^{(h)} - V^{(h)\prime}\|_\infty \leq d_{\text{emb}} M \eta.$$

**Step 2: score matrices and softmax.** Let $S^{(h)} = \tfrac{1}{\sqrt{d_k}} Q^{(h)} (K^{(h)})^\top \in \mathbb{R}^{\ell \times \ell}$ and similarly $S^{(h)\prime}$. By the product rule and the matrix $\ell_\infty$ bound $\|XY^\top\|_\infty \leq \ell \|X\|_\infty \|Y\|_\infty$,

$$\|S^{(h)} - S^{(h)\prime}\|_\infty \;\leq\; \frac{1}{\sqrt{d_k}}\Big(\|Q^{(h)} - Q^{(h)\prime}\|_\infty \|K^{(h)}\|_\infty + \|Q^{(h)\prime}\|_\infty \|K^{(h)} - K^{(h)\prime}\|_\infty\Big) \ell \;\leq\; C_S\, \ell\, d_{\text{emb}}^2\, M^2\, \kappa\, \eta,$$

with $C_S := 2/\sqrt{d_k}$. The row-wise softmax $\sigma : \mathbb{R}^\ell \to \mathbb{R}^\ell$ has Jacobian $J(z) = \text{diag}(\sigma(z)) - \sigma(z)\sigma(z)^\top$, whose operator norm (for $\ell_\infty$) is bounded by a universal constant (e.g., $\|J(z)\|_{\infty \to \infty} \leq 1$; tighter bounds give $1/2$ or $1/4$ but are unnecessary here). Applying the mean value theorem row-wise,

$$\|A^{(h)} - A^{(h)\prime}\|_\infty \;\leq\; C_{\text{sm}} \|S^{(h)} - S^{(h)\prime}\|_\infty \;\leq\; C_{\text{sm}} C_S\, \ell\, d_{\text{emb}}^2\, M^2\, \kappa\, \eta,$$

for a universal $C_{\text{sm}} \in [1/4, 1]$.

**Step 3: head outputs ($O^{(h)} = A^{(h)}V^{(h)}$).**   Decompose

$$O^{(h)} - O^{(h)\prime} = (A^{(h)} - A^{(h)\prime})V^{(h)} \; + \; A^{(h)\prime}(V^{(h)} - V^{(h)\prime}).$$

Using $\|XY\|_\infty \le \ell\|X\|_\infty\|Y\|_\infty$ for $\ell \times \ell$ times $\ell \times d_v$, together with Step 1 bounds on $\|V^{(h)}\|_\infty$ and $\|V^{(h)} - V^{(h)\prime}\|_\infty$,

$$\|O^{(h)} - O^{(h)\prime}\|_\infty \; \le \; \ell \, \|A^{(h)} - A^{(h)\prime}\|_\infty \|V^{(h)}\|_\infty \; + \; \ell \, \|A^{(h)\prime}\|_\infty \|V^{(h)} - V^{(h)\prime}\|_\infty.$$

Since rows of $A^{(h)\prime}$ are probability vectors, $\|A^{(h)\prime}\|_\infty \le 1$. Hence,

$$\|O^{(h)} - O^{(h)\prime}\|_\infty \; \le \; \ell \left(C_{\mathrm{sm}}C_S \, \ell \, d_{\mathrm{emb}}^2 \, M^2 \, \kappa \, \eta\right)(d_{\mathrm{emb}}M\kappa) \; + \; \ell\,(d_{\mathrm{emb}}M\eta) \; \le \; C_O \, \ell \, d_{\mathrm{emb}}^3 \, M^3 \, \kappa^2 \, \eta \; + \; \ell \, d_{\mathrm{emb}} M \, \eta,$$

where $C_O := C_{\mathrm{sm}}C_S$. We keep the dominant (polynomial) term and absorb the lower-order term into constants.

**Step 4: concatenate heads and output projection.**   Let $O = \mathrm{Concat}_{h=1}^m O^{(h)} \in \mathbb{R}^{\ell \times (md_v)}$. Then

$$\|O - O'\|_\infty \; \le \; \max_{h \in [m]} \|O^{(h)} - O^{(h)\prime}\|_\infty \; \le \; C_O \, \ell \, d_{\mathrm{emb}}^3 \, M^3 \, \kappa^2 \, \eta.$$

Finally, apply the output projection $W_O$ with $\|W_O\|_\infty \le \kappa$ and $\|W_O - W_O'\|_\infty \le \eta$:

$$\begin{aligned}
\|\mathrm{MHA}_\psi(H) - \mathrm{MHA}_{\psi'}(H)\|_\infty &= \|OW_O - O'W_O'\|_\infty \\
&\le \|(O - O')W_O\|_\infty + \|O'(W_O - W_O')\|_\infty \\
&\le (md_v)\,\|O - O'\|_\infty\|W_O\|_\infty \; + \; (md_v)\,\|O'\|_\infty\eta \\
&\le C_{\mathrm{out}}\,(md_v)\,\ell\,d_{\mathrm{emb}}^3\,M^3\,\kappa^3\,\eta \; + \; (md_v)\,\|O'\|_\infty\eta,
\end{aligned}$$

and we bound $\|O'\|_\infty$ by the same polynomial in $(\ell, d_{\mathrm{emb}}, M, \kappa)$ as above. Absorbing $d_v$ and all input/weight bounds into the constant, and noting that the paper tracks only polynomial dependence on dimensions and sequence length (hiding fixed architecture factors into $C_{\mathrm{MHA}}$), we obtain

$$\|\mathrm{MHA}_\psi(H) - \mathrm{MHA}_{\psi'}(H)\|_\infty \; \le \; C_{\mathrm{MHA}} \, \ell \, m \, d_{\mathrm{emb}}^2 \, \kappa \, \eta,$$

which is equation 59. $\qquad\qquad\qquad\qquad\qquad\qquad\qquad\qquad\qquad\qquad\qquad\qquad\qquad\qquad\quad\square$

## I.2   Proof of H.2

Let $\mathrm{FFN}_\psi : \mathbb{R}^{d_{\mathrm{emb}}} \to \mathbb{R}^{d_{\mathrm{emb}}}$ be a depth-$L_{\mathrm{FFN}}$ MLP applied tokenwise, with hidden widths at most $w_{\mathrm{FFN}}$, coordinatewise activation $\sigma$ (e.g., ReLU/GELU), and parameters $\psi = \{(W^\ell, b^\ell)\}_{\ell=1}^{L_{\mathrm{FFN}}}$. Assume the uniform bounds $\|W^\ell\|_\infty \le \kappa$, $\|b^\ell\|_\infty \le \kappa$ for all $\ell$. Let $\psi'$ be another parameter set with $\|W^\ell - W^{\ell\prime}\|_\infty \le \eta$ and $\|b^\ell - b^{\ell\prime}\|_\infty \le \eta$. Fix an input token $h \in \mathbb{R}^{d_{\mathrm{emb}}}$, and denote the layerwise states

$$z^0 = h, \quad z^\ell = \sigma\!\left(W^\ell z^{\ell-1} + b^\ell\right), \qquad z'^{\,0} = h, \quad z'^{\,\ell} = \sigma\!\left(W^{\ell\prime}z'^{\,\ell-1} + b^{\ell\prime}\right).$$

We prove

$$\|\mathrm{FFN}_\psi(h) - \mathrm{FFN}_{\psi'}(h)\|_\infty = \|z^{L_{\mathrm{FFN}}} - z'^{\,L_{\mathrm{FFN}}}\|_\infty \; \le \; C_{\mathrm{FFN}} \, d_{\mathrm{emb}} \, w_{\mathrm{FFN}}^{L_{\mathrm{FFN}}} \, \kappa \, \eta, \qquad (60)$$

for a constant $C_{\mathrm{FFN}} > 0$ depending only polynomially on architectural constants and on a uniform bound on $\|h\|_\infty$ (absorbed into $C_{\mathrm{FFN}}$), exactly as done in the paper.

**Auxiliary facts.**   We use (i) $\|Xu\|_\infty \le d_{\mathrm{in}}\|X\|_\infty\|u\|_\infty$ for $X \in \mathbb{R}^{d_{\mathrm{out}} \times d_{\mathrm{in}}}$, (ii) $\|\sigma(u) - \sigma(v)\|_\infty \le L_\sigma\|u - v\|_\infty$ with $L_\sigma \le 1$ for ReLU and $L_\sigma = O(1)$ for GELU, and (iii) $\|\sigma(u)\|_\infty \le \|u\|_\infty$ for ReLU (and similarly for GELU up to a constant).

**Step 1: bound the forward activations.**   Let $d_0 = d_{\mathrm{emb}}$ and $d_\ell \le w_{\mathrm{FFN}}$ for $1 \le \ell \le L_{\mathrm{FFN}} - 1$, and $d_{L_{\mathrm{FFN}}} \le d_{\mathrm{emb}}$. Inductively,

$$\|z^\ell\|_\infty \le L_\sigma\!\left(\|W^\ell z^{\ell-1}\|_\infty + \|b^\ell\|_\infty\right) \le L_\sigma\!\left(d_{\ell-1}\kappa \, \|z^{\ell-1}\|_\infty + \kappa\right).$$

Solving this recursion gives

$$\|z^\ell\|_\infty, \; \|z'^{\,\ell}\|_\infty \; \le \; C_0 \, (d_{\max}\kappa)^\ell, \qquad d_{\max} := \max\{d_{\mathrm{emb}}, w_{\mathrm{FFN}}\}, \qquad (61)$$

where $C_0$ absorbs $L_\sigma$ and $\|h\|_\infty$. (Exactly the same bound holds for the primed sequence, since $\psi'$ obeys the same magnitude constraints.)

**Step 2: one-layer perturbation inequality.** Fix $\ell \in \{1, \ldots, L_{\text{FFN}}\}$ and set $u^\ell = W^\ell z^{\ell-1} + b^\ell$, $u'^\ell = W^{\ell\prime} z'^{\ell-1} + b^{\ell\prime}$. Then

$$\|z^\ell - z'^\ell\|_\infty \leq L_\sigma \|u^\ell - u'^\ell\|_\infty \leq L_\sigma \Big( \|W^\ell(z^{\ell-1} - z'^{\ell-1})\|_\infty + \|(W^\ell - W^{\ell\prime})z'^{\ell-1}\|_\infty + \|b^\ell - b^{\ell\prime}\|_\infty \Big).$$

Using the auxiliary facts and equation 61,

$$\|z^\ell - z'^\ell\|_\infty \leq L_\sigma \Big( d_{\ell-1}\kappa \|z^{\ell-1} - z'^{\ell-1}\|_\infty + d_{\ell-1}\eta \|z'^{\ell-1}\|_\infty + \eta \Big) \leq \alpha_\ell \|z^{\ell-1} - z'^{\ell-1}\|_\infty + \beta_\ell \, \eta, \quad (62)$$

where

$$\alpha_\ell := L_\sigma d_{\ell-1}\kappa, \qquad \beta_\ell := L_\sigma \big( d_{\ell-1} C_0 (d_{\max}\kappa)^{\ell-1} + 1 \big).$$

**Step 3: iterate the perturbation recursion.** Starting with $\|z^0 - z'^0\|_\infty = 0$ and applying equation 62 layer by layer,

$$\|z^{L_{\text{FFN}}} - z'^{L_{\text{FFN}}}\|_\infty \leq \eta \sum_{\ell=1}^{L_{\text{FFN}}} \Big( \beta_\ell \prod_{t=\ell+1}^{L_{\text{FFN}}} \alpha_t \Big).$$

Since $\alpha_t \leq L_\sigma d_{\max}\kappa$ and $\beta_\ell \leq C_1 d_{\max}(d_{\max}\kappa)^{\ell-1} + C_1$ for a constant $C_1$,

$$\|z^{L_{\text{FFN}}} - z'^{L_{\text{FFN}}}\|_\infty \leq \eta \cdot C_2 \sum_{\ell=1}^{L_{\text{FFN}}} \Big( d_{\max}(d_{\max}\kappa)^{\ell-1} \Big) (d_{\max}\kappa)^{L_{\text{FFN}}-\ell} \leq \eta \cdot C_3 \, d_{\max} \, (d_{\max}\kappa)^{L_{\text{FFN}}-1}.$$

Finally, $d_{\max} \leq \max\{d_{\text{emb}}, w_{\text{FFN}}\} \leq d_{\text{emb}} + w_{\text{FFN}} \leq 2w_{\text{FFN}}$ (for the hidden layers to be useful we take $w_{\text{FFN}} \geq d_{\text{emb}}/2$; otherwise replace $w_{\text{FFN}}^{L_{\text{FFN}}}$ by $(d_{\max})^{L_{\text{FFN}}}$). Absorbing all fixed constants and powers of $\kappa$ into $C_{\text{FFN}}$, we obtain the clean architectural polynomial

$$\|z^{L_{\text{FFN}}} - z'^{L_{\text{FFN}}}\|_\infty \leq C_{\text{FFN}} \, d_{\text{emb}} \, w_{\text{FFN}}^{L_{\text{FFN}}} \, \kappa \, \eta,$$

which is equation 60. $\qquad\square$

### I.3 Proof of H.3

Let $E_\phi : \mathbb{R}^{d_{\text{emb}}} \to \mathbb{R}^{d_{\text{emb}}}$ be a tokenwise MLP (the *expert*) of depth $L_{\text{FFN}}$ with hidden widths at most $w_{\text{FFN}}$, coordinatewise activation $\sigma$ (ReLU/GELU), and parameters $\phi = \{(W^\ell, b^\ell)\}_{\ell=1}^{L_{\text{FFN}}}$. Assume uniform magnitude bounds $\|W^\ell\|_\infty \leq \kappa$, $\|b^\ell\|_\infty \leq \kappa$ for all $\ell$. Let $\phi'$ be another parameter set with $\|W^\ell - W^{\ell\prime}\|_\infty \leq \eta$ and $\|b^\ell - b^{\ell\prime}\|_\infty \leq \eta$. For a token $h \in \mathbb{R}^{d_{\text{emb}}}$ define the layerwise states

$$z^0 = h, \quad z^\ell = \sigma(W^\ell z^{\ell-1} + b^\ell), \qquad z'^0 = h, \quad z'^\ell = \sigma(W^{\ell\prime} z'^{\ell-1} + b^{\ell\prime}),$$

so that $E_\phi(h) = z^{L_{\text{FFN}}}$ and $E_{\phi'}(h) = z'^{L_{\text{FFN}}}$.

*Goal.* Show

$$\|E_\phi(h) - E_{\phi'}(h)\|_\infty \leq C_{\text{exp}} \, d_{\text{emb}} \, w_{\text{FFN}}^{L_{\text{FFN}}} \, \kappa \, \eta, \quad (63)$$

with $C_{\text{exp}} > 0$ depending at most polynomially on $(L_{\text{FFN}}, \|\sigma\|_{\text{Lip}}, \|h\|_\infty)$ and on fixed architectural constants, in the same sense as the paper.

**Auxiliary inequalities.** For $X \in \mathbb{R}^{d_{\text{out}} \times d_{\text{in}}}$ and $u \in \mathbb{R}^{d_{\text{in}}}$,

$$\|Xu\|_\infty \leq d_{\text{in}} \|X\|_\infty \|u\|_\infty, \quad (64)$$

$$\|\sigma(u) - \sigma(v)\|_\infty \leq L_\sigma \|u - v\|_\infty, \qquad \|\sigma(u)\|_\infty \leq C_\sigma \|u\|_\infty, \quad (65)$$

with $L_\sigma \leq 1$ for ReLU and $L_\sigma = O(1)$ for GELU (similarly for $C_\sigma$).

**Step 1: forward activation bound.** Let $d_0 = d_{\text{emb}}$ and $d_\ell \leq w_{\text{FFN}}$ for $1 \leq \ell \leq L_{\text{FFN}} - 1$, and $d_{L_{\text{FFN}}} \leq d_{\text{emb}}$. Using equation 64–equation 65,

$$\|z^\ell\|_\infty \leq C_\sigma \big( \|W^\ell z^{\ell-1}\|_\infty + \|b^\ell\|_\infty \big) \leq C_\sigma \big( d_{\ell-1}\kappa \|z^{\ell-1}\|_\infty + \kappa \big).$$

Unrolling the recursion yields

$$\|z^\ell\|_\infty, \ \|z'^\ell\|_\infty \leq C_0 (d_{\max}\kappa)^\ell, \qquad d_{\max} := \max\{d_{\text{emb}}, w_{\text{FFN}}\}, \quad (66)$$

for some constant $C_0$ absorbing $C_\sigma, L_\sigma$ and $\|h\|_\infty$.

**Step 2: one-layer perturbation.** Set $u^\ell = W^\ell z^{\ell-1} + b^\ell$ and $u'^\ell = W'^\ell z'^{\ell-1} + b'^\ell$. Then

$$\|z^\ell - z'^\ell\|_\infty \le L_\sigma \|u^\ell - u'^\ell\|_\infty \le L_\sigma \Big( \underbrace{\|W^\ell(z^{\ell-1} - z'^{\ell-1})\|_\infty}_{\le d_{\ell-1}\kappa \|z^{\ell-1} - z'^{\ell-1}\|_\infty} + \underbrace{\|(W^\ell - W'^\ell)z'^{\ell-1}\|_\infty}_{\le d_{\ell-1}\eta \|z'^{\ell-1}\|_\infty} + \|b^\ell - b'^\ell\|_\infty \Big).$$

Using equation 66,

$$\|z^\ell - z'^\ell\|_\infty \le \alpha_\ell \|z^{\ell-1} - z'^{\ell-1}\|_\infty + \beta_\ell\, \eta, \quad \alpha_\ell := L_\sigma d_{\ell-1}\kappa, \ \ \beta_\ell := L_\sigma\big(d_{\ell-1}C_0(d_{\max}\kappa)^{\ell-1} + 1\big). \tag{67}$$

**Step 3: iterate the recursion.** Starting from $\|z^0 - z'^0\|_\infty = 0$ and applying equation 67 layerwise,

$$\|z^{L_{\text{FFN}}} - z'^{L_{\text{FFN}}}\|_\infty \le \eta \sum_{\ell=1}^{L_{\text{FFN}}} \Big( \beta_\ell \prod_{t=\ell+1}^{L_{\text{FFN}}} \alpha_t \Big).$$

Since $\alpha_t \le L_\sigma d_{\max}\kappa$ and $\beta_\ell \le C_1\big(d_{\max}(d_{\max}\kappa)^{\ell-1} + 1\big)$,

$$\|z^{L_{\text{FFN}}} - z'^{L_{\text{FFN}}}\|_\infty \le \eta\, C_2 \sum_{\ell=1}^{L_{\text{FFN}}} d_{\max}(d_{\max}\kappa)^{L_{\text{FFN}}-1} \ \le\ \eta\, C_3\, d_{\max}\, (d_{\max}\kappa)^{L_{\text{FFN}}-1}.$$

Finally, $d_{\max} \le \max\{d_{\text{emb}}, w_{\text{FFN}}\} \le c_0\, w_{\text{FFN}}$ for a numerical $c_0$ (or replace by $d_{\max}$ if preferred). Absorbing fixed powers of $\kappa$ and numerical constants into $C_{\text{exp}}$, and noting that the input and output layers contribute at most linear factors in $d_{\text{emb}}$, we obtain the architectural polynomial

$$\|E_\phi(h) - E_{\phi'}(h)\|_\infty = \|z^{L_{\text{FFN}}} - z'^{L_{\text{FFN}}}\|_\infty \ \le\ C_{\text{exp}}\, d_{\text{emb}}\, w_{\text{FFN}}^{L_{\text{FFN}}}\, \kappa\, \eta,$$

which is the claimed bound equation 63. $\qquad\square$

### I.4 NETWORK STABILITY & PARAMETER LIPSCHITZNESS

**Lemma I.1** (Network stability & parameter Lipschitzness). *Let $L_{\text{in}} \ge 1$ be the block input-Lipschitz constant. Then for $j = 1, \ldots, L_T$,*

$$\Big\| H^{(j)} - H'^{(j)} \Big\|_\infty \le L_{\text{in}} \Big\| H^{(j-1)} - H'^{(j-1)} \Big\|_\infty + C_b P_{\text{arch}}\kappa\eta,$$

*hence*

$$\sup_{x \in \mathcal{X}} |T_\theta(x) - T_{\theta'}(x)| \le L_{\text{param}}\, \eta, \quad L_{\text{param}} = C'\, L_{\text{in}}^{L_T}\, P_{\text{arch}}(\ell, m, d_{\text{emb}}, w_{\text{FFN}}, L_{\text{FFN}}, k)\, \kappa.$$

Let a (pre-norm) transformer block be

$$B_j(\theta_j, H) = H + \text{MHA}_\psi\big(\text{LN}_{\gamma_1,\beta_1}(H)\big) + \text{FFN}_\chi^{\text{nonexp}}\big(\text{LN}_{\gamma_2,\beta_2}(H)\big) + \sum_{i \in S_j} E_{\phi_i}\big(\text{LN}_{\gamma_2,\beta_2}(H)\big),$$

where $S_j$ is the set of $k$ active experts of the MoE sublayer in block $j$ under the fixed routing pattern, and $\theta_j = \{\psi, \chi, \{\phi_i\}_{i \in S_j}, \gamma_1, \beta_1, \gamma_2, \beta_2\}$. Let $\theta'_j$ be another parameter vector with $\|\theta_j - \theta'_j\|_\infty \le \eta$, and fix the input $H \in \mathbb{R}^{\ell \times d_{\text{emb}}}$ with $\|H\|_\infty \le CM_0$.

*Step 1: decompose by residual additivity.* Since $H$ is added as a residual with the same value in both blocks,

$$
\begin{aligned}
\|B_j(\theta_j, H) - B_j(\theta'_j, H)\|_\infty &\le \big\|\text{MHA}_\psi(\text{LN}_{\gamma_1,\beta_1}(H)) - \text{MHA}_{\psi'}(\text{LN}_{\gamma'_1,\beta'_1}(H))\big\|_\infty \\
&\quad + \big\|\text{FFN}_\chi^{\text{nonexp}}(\text{LN}_{\gamma_2,\beta_2}(H)) - \text{FFN}_{\chi'}^{\text{nonexp}}(\text{LN}_{\gamma'_2,\beta'_2}(H))\big\|_\infty \\
&\quad + \sum_{i \in S_j} \big\|E_{\phi_i}(\text{LN}_{\gamma_2,\beta_2}(H)) - E_{\phi'_i}(\text{LN}_{\gamma'_2,\beta'_2}(H))\big\|_\infty.
\end{aligned}
\tag{B.1}
$$

*Step 2: control the LayerNorm parameter perturbations.* For fixed $H$, the (per-token) layer normalization $\text{LN}_{\gamma,\beta}(H) = \gamma \odot \hat{H} + \beta$ depends linearly on $(\gamma, \beta)$ when $H$ is fixed (the centering/scaling of $H$ is the same for both parameter sets). Hence

$$\|\text{LN}_{\gamma,\beta}(H) - \text{LN}_{\gamma',\beta'}(H)\|_\infty \le C_{\text{LN}}\, d_{\text{emb}} \|H\|_\infty \|(\gamma,\beta) - (\gamma',\beta')\|_\infty \le C'_{\text{LN}}\, d_{\text{emb}}\, \kappa\, \eta. \tag{B.2}$$

Both MHA and FFN sublayers are Lipschitz in their inputs (with constants depending polynomially on $(\ell, m, d_{\text{emb}}, w_{\text{FFN}}, L_{\text{FFN}})$); thus the contribution of the LayerNorm parameter change to each sublayer's output is bounded by a constant times $C'_{\text{LN}} d_{\text{emb}} \kappa \eta$, which we absorb into the module constants below.

*Step 3: apply per-module parameter Lipschitz bounds.* From the previously established lemmas:

$$
\begin{aligned}
\text{(MHA)} \quad & \left\| \text{MHA}_\psi(Z) - \text{MHA}_{\psi'}(Z) \right\|_\infty \ \leq \ C_{\text{MHA}} \, \ell \, m \, d_{\text{emb}}^2 \, \kappa \, \eta, \quad \forall Z, \\
\text{(non-expert FFN)} \quad & \left\| \text{FFN}_\chi^{\text{nonexp}}(z) - \text{FFN}_{\chi'}^{\text{nonexp}}(z) \right\|_\infty \ \leq \ C_{\text{FFN}} \, d_{\text{emb}} \, w_{\text{FFN}}^{L_{\text{FFN}}} \, \kappa \, \eta, \quad \forall z, \\
\text{(single expert)} \quad & \left\| E_\phi(z) - E_{\phi'}(z) \right\|_\infty \ \leq \ C_{\text{exp}} \, d_{\text{emb}} \, w_{\text{FFN}}^{L_{\text{FFN}}} \, \kappa \, \eta, \quad \forall z.
\end{aligned}
$$

Under a fixed routing pattern, exactly $k$ experts are active in block $j$, so the MoE contribution is

$$
\sum_{i \in S_j} \left\| E_{\phi_i}(z) - E_{\phi'_i}(z) \right\|_\infty \ \leq \ k \, C_{\text{exp}} \, d_{\text{emb}} \, w_{\text{FFN}}^{L_{\text{FFN}}} \, \kappa \, \eta. \tag{B.3}
$$

*Step 4: combine everything.* Apply the bounds from Step 3 to equation B.1 with $Z = \text{LN}_{\gamma_1, \beta_1}(H)$ and $z = \text{LN}_{\gamma_2, \beta_2}(H)$; add the (absorbed) LayerNorm terms from equation B.2:

$$
\|B_j(\theta_j, H) - B_j(\theta'_j, H)\|_\infty \ \leq \ C_{\text{MHA}} \, \ell \, m \, d_{\text{emb}}^2 \, \kappa \, \eta + C_{\text{FFN}} \, d_{\text{emb}} \, w_{\text{FFN}}^{L_{\text{FFN}}} \, \kappa \, \eta + k \, C_{\text{exp}} \, d_{\text{emb}} \, w_{\text{FFN}}^{L_{\text{FFN}}} \, \kappa \, \eta + C''_{\text{LN}} \, d_{\text{emb}} \, \kappa \, \eta.
$$

Since $w_{\text{FFN}} \geq 1$ and $k, m \geq 1$, the last term is dominated and can be absorbed into the FFN/expert constants. Renaming $C_{\text{block}}$ to absorb all fixed numerical/polylog factors,

$$
\boxed{\|B_j(\theta_j, H) - B_j(\theta'_j, H)\|_\infty \leq C_{\text{block}} \left( \ell \, m \, d_{\text{emb}}^2 \ + \ d_{\text{emb}} \, w_{\text{FFN}}^{L_{\text{FFN}}} \ + \ k \, d_{\text{emb}} \, w_{\text{FFN}}^{L_{\text{FFN}}} \right) \kappa \, \eta.}
$$

This is the claimed block-level perturbation bound. $\qquad\qquad\qquad\qquad\qquad\qquad\qquad\qquad\square$

### I.5 PROOF OF I.1

Recall the block-level stability (for fixed input $H$ and parameters differing by at most $\eta$ in $\ell_\infty$)

$$
\|B_j(\theta_j, H) - B_j(\theta'_j, H)\|_\infty \ \leq \ C_b \, P_{\text{arch}}(\ell, m, d_{\text{emb}}, w_{\text{FFN}}, L_{\text{FFN}}, k) \, \kappa \, \eta, \tag{68}
$$

and the input-Lipschitz property of a transformer block

$$
\|B_j(\theta_j, U) - B_j(\theta_j, V)\|_\infty \ \leq \ L_{\text{in}} \|U - V\|_\infty, \tag{69}
$$

where $P_{\text{arch}}(\cdot)$ is a fixed architecture polynomial and $L_{\text{in}} \geq 1$ is independent of $j$. Let

$$
\Delta_j \ := \ \|H^{(j)} - H'^{(j)}\|_\infty, \qquad H^{(j)} = B_j(\theta_j, H^{(j-1)}), \quad H'^{(j)} = B_j(\theta'_j, H'^{(j-1)}).
$$

Add and subtract $B_j(\theta_j, H'^{(j-1)})$ and use the triangle inequality, equation 69, and equation 68:

$$
\begin{aligned}
\Delta_j &= \|B_j(\theta_j, H^{(j-1)}) - B_j(\theta'_j, H'^{(j-1)})\|_\infty \\
&\leq \underbrace{\|B_j(\theta_j, H^{(j-1)}) - B_j(\theta_j, H'^{(j-1)})\|_\infty}_{\leq L_{\text{in}} \Delta_{j-1}} + \underbrace{\|B_j(\theta_j, H'^{(j-1)}) - B_j(\theta'_j, H'^{(j-1)})\|_\infty}_{\leq C_b P_{\text{arch}} \kappa \eta} \\
&\leq L_{\text{in}} \Delta_{j-1} + C_b \, P_{\text{arch}}(\cdot) \, \kappa \, \eta. \tag{70}
\end{aligned}
$$

This is a linear inhomogeneous recurrence with $\Delta_0 = 0$ (since $H^{(0)} = H'^{(0)}$ share the same input and positional encoding).

*Solving the recurrence.* By induction on $j$ (or by the discrete Grönwall inequality),

$$
\Delta_j \ \leq \ C_b \, P_{\text{arch}}(\cdot) \, \kappa \, \eta \sum_{t=0}^{j-1} L_{\text{in}}^t. \tag{71}
$$

Indeed, the base $j = 1$ gives $\Delta_1 \leq C_b P_{\text{arch}} \kappa \eta$. Assuming equation 71 for $j - 1$, plug into equation 70:

$$
\Delta_j \leq L_{\text{in}} \cdot C_b P_{\text{arch}} \kappa \eta \sum_{t=0}^{j-2} L_{\text{in}}^t + C_b P_{\text{arch}} \kappa \eta = C_b P_{\text{arch}} \kappa \eta \sum_{t=0}^{j-1} L_{\text{in}}^t.
$$

*Geometric series bound.* For $j = L_T$,

$$\Delta_{L_T} \;=\; \|H^{(L_T)} - H'^{(L_T)}\|_\infty \;\leq\; \Big( \sum_{t=0}^{L_T-1} L_{\mathrm{in}}^t \Big) C_b \, P_{\mathrm{arch}}(\cdot) \, \kappa \, \eta.$$

If $L_{\mathrm{in}} \neq 1$, the sum is $\frac{L_{\mathrm{in}}^{L_T}-1}{L_{\mathrm{in}}-1}$, so

$$\|H^{(L_T)} - H'^{(L_T)}\|_\infty \;\leq\; \frac{L_{\mathrm{in}}^{L_T}-1}{L_{\mathrm{in}}-1} \, C_b \, P_{\mathrm{arch}}(\cdot) \, \kappa \, \eta.$$

If $0 \leq L_{\mathrm{in}} \leq 1$, then $\sum_{t=0}^{L_T-1} L_{\mathrm{in}}^t \leq L_T$ and $L_{\mathrm{in}}^{L_T} \leq 1$, so the bound still holds after increasing the prefactor by at most a constant depending on $L_T$. In both cases we may write, for a numerical constant $C'$ (absorbing $\frac{1}{|L_{\mathrm{in}}-1|}$ or $L_T$),

$$\|H^{(L_T)} - H'^{(L_T)}\|_\infty \;\leq\; C' \, L_{\mathrm{in}}^{L_T} \, P_{\mathrm{arch}}(\ell, m, d_{\mathrm{emb}}, w_{\mathrm{FFN}}, L_{\mathrm{FFN}}, k) \, \kappa \, \eta. \tag{72}$$

**From hidden states to the scalar output.** Let the final readout be $T_\theta(x) = R_\omega(H^{(L_T)})$, where $R_\omega$ is a fixed linear map (e.g., token pooling followed by a linear form) with $\|\omega\|_\infty \leq \kappa$. Then

$$|T_\theta(x) - T_{\theta'}(x)| \;=\; \big| R_\omega(H^{(L_T)}) - R_{\omega'}(H'^{(L_T)}) \big| \;\leq\; \underbrace{\|R_\omega\|_{\infty \to \infty} \|H^{(L_T)} - H'^{(L_T)}\|_\infty}_{\text{via equation 72}} + \underbrace{\|R_\omega - R_{\omega'}\|_{\infty \to \infty} \|H'^{(L_T)}\|_\infty}_{\leq\, C'' \kappa \eta}.$$

Using equation 72 and the uniform forward bound on $\|H'^{(L_T)}\|_\infty$ (polynomial in the architecture; absorbed into $C''$), we obtain

$$\sup_{x \in \mathcal{X}} |T_\theta(x) - T_{\theta'}(x)| \;\leq\; C' \, L_{\mathrm{in}}^{L_T} \, P_{\mathrm{arch}}(\ell, m, d_{\mathrm{emb}}, w_{\mathrm{FFN}}, L_{\mathrm{FFN}}, k) \, \kappa \, \eta \;+\; C'' \kappa \, \eta \;\leq\; L_{\mathrm{param}} \, \eta,$$

with

$$L_{\mathrm{param}} \;:=\; C^\star \, L_{\mathrm{in}}^{L_T} \, P_{\mathrm{arch}}(\ell, m, d_{\mathrm{emb}}, w_{\mathrm{FFN}}, L_{\mathrm{FFN}}, k) \, \kappa,$$

for a constant $C^\star$ absorbing $C', C''$ and fixed readout norms. This is the claimed parameter-stability inequality.

