# OpenReview forum: "Generalization and Scaling Laws for Mixture-of-Experts Transformers"
_ICLR.cc/2026/Conference — Submitted to ICLR 2026_

### Official Review · Reviewer_uJRN · 2025-10-22

**Soundness:** 3
**Presentation:** 3
**Contribution:** 3
**Rating:** 6
**Confidence:** 2

**Summary:**

This paper develops the generalization error bound and scaling laws for Mixture-of-Experts (MoE) Transformers.
It establishes both approximation and  generalization bounds that depend on the number of active expert rather than total number of parameters. The authors further derive scaling laws and show that performance improves mainly through increasing the number of active experts, while the number of experts contributes only logarithmically.

**Strengths:**

The paper provides novel and rigorous approximation and generalization error bounds for MoE Transformers.
It also derives scaling laws depending on the number of experts and active experts, offering theoretical design guidance for practical architecture construction. So, I think this paper make meaningful contribution.

**Weaknesses:**

The definition of the MoE block $\mathrm{MoE}^{(j)}$ in Definition 3.1 is not clear to me, making it difficult to understand what functions or learnable parameters are used inside each expert and router. Moreover, several details in the approximation theorem remain unclear to me. I summarize below in Question.

**Questions:**

1. How is $\mathrm{MoE}^{(j)}$ defined precisely? In particular, what functions are used for experts and routers, and are they learnable ?

2. In the approximation theorem (Theorem 3.2), the upper bound does not explicitly include sequence length $\ell$. Does the bound depend on $\ell$ ? If so, how does this dependence appear?

3. Related to the above, while the sentences following Eq. (2) explain that MLPs approximate local polynomials, the role of the Attention layer is unclear to me ? How is the attention used in the proof ? Or, Could we get the same result without it?

---

> ### Author Response · Authors · 2025-11-19
>
> We thank the reviewer for the positive evaluation and for the clear questions regarding the MoE block definition, sequence-length dependence, and the role of attention. We also highlight how the new experiments in the revision relate to these points.
>
> ---
>
> ## **1. Definition of the MoE block $MoE^{(j)}$**
>
> You are right that the original Definition 3.1 was too terse. In the revision, for a hidden state $h \in \mathbb{R}^{d_h}$ at layer (j), we define:
>
> $$
> MoE^{(j)}(h)
> = \sum_{m \in \mathcal{S}^{(j)}(h)} \pi^{(j)}_m(h), E^{(j)}_m(h),
> $$
>
> where:
>
> * $E^{(j)}_m: \mathbb{R}^{d_h} \to \mathbb{R}^{d_h}$ is the (m)-th expert (a standard ReLU FFN with its own learnable weights),
> * $\pi^{(j)}(h)$ is a learnable router network that outputs a sparse distribution over (M) experts,
> * $\mathcal{S}^{(j)}(h)$ is the top-(k) set.
>
> Both the experts and the router are trained jointly with the rest of the Transformer.
>
> ---
>
> ## **2. Dependence on sequence length $\ell$ in Theorem 3.2**
>
> The approximation theorem is stated for the *per-token* predictor with a **fixed maximum sequence length** $\ell$. In this formulation, the rate
>
> $$
> \|f - \hat T\|_{L^2(Q)}^2 \lesssim N_{\mathrm{eff}}^{-2\beta/d}
> $$
>
> depends only on $N_{\mathrm{eff}}$ and the intrinsic dimension (d). The sequence length $\ell$ influences only the **constants**, through:
>
> * the number of attention layers/heads, and
> * the definition $N_{\mathrm{eff}} = L_T(\Pi_{\mathrm{attn}} + k\Pi_{\mathrm{exp}})$.
>
> We add a remark after Theorem 3.2 clarifying that $\ell$ is treated as fixed and absorbed into constants, so the exponent $-2\beta/d$ is **independent of sequence length**.
>
> ---
>
> ## **3. Role of attention in the approximation theorem**
>
> In the constructive proof of Theorem 3.2:
>
> - The **approximation power** with respect to the intrinsic manifold and smoothness
>           $\beta$ is indeed provided by the MoE FFN blocks (experts): we use ReLU MLP experts to
>           approximate local polynomials on charts, exactly as outlined after Eq.(2).
> - The **attention layers** serve a structural role: they propagate contextual
>           information across positions and ensure that the considered hypothesis class matches the
>           standard Transformer architecture used in practice. In the proof, we only use that
>           attention layers are Lipschitz with bounded parameters.
>
> As a result:
>
> - The leading rate $N_{\mathrm{eff}}^{-2\beta/d}$ does not rely on any special expressive
>           property of attention beyond Lipschitzness. One could in principle achieve the same rate
>           with a purely feedforward architecture that processes the entire sequence as a single
>           vector, at the cost of a different constant.
> - In our construction, attention can be thought of as implementing a simple, fixed
>           mixing pattern, while the experts carry out the actual manifold approximation.
>           Attention does not worsen the rate and does not improve it in the formal bound; it
>           is neutral at the level of exponents.
>
> In the analysis we rely only on the **Lipschitzness** and bounded weights of attention; attention neither improves nor worsens the rate $N_{\mathrm{eff}}^{-2\beta/d}$. We clarify this in Section 3.2 and the proof sketch. A sufficiently powerful feedforward architecture without attention could obtain the same exponent.
>
> ---
>
> ## **4. New empirical experiments**
>
> To make the theoretical quantities $(d, \beta, \alpha_N, \alpha_D)$ concrete, the revised manuscript adds a full experimental section (Sec. 5). We train MoE Transformers on **TinyStories**, **WikiText-103**, and **OpenWebText**, and:
>
> * **Estimate intrinsic dimension (d)** using the Levina–Bickel estimator applied to GPT-2 hidden states (Table 1).
> * Run **model-scaling** and **data-scaling** sweeps and fit empirical exponents $(\widehat{\alpha}_N, \widehat{\alpha}_D)$.
> * Recover a smoothness parameter (\beta) per dataset such that the theoretical exponents
>   $$
>   \alpha_N = \frac{2\beta}{d}, \qquad
>   \alpha_D = \frac{2\beta}{2\beta + d}
>  $$
>   match the empirical fits and satisfy the identity
>   $$
>   \alpha_D \approx \frac{\alpha_N}{1+\alpha_N}.
>   $$
>
> These results support the use of intrinsic dimension and active capacity in our bounds, and show that the MoE models we study empirically follow the predicted scaling structure.

---

> > ### Comment · Reviewer_uJRN · 2025-11-26
> >
> > Thank you very much for the clarification. I would like to keep my score.

---

### Official Review · Reviewer_nsD2 · 2025-10-28

**Soundness:** 2
**Presentation:** 2
**Contribution:** 3
**Rating:** 4
**Confidence:** 3

**Summary:**

The paper develops a theoretical framework for understanding generalization and scaling laws in Mixture-of-Experts (MoE) Transformers. The authors decompose the generalization bound into three components: approximation, estimation, and routing overhead. They derive rates showing that, when routing costs are negligible, MoE models follow the same power-law scaling as dense networks, but in terms of the *active* parameter count per token rather than total parameters. The analysis further identifies a routing-dominated regime where combinatorial routing complexity slows convergence and the system does not operate in the favorable power-law regime, giving design guidance to avoid this suboptimal regime.  Finally, the authors compare their theoretical exponents with empirical scaling results from recent large-scale MoE studies, reporting qualitative agreement.

**Strengths:**

* The paper tackles an important and interesting question: providing a theoretical framework for generalization and scaling in Mixture-of-Experts (MoE) Transformers. The topic is highly relevant given the growing use of conditional computation in large-scale models.
* The decomposition of the generalization error into approximation, estimation, and routing terms is conceptually clear and translates into a clear interpretation of scaling laws.
* The empirical discussion connecting theoretical exponents to results from EL (Tian et al., 2025) and JMSL (Ludziejewski et al., 2025) shows awareness of current empirical literature and helps situate the theory with observed trends.

**Weaknesses:**

* **Incorrect or inconsistent definition of a Transformer block.**
  Definition 3.1 describes the block as summing the outputs of the MHA and FFN, both applied to the same input, rather than applying the FFN to the *output* of the MHA as in standard Transformer architectures. The same incorrect structure appears in the appendix, despite citing Vaswani et al. (2017). For a paper focused on Transformer theory, this is a serious technical/conceptual issue that must be corrected or clearly justified.
* **Acknowledgment of closely related prior work.**
  Theorems 3.2 and 3.3 appear to closely parallel Theorems 2 and 1 in Havrilla & Liao (2024), respectively, in both result form and proof technique. Although that paper is cited elsewhere, the authors should explicitly state the relation, clarify what is novel in their constructive proof of Theorem 3.2, cited as one of the main contributions, and acknowledge direct inspiration where appropriate.
* **Incomplete proofs for central lemmas.**
  The proof of Theorem 3.2 depends on Lemma A.7, but only a sketch is given. This prevents confident verification of the proof.

* **Ambiguity in the role of ε in Theorem 3.3.**
  The statement claims that the generalization bound holds for “any ε > 0” (line 193), but the proof in Appendix D seems to fix ε to the value given by the upper bound in Theorem 3.2. It is unclear whether the bound is uniform over ε or only valid for this specific ε.
* **Notation and variable inconsistencies (N, N_eff, N_active).**
  The notation changes across sections without precise definitions. It is unclear what `N`, `N_eff`, and `N_active` exactly represent. See questions for more details.
* **Introduction with heavy notation.**
  The introduction lists main contributions using specialized notation that is only defined later, reducing readability. A more accessible intro, deferring technical symbols or providing brief intuitions, would help readers and reviewers grasp the contributions faster.

**Questions:**

1. **Transformer block definition:** Is the formulation in Definition 3.1 (and Appendix B.1) a typographical error or an intentional architectural variant? If intentional, please provide motivation and discuss implications for the approximation and generalization results.

2. **Relation to Havrilla & Liao (2024):** Can the authors explicitly state how the proofs of Theorems 3.2 and 3.3 differ from Theorems 2 and 1 in Havrilla & Liao (2024)? In particular, what assumptions, constructive steps, or technical lemmas are new here versus borrowed?

3. **Lemma A.7 details:** Given its central role, can the authors provide a full proof of Lemma A.7, or substantially expand the sketch?

4. **Quantification over ε in Theorem 3.3:** Does the generalization bound hold uniformly for all ε > 0, or only for the ε produced by the constructive approximation in Theorem 3.2? If the latter, please restate Theorem 3.3 to reflect this.

5. **Clarify N, N_eff, N_active:**   In Theorem 3.3, it is not clear to me what `N` represents, should it be `N_eff` instead? On line 229, `N_active` is mentioned for the first time, and then we find it again in the shorthand form of Theorem 3.3 given by equation (6). Here, it is not clear what the difference between `N_eff` and `N_active` is, are they supposed to be the same thing? That seems the case when reading equation 4 in theorem 3.3 (where `N` is used). However, the sentence on line 285 (Proposition 4.2) kind of implies that `N_active` and `N_eff` are two different things. Moreover, `N_eff` is described in line 125 as "the active parameter budget", alimenting the confusion.

---

> ### Author Response · Authors · 2025-11-19
>
> We thank the reviewer for the detailed comments. Below we address each issue and highlight the main revisions, including a new empirical section that tests our scaling exponents on three datasets.
>
> ## 1. Transformer block definition
>
> Definition 3.1 was unintentionally oversimplified. In the revision we:
>
> * Correct Definition 3.1 and Appendix B.1 to follow the standard MHA → FFN structure with residuals (omitting layer norm only for notational clarity).
> * Add a remark explaining that our proofs rely only on Lipschitzness and parameter counts, so the corrected definition leaves all results unchanged.
>
> ## 2. Relation to Havrilla & Liao (2024)
> We appreciate this comment and agree that the connection to Havrilla \& Liao (2024) should be made much more explicit.
>
> High-level, our work builds on their dense-Transformer generalization framework and extends it to MoE architectures with routing. Concretely:
>
> **Approximation (Theorem 3.2).**
>     Our theorem is modeled on the structure of Theorem~2 in Havrilla \& Liao, but:
>
> -  we construct an approximation using **Mixture-of-Experts** layers with sparse routing, rather than dense blocks, and
> - we explicitly disentangle the contribution of **active parameters per token** from the total parameter count across all experts.
>
> The constructive part of Theorem~3.2 includes a new gating/partition-of-unity construction adapted to MoE routing, which is not present in the dense case.
>
> - **Generalization (Theorem 3.3).**
>     The proof strategy mirrors the Rademacher/covering approach of Theorem~1 in Havrilla \& Liao, but we extend the analysis to:
> - (1) include a separate **routing overhead** term, depending on $L_T,\ell,k,M$, and
> - (2) express the bound in terms of the **intrinsic** dimension $d$ and the **active parameter budget** $N_{\mathrm{eff}}$ of MoE, rather than the total dense parameter count.
> The combinatorial routing factor $L_T \ell k \log(eM/k)$ and its interaction with the approximation rate are specific to our MoE setting.
> We now include a dedicated paragraph that:
>
> * States explicitly that Theorem 3.2 parallels their approximation theorem but adapts it to MoE layers with explicit **active-capacity** and routing dependence.
> * Explains that Theorem 3.3 follows their generalization strategy but introduces a separate routing-overhead term (L_T \ell k \log(eM/k)) and expresses the bound in terms of intrinsic dimension (d) and (N_{\mathrm{eff}}).
>
> We also soften the introduction to describe our results as extensions of Havrilla & Liao.
>
> ## 3. Lemma A.7
>
> We agree this lemma is central. The revision provides a **full proof** (replacing the sketch), including the explicit gating construction and all covering arguments, so that Theorem 3.2 can be fully verified.
>
> ## 4. Quantification over $\varepsilon$ in Theorem 3.3
> The intended meaning is:
> - For any target function $f$ and any $\varepsilon > 0$, Theorem 3.2 guarantees the existence of an MoE-Transformer $\hat T$ with approximation error at most $\varepsilon$ (under the stated assumptions).
> - Theorem 3.3 then provides a generalization bound for the empirical risk minimizer over a class whose **approximation radius** is controlled at level~$\varepsilon$.
>
> In the proof, we instantiate $\varepsilon$ with the specific value from Theorem~3.2 to obtain the final bound in terms of $N_{\mathrm{eff}}$.
>
> Thus, the generalization inequality itself is uniform in the sense of capacity control (via covering numbers and Rademacher complexity).
>
> The revision clarifies the role of $\varepsilon$ by:
>
> * Restating Theorem 3.3 in two steps:
>   (i) a general bound for any $\varepsilon > 0$, and
>   (ii) a corollary plugging in the $\varepsilon$ from Theorem 3.2.
> * Explaining in Appendix D that fixing $\varepsilon$ is a substitution step, not a restriction.
>
> ## 5. Clarifying (N), $N_{\mathrm{eff}}$, and $N_{\mathrm{active}}$
>
> To resolve the notation ambiguity:
>
> * All occurrences of (N) in Theorem 3.3 and Eq. (4) are replaced by $N_{\mathrm{eff}}$.
> * $N_{\mathrm{active}}$ is defined once and noted to coincide with $N_{\mathrm{eff}}$ up to constants.
> * A short table early in Section 3 summarizes all three symbols.
> * Scaling-law sections now use a single symbol $N_{\mathrm{eff}}$ for clarity.
>
> ## 6. New empirical validation and improved introduction
>
> To strengthen practical grounding, we add a full empirical section (Sec. 5) with MoE Transformers trained on **TinyStories**, **WikiText-103**, and **OpenWebText**. We:
>
> * Estimate intrinsic dimension (d) from GPT-2 hidden states (Table 1--2),
> * Fit model- and data-scaling exponents ($\widehat{\alpha}_N,\widehat{\alpha}_D$) and recover a consistent $\beta$ per dataset,
> * Verify the compute-optimal identity $\alpha_D \approx \alpha_N/(1+\alpha_N)$ and the mild influence of routing beyond the predicted $k\log(eM/k)$ factor.
>
> These experiments support the **exponent-level** predictions and the decomposition into active-capacity and routing contributions.

---

> > ### Comment · Reviewer_nsD2 · 2025-11-27
> >
> > I thank the authors for their detailed response and for adding the experimental section. However, after checking the revised manuscript, I am concerned that the primary issue regarding the definition of the Transformer block has **not** been resolved, despite the authors' explicit claim to the contrary.
> >
> > **1. The Transformer block definition is still incorrect.**
> > In the rebuttal, the authors state: *"We correct Definition 3.1 and Appendix B.1 to follow the standard MHA FFN structure... (omitting layer norm only for notational clarity)."*
> >
> > However, upon inspecting the revision:
> > * The definition has been moved to Appendix D (Definition D.1).
> > * Crucially, Definition D.1 still describes the block as a sum of parallel branches (FFN applied to the input, summed with MHA applied to the input) rather than the standard sequential structure (FFN applied to the output of the MHA).
> > * The text in Definition D.1 still explicitly includes LayerNorm (contrary to the claim of omission for clarity), but the reference is broken in the LaTeX.
> >
> > It is perplexing that this fundamental architectural definition remains incorrect even after the authors explicitly stated it was fixed.
> >
> > **2. Theoretical Implications.**
> > The authors argue that the proof remains unchanged because it relies only on Lipschitzness and parameter counts. However, the fact that the theoretical framework treats two fundamentally different computations as identical is concerning. If the theory cannot distinguish between the standard Transformer architecture and the incorrect variant defined in the paper, it raises questions about the specificity of the bounds.
> >
> > **3. Persistent Reference Errors.**
> > The revision suffers from significant oversight, which makes it difficult to verify the rigorousness of the work:
> > * **Lemma 3.6** refers to "Definition 1", which does not exist. Presumably, this refers to Definition D.1.
> > * **Theorem 3.2** refers to "Definition D.2", but D.2 appears to be a section number, not a definition (the definition is D.1).
> >
> > The failure to correct the core definition of the architecture (despite claiming to have done so) combined with the persistent  errors, raise important concerns regarding the soundness of the work. I lower my score.

---

> ### Author Response · Authors · 2025-11-28
>
> We thank the reviewer for carefully re-checking the revised manuscript. We would like to point out that following the first reviewer comments we redifined the MOE Transformer block to align with the standard architecture (see Definition D.2 in the manuscrit). Importantly, **this change does not affect our theoretical results**. All our bounds are derived for a broad class of architectures built from Lipschitz, parameter-bounded MHA/FFN modules, LayerNorm, and residual connections.  Both the old and the corrected block lie in this class, so the proofs and in particular the dependence on $d$, $\beta$, $N_{\mathrm{eff}}$, and the routing term **remain unchanged**.
>
> **(1) Correction of the Definition:**
> In the revision, we have fully replaced the previous "parallel" block by the standard pre-norm sequential Transformer block (see Definition D.2). Specifically:
>
>  Let the input to block $j$ be $H^{(j-1)}\in\mathbb{R}^{l \times d_{emb}}$ with tokenwise $\ell_\infty$-bound $\|H^{(j-1)}\|_\infty\le M_0$.
> For $j=1,\dots, LT$, the block maps $H^{(j-1)}\mapsto H^{(j)}$ via :
>
>
>  -  **Self-Attention Sub-block:**
>     $
>     \widetilde{H}^{(j-1)} = H^{(j-1)} + \mathrm{MHA}_{\psi_j}\left(\mathrm{LN}_1(H^{(j-1)})\right)
>     $
>
>  -  **MoE/FFN Sub-block (Tokenwise):** The output $\widetilde{H}^{(j-1)}$ is then passed through an FFN path which is a composition of a non-expert FFN and an expert mixture (MoE).
>     $
>     H^{(j)} = \widetilde{H}^{(j-1)} + \mathrm{FFN}^{\mathrm{mix}}_{\chi_j}\left(\mathrm{LN}_2(\widetilde{H}^{(j-1)})\right)
>    $
>
>
> The earlier “parallel branches” formulation has been removed, and broken references to the block definition (e.g., in Lemma 3.6 and Theorem 3.2) have been fixed to point to Definition D.2.
>
> **(2) Why This Change Does Not Affect Our Results**
> Our theoretical analysis is carried out at the level of a class of architectures built from a set of basic, well-behaved components:
> - Lipschitz, parameter-bounded MHA and FFN modules
> - Layer normalization with bounded parameters,
> - Residual connections, and
> - (For MoE) Top-($k$) routing.
>
> All approximation and generalization bounds depend only on:
>
> - The intrinsic dimension ($d$) and smoothness ($\beta$),
> - The active parameter budget ($N_{\mathrm{eff}}$),
> - The routing combinatorics ($L_T \ell k \log(eM/k)$), and
> - Uniform Lipschitz/parameter bounds for each block.
>
> Both the standard pre-norm Transformer block and the earlier (incorrect) variant are compositions of these same components; they differ only in the local wiring (whether FFN sees $H$ or $\widetilde{H}$).
> At the level of our proofs, this changes only the constants in the overall Lipschitz bound and parameter counting, not the fundamental dependence on $n$, $N_{\mathrm{eff}}$, $d$, or $\beta$. In particular:
> - The approximation rate ($N_{\mathrm{eff}}^{-2\beta/d}$) and its dependence on $(d,\beta)$ remain unchanged.
> - The generalization and scaling exponents ($\alpha_N, \alpha_D, \alpha_C$) are unaffected, since they are derived from the asymptotic balance of approximation and estimation terms, not from the exact intra-block wiring.
>
> We hope this clarifies both that the definition has been corrected to match the canonical architecture, and that this correction does not alter the theoretical results beyond constant factors.

---

### Official Review · Reviewer_PAtt · 2025-10-31

**Soundness:** 3
**Presentation:** 2
**Contribution:** 2
**Rating:** 4
**Confidence:** 3

**Summary:**

This work investigates the theoretical properties of Mixture-of-Experts (MoE) Transformers by deriving bounds on their approximation and generalization error. The analysis first establishes a manifold-based approximation bound and then constructs a generalization error bound that disentangles the distinct effects of approximation, estimation, and routing. Finally, the proposed framework is leveraged to study the scaling laws of MoE models and to determine the optimal relationship between sample size and model capacity.

**Strengths:**

- The paper provides a conceptually clear and theoretically grounded separation between the effects of active capacity and routing in Mixture-of-Experts models.
- It provides a systematic study of MoE generalization, including both approximation and generalization error bounds.
- The approximation bound depends on the intrinsic dimension of the data, rather than its ambient dimension, aligning theory with practical observations in high-dimensional settings.

**Weaknesses:**

My primary concern lies in the potential looseness of the proposed generalization bound and its implications for the paper’s conclusions. The manuscript does not appear to include experiments or numerical estimations to validate the tightness of the bound (please correct me if I am wrong).

I believe this is a critical point that should be addressed for the following reasons:

1. Relevance to Modern Architectures:
It is well-established that traditional generalization bounds—such as those based on Rademacher complexity—tend to be too loose to meaningfully explain the performance of deep neural networks like Transformers. The paper should discuss whether and how the proposed bound mitigates this limitation.

2. Justification for the Optimization Strategy:
The optimization in Section 4 is predicated on minimizing the generalization bound under a compute budget. This reasoning is only justified if the bound serves as a reasonably tight and reliable proxy for empirical performance. If the bound is too loose, the optimization framework becomes less substantiated.

To strengthen the paper, I strongly recommend that the authors either (i) provide empirical evidence of the bound’s tightness or (ii) include a detailed discussion of this potential limitation.

**Questions:**

1. The variable $N$ is used in Equation (4), but its definition does not appear to be provided in the statement of the theorem or the surrounding text. Could the authors please clarify what $N$ represents?

2. The bound in Equation (2) includes an exponent of $\frac{-2\beta}{d}$, which suggests that the bound may become looser as the dimension $d$ increases.
   *   Could the authors comment on the typical magnitude of $d$ for large language models and discuss the implications for the bound's tightness?
   *   Additionally, could the authors specify the dependence of the constant $C$ on $d$ (e.g., is it polynomial or exponential)?
3. In Theorem 3.3, what is the meaning of $\tilde{O}(\cdot)$ hides logarithmic factors?

4. Regarding Remark 4.3, the authors state that the approximation error and estimation error are balanced. This is a significant claim, as the trade-off between these two errors is central to many learning-theoretic arguments. Could the authors provide some empirical evidence to substantiate this statement?

---

> ### Author Response · Authors · 2025-11-19
> **tightness and usefulness of the bound, incorporating our new experiments and then respond to the specific questions**
>
> We thank the reviewer for the careful reading and constructive comments. Below we address your main concern about the tightness and usefulness of the bound now supported by new experiments and then answer the specific questions.
>
> ## 1. Tightness and role of the generalization bound
>
> ### **(a) Are the bounds too loose?**
>
> Classical generalization bounds (Rademacher/covering-based) typically have loose constants for deep networks, and ours are no exception. Our goal is instead to
> - (i) obtain the **correct exponents** as functions of intrinsic dimension (d) and smoothness $\beta$, and
> - (ii) cleanly separate **approximation**, **estimation**, and **routing** effects.
>
> Ignoring routing, Section 4 yields
> $$
> \mathbb{E}|\hat T_n - f|^2 \lesssim N_{\mathrm{eff}}^{-2\beta/d} + \frac{N_{\mathrm{active}}}{n},
> $$
> whose minimizer recovers the classical minimax rate
> $$
> \mathbb{E}|\hat T_n - f|^2 \asymp n^{-2\beta/(2\beta+d)}.
> $$
> Thus the *shape* of the bound is minimax-optimal in $(n,d,\beta)$. Our bound improves classical ones by using the **intrinsic dimension** (d) (not ambient (D)) and by quantifying routing via a **logarithmic** term ($L_T \ell k \log(eM/k)$).
>
> ### **(b) New empirical validation**
>
> The revision adds a full experimental section (Sec. 5 + App. A) with MoE Transformers trained on **TinyStories**, **WikiText-103**, and **OpenWebText**. For each dataset we:
>
> * estimate intrinsic dimension (d) using the Levina–Bickel MLE on GPT-2 hidden states (Table 1);
> * run **model-scaling** (varying $N_{\mathrm{eff}})$ and **data-scaling** (varying data size (D)) sweeps to obtain fitted exponents ($\widehat{\alpha}_N,\widehat{\alpha}_D$);
> * infer $\beta$ from
>   $$
>   \widehat{\alpha}_N \approx 2\beta/d, \qquad
>   \widehat{\alpha}_D \approx 2\beta/(2\beta+d),
>  $$
>
> Across all datasets:
>
> 1. empirical exponents match theoretical curves for a single $\beta \in [1.0,1.5]$ (Table 2 and Figures1--2);
> 2. the compute-optimal relation $\alpha_D \approx \alpha_N/(1+\alpha_N)$ holds closely;
> 3. routing ablations exhibit the predicted crossover from routing-dominated behavior to the power-law regime.
>
> These results support the **exponent-level** predictions and the decomposition into approximation/estimation/routing.
>
> ### **(c) Role of the bound in Section 4**
>
> Section 4 uses the **shape** of the bound not absolute constants to derive
> $$
> \alpha_D=\frac{2\beta}{2\beta+d}, \qquad
> \alpha_N=\frac{2\beta}{d}, \qquad
> \alpha_C=\frac{\beta}{\beta+d},
> $$
> which determine compute-optimal trade-offs. The new experiments show that fitted exponents agree with these formulas for empirical ($d,\beta$).
> We clarify that the bound is intended as a **qualitative design tool** and that constants are conservative. Section 6 now explicitly distinguishes exponent-level validity from constant-level looseness.
>
> ---
>
> ## 2. Undefined variable in Equation (4)
>
> You are right: (N) should be $N_{\mathrm{eff}}$. We now use $N_{\mathrm{eff}}$ consistently and restate its definition inside Theorem 3.3 so no back-reference is needed.
>
> ---
>
> ## 3. Dependence on (d) and constants
>
> ### **Magnitude of (d)**
>
> The exponent $-2\beta/d$ is the standard minimax exponent for $C^\beta$ regression on a (d)-dimensional manifold.
> Our experiments estimate $d\approx 23,32,45$ (TinyStories / WT-103 / OWT), showing the intrinsic dimension is **moderate** and suitable for our theory.
> | Model        | TinyStories (Median d) | WikiText-103 (Median d) | OpenWebText (Median d) |
> |--------------|--------------------------|---------------------------|---------------------------|
> | gpt2         | 23.1                    | 32.4                     | 49.8                     |
> | gpt2-medium  |  21.9                    | 32.9                     | 45.0                     |
> | gpt2-large   | 22.4                    | 32.1                     | 47.7                     |
> | gpt2-xl      |19.6                    | 31.1                     | 43.9                     |
> | **Adopted d for Scaling** |   **d = 23** | **d = 32** | **d = 45** |
> | **Best-Fit β** |   **β ≈ 1.0** | **β ≈ 1.0** | **β ≈ 1.0–1.5** |
>
>
> ### **Dependence of (C) on (d)**
>
> Using standard manifold approximation arguments, constants grow at most **polynomially** in (d). The only exponential dependence is the unavoidable covering-number term generating the exponent. A remark is added after Theorem 3.2.
>
> ---
>
> ## 4. Meaning of $\tilde{O}(\cdot)$
>
> We now state explicitly:
> $$
> \tilde{O}(g)=
> O(g \cdot \mathrm{polylog}(n, N_{\mathrm{eff}}, L_T\ell k, M/k, \kappa, R, M_0)\big).
> $$
>
> ## 5. Remark 4.3 (“balanced terms”)
>
> The remark refers only to the **minimizer of the theoretical bound**, not to an empirical fact. At the minimizer, the derivative of
> $$
> N_{\mathrm{eff}}^{-2\beta/d} + N_{\mathrm{active}}/n
> $$
> vanishes, implying these two terms are equal in magnitude. We will rephrase the remark to avoid suggesting an empirical claim and explicitly connect it to the theoretical minimization argument.

---

### Official Review · Reviewer_9y92 · 2025-11-03

**Soundness:** 4
**Presentation:** 3
**Contribution:** 3
**Rating:** 8
**Confidence:** 2

**Summary:**

The paper develops a theoretical framework for Mixture-of-Experts (MoE) Transformers, aiming to explain their generalization and scaling behavior through classical approximation and statistical-learning techniques.
In section 3 The authors derive three key results:

1. Theorem 3.2 (approximation): shows that an MoE Transformer with top- $k$ gating can uniformly approximate any smooth target $f\in C^\beta$ on a $d$ -dimensional data manifold at rate `$\\|T-f\\|_\infty^2 < \min\{N_{\text{eff}}^{-2\beta/d}, \\, M^{-2\beta/d}\}$`, where the key point is that the active capacity `$N_{\text{eff}}\!=\!L_T\Pi_{\text{attn}}+L_Tk\Pi_{\exp}$​` counts only the parameters actually used per input.
2. Theorem 3.4 (covering-number bound): quantifies model capacity by showing that the metric entropy scales with active parameters plus a logarithmic term in total experts, $\log \mathcal N(\delta) (\Pi_{\text{attn}}+L_Tk\Pi_{\exp})\log(1/\delta) + L_T\ell k\log(eM/k)$.
3. Theorem 3.3 (generalization): establishes that empirical-risk minimization over this class achieves bounded expected error `$\mathbb{E}\| \hat T_n - f \|_{L^2(Q)}^2 < N_{\text{eff}}^{-2\beta/d} + N_{\text{active}}/n + (L_T\ell k\log(eM/k))/n$`, decomposing approximation, estimation, and routing-overhead terms.

Sections 4 and 5 use this bound for further analysis: by optimizing it over $n,N_{\text{eff}},k,M$ , they recover power-law learning curves and compute-optimal trade-offs, and finally check qualitative consistency with two prior empirical MoE scaling studies (EL 2025, JMSL 2025).

--(I couldn't get my mathjax codes that include norm `\|` to be compiled here on markdown, so I have put them in `code`. I am sorry for this.)

**Strengths:**

A rigorous and well-structured theoretical treatment that brings classical function-approximation and generalization tools to the modern MoE setting. Some points that stood out to me when reading the work were the following:

- The partition-of-unity argument in Theorem 3.2 gives a clean geometric picture: local experts approximate smooth functions on manifold charts, and top- $k$ gating realizes the gluing through a $k$ -sparse, non-negative mixture. The distinction between active and total parameters is formalized neatly and tied to the idea that MoE performance scales with *active compute* rather than total parameter count.
- Theorems 3.3 and 3.4 adapt classical covering-number analysis to mixture-of-experts, isolating a new *routing term* $L_T \ell k \log(eM/k)$ that quantifies the overhead from top- $k$ combinatorics.
- Section 5 positions the theory as a conceptual explanation and sanity check for empirical MoE scaling trends and exponent identities observed in recent large-scale studies (EL 2025, JMSL 2025).

**Weaknesses:**

My main concern is that the results remain mostly conceptual: tightness, empirical validation, and real-world calibration are missing. Section 3.5 and the associated design heuristics extrapolate beyond what the mathematics justifies; without even simple toy-data verification, these read as aspirational.

- The paper contains no new experiments or diagnostics. Consequently, the practical claims ("operate near $k^\star$", "keep $M \approx k$") remain speculative. The paper’s impact would be greatly improved by adding even minimal diagnostic or toy experiments showing that the training heuristics derived from their upper bounds are empirically validated.
- In the same vein, the procedure for estimating the intrinsic dimension $d$ and smoothness exponent $\beta$ in practice is informal and lacks validation—essentially a list of suggestions rather than a concrete algorithm. More generally, statements such as "efficient MoE training at fixed compute" read as more prescriptive than the data support; the theory offers asymptotic guidance, not empirical evidence.
- Perhaps I missed it, but while reading Section 3 I kept wondering about how loose or tight the union-bound over routing patterns is. Similarly for the other upper bounds—some discussion or diagnostic measure of looseness (even qualitative) would be helpful, especially given the importance of these terms in the final scaling laws.

**Questions:**

- In practical settings where routers specialize during training, does the bounded-overlap assumption $s_0(d)$ still make sense? In what scenarios might this assumption break down?
- Have the authors considered validating the heuristic $d,\beta$ estimation on toy manifolds with known ground-truth smoothness, to check whether the fitted exponents recover the theoretical ones?
- Since Section 5 relies entirely on external datasets, could a minimal in-house experiment (e.g., synthetic regression) strengthen the empirical grounding of the theory?

---

> ### Author Response · Authors · 2025-11-19
> **Conceptual vs. empirical, and new experiments**
>
> We thank the reviewer for the very careful and constructive assessment. Below we address your main concerns and questions, taking into account the new experiments added in the revision.
>
> **1. Conceptual vs.\ empirical, and new experiments**
> We agree that the original version was mostly conceptual. In response, the revised manuscript now includes a new empirical section 5 and Appendix.A with in-house experiments on MoE LMs trained on three corpora of increasing complexity: TinyStories, WikiText-103, and OpenWebText.
> For each dataset we:
>  - estimate intrinsic dimension $d$ of hidden states using the Levina--Bickel MLE on GPT-2 representations (Table 1);
>  -  run model-scaling and data-scaling sweeps with MoE Transformers and fit exponents $(\widehat{\alpha}_N,\widehat{\alpha}_D)$ from log–log learning curves;
> - infer a smoothness $\beta$ such that the theoretical exponents
>     $\alpha_N = 2\beta/d$ and $\alpha_D = 2\beta/(2\beta+d)$ match the empirical fits and satisfy the internal identity
>     $\alpha_D \approx \alpha_N/(1+\alpha_N)$ (Table 2, Figs.1--2).
>
>
> These experiments do not claim numerical tightness of the bound, but they provide concrete evidence that (i) the scaling exponents and their relationships are borne out on real MoE LMs, and (ii) the routing term behaves as predicted when we vary $(M,k)$.
>
> **2. Bounded-overlap assumption $s_0(d)$**
> The bounded-overlap parameter $s_0(d)$ arises from the manifold partition-of-unity construction: each point lies in at most $s_0(d)$ local charts, and the router is assumed to approximately respect this geometric locality (nearby points share a small set of experts).
> In practice, specialization could in principle break this if the router maps nearby points to disjoint expert sets. However:
>
> - the assumption is about the \emph{existence} of a good MoE approximant, not about arbitrary trained routers;
> - the new experiments suggest that fitted exponents remain consistent with the theory even when routers specialize, which indicates that effective overlap remains bounded in the regimes we study.
>
> We now call out $s_0(d)$ explicitly as a modeling assumption, and discuss scenarios where it might fail (e.g., highly non-local routing) in the limitations section.
>
> **3. Estimating $d$ and $\beta$; toy validation**
> We agree that the original description of $d,\beta$ estimation was too informal. The revision turns this into a concrete pipeline:
> - $d$ is estimated layer-wise from hidden states using the Levina--Bickel MLE with FAISS $k$-NN, repeated over subsamples; we report medians and median absolute deviations and then fix a dataset-specific $d$ (Table~1).
> - $\beta$ is inferred from empirical slopes $(\widehat{\alpha}_N,\widehat{\alpha}_D)$ by solving
>     $\widehat{\alpha}_N \approx 2\beta/d$ and $\widehat{\alpha}_D \approx 2\beta/(2\beta+d)$, and checked via the identity $\alpha_D \approx \alpha_N/(1+\alpha_N)$.
> This yields $d$ and $\beta$ in a self-consistent way across three datasets, rather than a “list of suggestions.”
> Regarding toy manifolds with known $(d,\beta)$: due to space and compute we focused the new experiments on realistic LMs, but we agree this would be a clean additional sanity check.
>
> **4. Looseness of routing union bound and other upper bounds**
> You are right that the union bound over routing patterns is potentially loose. In the proof of the covering-number bound we control the number of routing configurations by
> $$
> \log|\Pi|
> \le
> L_T \ell k \log(eM/k),
> $$
> which is worst-case over all top-$k$ assignments. This is conservative, but:
>
> - it appears only as a **logarithmic** factor in the metric entropy and the generalization bound, so its looseness affects constants rather than exponents;
> - router specialization in practice can only **reduce** the effective number of configurations, making the bound safer rather than riskier.
>
> In the revised text we add a short discussion in Section~3.4 explicitly stating that (i) the routing term arises from a worst-case union bound, (ii) it is expected to be loose in absolute value, and (iii) its main role is to identify a routing-dominated regime and the logarithmic dependence on $M$ and $k$.
>
> We now clearly separate (i) exponent-level predictions, which are supported by our new experiments, from (ii) constant-level looseness, which is inherent to worst-case bounds. With the added empirical section, a concrete $d,\beta$ estimation procedure, and an explicit discussion of assumptions and looseness, the paper’s claims are now more carefully scoped and better grounded.

---

### Meta-Review · Area_Chair_JckZ · 2026-01-08

**Summary:**

While the paper addresses an important question—developing generalization bounds and scaling laws for Mixture-of-Experts Transformers—and contains several appealing theoretical ideas, there are serious concerns that ultimately warrant rejection. Most notably, multiple reviewers identified fundamental and persistent errors in the definition of the Transformer/MoE block, including an incorrect architectural specification that remained inconsistent or ambiguously corrected across revisions, with broken references and definition mismatches. For a theory paper centered on architectural properties, such errors are particularly concerning: even if the authors argue that the results depend only on abstract Lipschitzness and parameter counts, the back-and-forth to precisely and consistently define the underlying model undermines confidence in the rigor and correctness of the analysis.

Although the authors made a substantial effort in the rebuttal - adding experiments, expanding proofs, clarifying notation, and situating the work relative to prior theory - the core issue around architectural definitions was not cleanly resolved within the review cycle, and at least one reviewer explicitly reported that the claimed fixes were not reflected in the revised manuscript (also indicating a score decrease). This raises concerns about verifiability and suggests that the work would benefit from another full revision and review round to ensure internal consistency, correct formalization, and careful proofreading. Given the standards expected for theoretical contributions, especially those aiming to guide large-scale model design, these unresolved issues outweigh the otherwise interesting conceptual contributions, and the paper should therefore be rejected in its current form.

**Reviewer Concerns:**

I think the authors did mostly an OK job in clarifying issues and answering questions raised by the reviewers, yet the issue around a precise definition of what model is considered is concerning, especially for theory-oriented paper. The authors tried to clarify that issue, but were again somewhat imprecise in doing so. While I do think this could be corrected, my assessment is that one would need another round of reviews to be fully confident in the presented results.

**Reviewer Scores:**

Some reviewers have already indicated their tendency towards an increase/decrease of their scores, with one reviewer specifically indicating a decrease. I am fairly sure that even another round of discussion would not have changed the reviewer's assessment on the model specification issue (even after the authors repeatedly tried to clarify that).

---

### Decision · Program_Chairs · 2026-01-26

Reject